**Review article**

# Sample delivery methods for protein X-ray crystallography with a special focus on sample consumption

Abhik Manna [1,2], Diandra Doppler [1,2], Manasa P. Sripati [2], Mukul Sonker [1,2] & Alexandra Ros [1,2] ✉

Serial crystallography (SX) has revolutionized structural biology by enabling high-resolution structure determination for important classes of proteins, including the study of relevant biomolecular reaction mechanisms. However, one of the ongoing challenges in this field remains the efficient use of precious macromolecule samples whose availability is often limited. Reducing sample consumption is thus critical in maximizing the potential of SX conducted at powerful X-ray sources such as synchrotrons and X-ray free-electron lasers (XFEL) to expand to a broader range of significant biological samples, gaining insights into unraveled biological reaction mechanisms. This review focuses on three primary sample delivery systems: fixed-targets, liquid injection, and hybrid methods, each with distinct advantages and limitations concerning sample consumption. The progress and challenges associated with these methods, highlighting advancements in reducing sample consumption and thus enabling the study of more diverse biological samples, are summarized. We compare the currently reported sample delivery methods in view of the minimum amount of sample required to obtain a full data set and discuss how the current approaches compare to this theoretical minimum. With this overview, we aim to provide a critical and comprehensive assessment of the current methods and experimental realizations for sample delivery in SX with proteins.

X-ray crystallography is a powerful technique for determining atomic resolution structures of molecules by analyzing the X-ray diffraction caused by their crystal lattice[1,2]. Since its inception, X-ray crystallography has enabled high-resolution structural determination of a plethora of biomolecules, with over 200,000 structures deposited in the PDB on proteins alone (www.wwpdb.org) and even played a key role in providing structural insights to combat the recent Covid-19 pandemic[3]. Despite its tremendous potential, the sample preparation requirements in terms of crystal size and quantities have always been a major challenge holding back the advancement of X-ray crystallography[4,5]. In traditional macro-scale X-ray crystallography,

the requirements of >100 μm crystals were a daunting challenge for structural biologists, especially considering the cumbersome task of producing pure protein samples for macromolecular crystallization[6–10]. However, more recent developments in brilliant synchrotron sources with micro-focused beams (below 10 μm in diameter) utilizing smaller crystals further piqued interest in the field of protein X-ray crystallography[11–15]. Concurrently, Neutze and coworkers[16] proposed the possibility of obtaining high-resolution structural information from micro-to-nano-sized crystals using ultra-bright femtosecond X-ray pulses. This newly proposed "diffraction before destruction" concept hypothesized that short ultra-bright X-ray

[1]School of Molecular Sciences, Arizona State University, Tempe, AZ, USA. [2]Center for Applied Structural Discovery, The Biodesign Institute, Arizona State University, Tempe, AZ, USA. ✉e-mail: aros2@asu.edu

pulses would be suitable to yield diffraction patterns before destroying the sample and paved the way for the development of crystallography methods at ultra-bright short-lived X-ray sources like X-ray free-electron lasers (XFELs)[17,18], commencing a paradigm shift that revolutionized the field of X-ray crystallography[19–22]. Using showers of small microcrystals in a serial manner at XFELs, liberated from large crystal size requirements, has opened the field of X-ray crystallography to a wider array of analytes and biomolecules, including single nanoparticles[23–29]. The methods developed in serial crystallography with XFELs then also allowed similar approaches at 3rd and 4th generation synchrotron sources[18,30,31].

However, these advancements came with another challenge-in-disguise due to the pulsed nature of these bright X-ray sources operating at a pulse rate ranging from 30 Hz to 4.5 MHz[20,22]. Since these pulses are fairly short-lived (femto-to-microseconds), a crystal is exposed only once to X-rays to record a diffraction pattern before it is destroyed or damaged due to the high radiation dose. New crystals must therefore be continuously replenished in the path of the X-ray beam to acquire a complete dataset, typically requiring around ten thousand patterns to resolve the electron density map of the protein structure, which depends on the space group and symmetry, as well as convergence of the data collection statistics. This technique is often referred to as Serial Crystallography (SX) and has been one of the most commonly used crystallography techniques in the past decade, regardless of its great appetite for sample consumption[3,4,20,22,32]. The new requirement of smaller microcrystals for SX not only simplified the sample preparation process but also ignited the evolution of new sample delivery methods from the traditional goniometry, as the crystal sizes were now compatible with the dimensions of fast-emerging microfluidic technologies[23,33–38]. Additionally, the need for rapidly replenishing crystals to keep up with high X-ray repetition rates gave rise to the low-volume fixed-target and liquid sample injection techniques, where the slurry of microcrystals can be directly loaded and scanned on a "fixed" microfluidic device or jetted as a liquid stream in the path of the X-ray beam to enable SX[39–42].

The first demonstration of such an SX experiment was reported by Chapman and collaborators[26] where "millions to billions" of 0.2–2 μm-sized crystals of Photosystem I were serially injected using a liquid jet in the path of an X-ray beam at the Stanford Linear Accelerator Center (SLAC) National Accelerator Laboratory Linac Coherent Light Source (LCLS) XFEL[43] to obtain a protein structure of Photosystem I at 8.5 Å. Later, Boutet et al.[44], utilizing the higher X-ray energy upgrade of the Coherent X-ray Imaging (CXI) instrument at LCLS[45], resolved the structure of Lysozyme at 1.9 Å from <3 μm crystals. These pioneering studies, successfully demonstrating SX with XFELs and thus termed serial femtosecond crystallography (SFX), unfolded a new era of serial crystallography experiments with XFELs, and several more complex biomolecular systems, including G-protein coupled receptors, were determined with SX soon after[42,46,47]. While the initial successful SFX experiments focused on static crystallography, it was soon realized that the technique could prove useful in studying macromolecular reaction mechanisms by the initiation of a reaction either by light (for light-activated proteins) or by rapid mixing of reactants with proteins (such as substrates with enzymes). The method was termed time-resolved SFX (TR-SFX)[48–52]. TR-SFX experiments were performed using photosensitive proteins and a pump-probe laser to study light-activated proteins with typical reaction timescales of microseconds to femtoseconds[48,51–59]. Another widely explored TR-SFX method is termed mix and inject serial crystallography (MISC), where reactants and substrates are mixed with protein crystals to induce conformational changes directly before X-ray exposure to study the structural changes in seconds to sub-millisecond time scales[60–64]. Evolving from static structures to obtaining TR-SFX led to coining the term "molecular movies," allowing to study various biomolecular reactions in real-time. However, the extensive applications of SFX have been largely constrained by the limited availability of XFEL experimental time. Meanwhile, the continued advancements in next-generation synchrotron facilities have increased opportunities for conducting SX experiments[11,18,30,31]. Due to the relatively longer X-ray exposure times at synchrotrons compared to XFELs, this method is known as serial millisecond crystallography (SMX). Consequently, the use of SMX has been expanding with several experiments published to date[65–67]. For simplicity, we will use the term SX (serial crystallography) throughout this review as a broader term encompassing both SFX and SMX. Despite the remarkable success and interest SX has gained, the high amount of protein consumed to obtain a structure still concerns the scientific community. Essentially, the sample injected between the X-ray pulses goes to waste and never interacts with the X-ray beam[4]. TR-SX further exacerbates the sample consumption requirement of these experiments, as the sample consumption is multiplied for each time point probed. In the pioneering SX experiments, samples were injected at high sample flow rates (>10 μL/min) and crystal density (~10⁹ crystals/mL) for hours or days to obtain the required amount for a dataset, which signified that a few grams of purified protein needed to be available for the experiments[26,44]. A few grams might not be a large quantity for off-the-shelf proteins like Lysozyme, a well-characterized and commonly used model protein in crystallography[44,68–72], but may require a considerable amount of experimental time, effort, and biochemical infrastructure for biologically and medically relevant as well as hard-to-crystallize proteins, making their use in SX initially prohibitive[8,23,38,46,73,74].

This challenge of sample consumption, along with well-timed advances in microfabrication and 3D printing techniques in the last decade, led to a wildfire of new sample injection techniques allowing SX as well as other approaches to conserving precious crystal samples[4,5,75–77]. As a result, the sample consumption that used to be a few grams of proteins in the early SX experiments has shrunk down to microgram amounts in more recent studies[3,4,19,20,78,79]. The approaches for sample delivery, however, vary significantly, from continuous injection methods over droplet approaches and fixed-target devices to hybrid methods thereof. Some are strikingly straightforward to integrate into a variety of existing X-ray crystallography end stations, and some are specific to the requirements of a particular instrument. The reported protein consumption, thus, not surprisingly, varies significantly between methods and experiments. To allow a comparison of sample consumption techniques and provide a critical assessment of the state-of-the-art approaches, we establish an ideal sample consumption requirement for SX, where 10,000 indexed patterns are sufficient for a full dataset, assuming that each crystal hit by an X-ray pulse provides an indexable diffraction pattern. Further assuming a microcrystal size of $4 \times 4 \times 4$ μm and a protein concentration in the crystal of ~700 mg/mL (based on an example for a small 31 kDa protein such as the enzyme NAD(P)H:quinone oxidoreductase 1 (NQO1)[80]), we estimate that ~450 ng of protein will be ideally required for an SX experiment, providing a full dataset.

In this scenario, we do not consider sample needs for screening and finding optimized crystallization conditions. These can be a challenging topic on their own, and we assume that protein crystallization conditions to generate micrometer-sized crystals were already established prior to the estimates on sample consumption used in this review. Referring to these prerequisites, this review provides an overview of sample delivery approaches reported in the literature and discusses different sample injection techniques used in SX (and TR-SX) over the past decade, with a special focus on the sample consumption aspect of each protein crystallography experiment. The sample injection techniques discussed here are categorized as: (1) fixed-target systems exploring various low X-ray background materials; (2) liquid injection systems including continuous injection, droplet-based injection, high-viscosity extruders, etc.; and (3) hybrid methods combining a variation of the two aforementioned approaches. The major

advantages and disadvantages of these individual injection methods are reviewed, and protein sample consumption is compared.

## Fixed-target sample delivery methods

Positioning an ordered or non-ordered array of protein crystals on a solid support has been proposed to alleviate issues related to sample consumption[4,19,75,81–83]. The idea of the so-called fixed-target method is very similar to traditional goniometer-based crystal delivery to synchrotron sources[12,19,84,85]. In contrast to the traditional method, where a single crystal is mounted in the X-ray beam path, recent fixed-target systems can deliver multiple crystals on a solid (or "fixed") support where crystals are held in place at certain positions (as shown in Fig. 1a), making it amenable for serial crystallography experiments. The fixed-target systems have further revolutionized the field of X-ray crystallography by allowing on-chip crystallization and subsequent structure determination, which eliminates the need to harvest the precious crystals[4,19,75,83].

Fixed-target delivery systems constitute a promising approach for reducing sample consumption. Assuming that the stage translation on which the fixed targets are mounted can be ideally tuned to the repetition rate of a pulsed X-ray beam, and further assuming that a complete dataset is collected with 10,000 diffraction patterns, the same number of crystals is required. For an "ideal" situation, let's assume that 10,000 crystals of $4 \times 4 \times 4$ μm size (we chose this size here as the same assumption is used in the liquid injection techniques discussed below, which was adapted to the typical diameter of a jet) are placed inside a sheet-on-sheet type fixed-target device and that they are arranged in a $100 \times 100$ array. We further assume that the beam diameter is 4 μm, and to avoid radiation damage to the neighboring crystals, there is a 20 μm spacing between each crystal. To fulfill these assumptions, a fixed-target device of ~$2400 \times 2400$ μm area with a depth of 5 μm (slightly higher than the crystal dimension) is required, which will hold a volume of ~29 nL. Based on the unit cell volume and molecular weight of human NQO1 crystals[80], the protein concentration per unit cell results in ~700 mg/mL. Therefore, for 10,000 crystals with the above-mentioned dimension, ~450 ng of protein will be consumed during the crystallography experiment. Furthermore, based on the number of crystals and the volume needed to fill the array in a hypothetical fixed-target device, the required crystal concentration is ~$3.5 \times 10^8$ crystals/mL. These estimates benchmark the amount of protein theoretically required for obtaining a structure. We are contrasting it in our discussion below with the crystal slurry volumes used in fixed-target approaches and the associated protein amount in milligrams, based on conditions reported in the literature (listed in Table 1).

It is important to note that fixed-target sample delivery systems are not free from flaws and that other considerations related to sample delivery influence sample consumption. One major disadvantage of the fixed-target system is the dehydration of crystal samples during X-ray crystallography experiments, which can be mitigated either by the introduction of a humidification system to the experimental chamber or by enclosing the sample delivery chips with a material having low water permeability[19,75,83]. This introduces another disadvantage of background contribution from the chip materials themselves[75,83,86,87]. Furthermore, crystal slurries need to be loaded in fixed-target devices, which is typically achieved through manual pipetting, thus realistically requiring several to tens of μL for sample loading. Sample loading procedures also need to be gentle enough not to include shear forces to destroy crystals while loading (unless crystallization takes place in the fixed-target devices). We therefore also discuss and compare various strategies explored by different research groups to reduce sample consumption, reduce background scattering, and prevent dehydration during crystallography experiments in the next sections. To ease the discussion, we grouped the fixed-target sample delivery devices by the material used to fabricate them.

### Silicon-based fixed-target methods

Silicon as well as silicon nitride ($Si_3N_4$) and to a lesser extent $SiO_2$ have been used as support material for delivering protein crystals to X-ray beams (an example is shown in Fig. 1b). Advances in Si-microfabrication have made possible the design of large arrays of fixed-target devices holding thousands of protein crystals.

In 2012, Zarrine-Afsar et al.[88] demonstrated the use of a patterned Si mesh, sandwiched between two Kapton films, to deliver lysozyme and ferritin crystals to a synchrotron source. The authors used differently sized glass beads with the crystal suspension, which they claimed induced different crystal orientations inside the Si apertures. The major downside of this method was the background contribution from the glass beads, which was eliminated by Mueller et al.[89] in 2015 (Fig. 1b). Random orientations of crystals inside the different apertures of the Si chip were achieved just by filling the Si mesh with a pipette tip and blotting off the excess liquid to reduce background contributions from the mother liquor. This fixed-target design further demonstrated the possibility of light-induced TR-SX with fixed-target devices by collecting diffraction data from sperm whale myoglobin while probing dynamics of CO dissociation at time points ranging from 200 fs to 32 ps.

In 2014, Hunter et al.[41] performed fixed-target X-ray crystallography experiments at the LCLS XFEL using a patterned Si mesh. Microcrystals of rapid encystment protein (REP24) were deposited

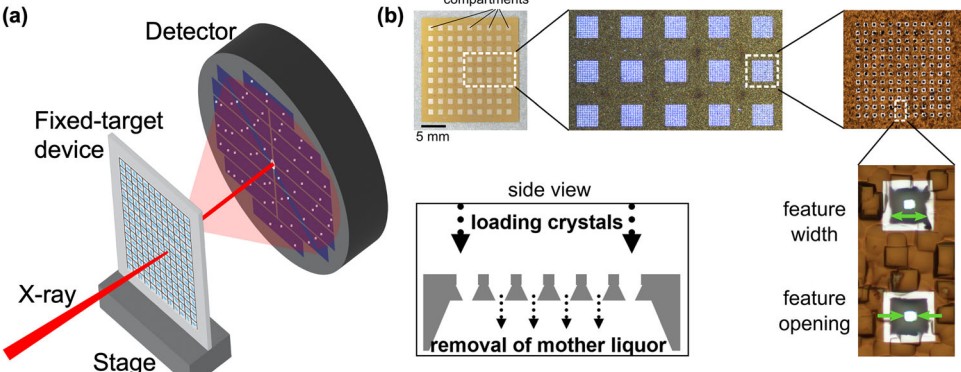

**Fig. 1 | Fixed-target sample delivery devices. a** Schematic representation of a fixed-target sample delivery system with an array of crystals positioned on a solid support. The solid support is mounted on a stage to position a specific crystal in the path of the X-ray beam to collect diffraction data on the detector. **b** Si-based fixed-target device developed by Mueller et al.[89] where crystal slurry was loaded with a pipette tip, and the excess mother liquor was removed from the back of the chip (reproduced with permission from ref. 89).

**Table 1 | Overview of fixed-target systems reported in the literature, including details on X-ray source, analyte, protein concentration, crystal concentration, and sample consumption**

| Author | Device material | X-ray source, beamline | Analyte | Protein conc. (mg/mL) | Crystal conc. (particles/mL) | Sample consumption [volume/amount[a]] |
|---|---|---|---|---|---|---|
| **Silicon-based fixed-target devices** | | | | | | |
| Zarrine-Afsar et al.[88] | Si | Swiss Light Source (SLS), X10S A | Lysozyme (1), ferritin (2) | (1) 60, (2) 30 | NI | <1 µL/–(1) 0.06 mg, (2) 0.03 mg |
| Mueller et al.[89] | Si | SLAC, LCLS, XPP | Thaumatin (1), Proteinase K (2), sperm whale myoglobin | (1) 50, (2) 20, (3) 40–50 | NI | 0.9–10.35 µL/(1) 0.05–0.52 mg, (2) 0.02–0.21 mg, (3) ~0.04–0.5 mg |
| Hunter et al.[41] | Si; Si$_3$N$_4$ | SLAC, LCLS, CXI | Rapid encystment protein | 14.4 | 1×10$^9$ | ~38 µL$^b$/0.55 mg |
| Roedig et al.[90] | Si | Diamond Light Source, I02 | Polyhedrin, lysozyme | NI, 50 | 1–2×10$^3$ | 1–3 µL/0.05–0.15 mg |
| Murray et al.[91] | Si; Si$_3$N$_4$, Kapton | Stanford Synchrotron Radiation Light Source (SSRL), BL12-2 | Lysozyme | 20 | NI | <2 µL/0.04 mg |
| Oghbaey et al.[79] | Si | SLAC, LCLS, XPP | Myoglobin | 50–60 | NI | 60–80 µL/3–4.8 mg |
| Owen et al.[92] | Si; Mylar | Diamond Light Source, I24 | Orthorhombic sperm whale myoglobin | 55–60 | NI | 60–80 µL/3.3–4.8 mg |
| Lieske et al.[93] | Si | SLAC, LCLS, MFX, and CXI | Proteinase K (1), Thermolysin (2) | (1) 15, (2) 20 | NI | 2–3 µL for roadrunner 1, ~100 µL$^b$ for roadrunner 2/(1) 0.05 mg, (2) 0.06 mg for roadrunner 1 and (1) 1.5 mg, (2) 2 mg for roadrunner 2 |
| Tolstikova et al.[94] | Si | European Synchrotron Radiation Facility (ESRF), ID09 | Lysozyme (1), Proteinase K (2) | (1) 40–60, (2) 20 | NI | ~100 µL$^b$/(1) 4–6 mg, (2) 2 mg |
| Ebrahim et al.[95] | Si | Diamond Light Source, I24 | Copper nitrite reductase | 20 | NI | 100–200 µL/2–4 mg |
| **Single polymer-based fixed-target devices** | | | | | | |
| Ng et al.[96] | COC | SSRL, BL1-5 | Bacteriorhodopsin | 50 | NI | 30 µL/1.5 mg |
| Pinker et al.[98] | COC | SLS, X10SA and X06DA | Lysozyme (1), thaumatin (2), insulin (3) | (1) 50, (2) 30, (3) 20 | NI | ~2 µL$^b$/(1) 0.1 mg, (2) 0.06 mg, (3) 0.04 mg |
| De Wijn et al.[99] | COC | SLS, PXII, and PXIII ESRF, ID30B French Synchrotron Facility (SOLEIL), PROXIMA-2A | Protease 1 (1), CCA-adding enzyme (2), lipase (3), Hemoglobin (4) | (1) 7.4, (2) 5.5, (3) 30, (4) 20 | NI | ~2 µL$^b$/(1) 0.02 mg, (2) 0.01 mg, (3) 0.06 mg, (4) 0.04 mg |
| Karpik et al.[100] | COC | SLS, PXI X06SA SwissFEL | Lysozyme | 50 | NI | ~2 µL$^b$/0.1 mg |
| Martiel et al.[101] | COC | SLS, PXI | Rhodopsin-miniG$_o$ | 25 | 3.4×10$^6$ | ~2 µL$^b$/0.05 mg |
| Manna et al.[72] | COC | ESRF, ID29 | Lysozyme | 40 | NI | 30 µL/1.2 mg |
| Emamzadeh et al.[105] | COP | ESRF, FIP-BM30A | Lysozyme (1), trypsin (2) | (1) 22–30, (2) 25–30 | NI | 160 µL/(1) 3.52–4.8 mg, (2) 4–4.8 mg |
| Cohen et al.[106] | PC | SSRL | CpI [FeFe]-hydrogenase from Clostridium pasteurianum | 30 | NI | ~2.5 µL$^b$/0.08 mg |
| Lee et al.[108] | Kapton | Pohang Accelerator Laboratory X-ray Free Electron Laser (PAL XFEL) | Glucose isomerase (1), lysozyme (2) | (1) NI, (2) 50 | 3×10$^8$ (1), 5×10$^7$ (2) | 20 µL/(1) NI, (2) 1 mg |
| Lee et al.[109] | Kapton | PAL XFEL, NCI | Proteinase K | 80 | 5×10$^7$ | 15.7 µL/1.25 mg |
| Nam et al.[111] | Kapton | Pohang Light Source, 11C | Lysozyme | 50 | NI | ~8 µL$^b$/0.4 mg |
| Illava et al.[112] | Kapton | National Synchrotron Light Source II, XF17ID2 Cornell High Energy Synchrotron Source (CHESS), ID7B2 | fluoroacetate dehalogenase (1), lysozyme (2), human glutaminase C (3) | (1) 18, (2) 100, (3) 20 | NI | 5–10 µL/(1) 0.09–0.18 mg, (2) 0.5–1 mg, (3) 0.1–0.2 mg |
| Doak et al.[113] | Mylar | Spring-8 Angstrom Compact free-electron Laser (SACLA), BL2 | Lysozyme | 32 | NI | 14 µL/0.45 mg |
| Feld et al.[110] | SU-8 | SLAC, LCLS | Bacteriorhodopsin | 1–3 | NI | 2 µL/0.002–0.01 mg |

**Table 1 (continued) | Overview of fixed-target systems reported in the literature, including details on X-ray source, analyte, protein concentration, crystal concentration, and sample consumption**

| Author | Device material | X-ray source, beamline | Analyte | Protein conc. (mg/mL) | Crystal conc. (particles/mL) | Sample consumption [volume/amount[a]] |
|---|---|---|---|---|---|---|
| Saha et al.[114] | SU-8 | SSRL, BL-12-1 | Lysozyme (1), Ca2+/calmodulin-dependent protein kinase II β hub domain (2) | (1) 120, (2) 29 | NI | 4 µL/(1) 0.48 mg, (2) 0.12 mg |
| Saha et al.[115] | SU-8 | SSRL, BL-12-1 | Lysozyme (1), thaumatin (2), proteinase K (3) | (1) 200, (2) 25, (3) 20 | NI | 0.5 µL/(1) 0.1 mg, (2) 0.0125 mg, (3) 0.01 mg |
| **Multiple polymer-based fixed-target devices** | | | | | | |
| Dhouib et al.[86] | PDMS, PMMA, SU-8, COP, COC | ESRF, FIP-BM30A | Lysozyme (1), thaumatin (2) | (1) 80, (2) 34–47 | NI | 1.2 µL/ (1) 0.10 mg, (2) 0.04–0.06 mg |
| Guha et al.[87] | COC, PDMS | LS-CAT, Argonne National Lab | Lysozyme (1), Thaumatin (2), Ribonuclease A | (1) 100, (2) 82, (3) 229 | NI | 1.4 µL/(1) 0.14 mg, (2) 0.11 mg, (3) 0.32 mg |
| Khvostichenko et al.[116] | COC, PDMS | Advanced Photon Source (APS), Argonne National Lab (ANL), 21-ID-F | Rhodobacter sphaeroides | 6 | NI | ~4 µL/0.2 mg |
| Lyubimov et al.[117] | PDMS, PMMA | SLAC, LCLS, XPP SSRL, BL 12-2 | Lysozyme | 20 | NI | ~5 µL[b]/0.1 mg |
| Gilbile et al.[118] | COC, PMMA | SSRL, BL 12-1 | Lysozyme (1), thaumatin (2), Concanavalin-A (3), Bovine Liver Catalase (4), | (1) 30, (2) 25, (3) 70, (4) 40 | NI | 8–10 µL/(1) 0.24–0.3 mg, (2) 0.2–0.25 mg, (3) 0.56–0.7 mg, (4) 0.32–0.4 mg |
| Liu et al.[119] | COC, PMMA | SSRL, BL 12-1 SLAC, LCLS, MFX | Lysozyme (1), thaumatin (2), | (1) 50, (2) 25 | NI | 15 µL/ (1) 0.75 mg, (2) 0.38 mg |
| Sui et al.[120] | COC, PMMA, Graphene | APS, ANL, 14-ID-B | Lysozyme | 80 | NI | 16 µL/1.28 mg |
| Sui et al.[121] | COC, PMMA, Graphene | University of Massachusetts Amherst Institute for Applied Life Sciences Biophysical Characterization Facility | Lysozyme | 80 | NI | 3.2 µL/ 0.26 mg |
| Shelby et al.[122] | PMMA, Graphene | SLAC, LCLS, MFX, CXI | Rapid encystment protein | 14.4 | 2.2 × 10^6 | ~20 µL[b]/0.29 mg |
| Zhao et al.[123] | PMMA, Mylar | Shanghai Synchrotron Radiation Facility (SSRF), BL 18U1 | Lysozyme (1), proteinase K (2) | (1) 40, (2) 25 | NI | ~6 µL[b]/ (1) 0.24 mg, (2) 0.15 mg |
| Zhao et al.[124] | PMMA, Mylar | SSRF, BL 18U1 | Lysozyme | 40 | NI | 5 µL/0.2 mg |
| Carrillo et al.[125] | COC, COP | SwissFEL, SwissMX | Lysozyme | 25 | 3.2 × 10^8 | ~1–10 µL[b]/0.03–0.3 mg |

Sample consumption is listed in volume and protein amount. NI refers to data not included. N/A refers to not applicable. Numbers in parentheses refer to more than one analyte reported in a particular publication (and the numbering is defined in the "analyte" column).
[a]The protein amount is estimated from the reported protein concentration in the mother liquor, considering that all protein molecules were converted to crystals, taking into account the reported volumes consumed unless otherwise stated.
[b]For these fixed-target devices, the sample volumes were not reported and were therefore estimated from the provided device dimensions.

onto a 50 nm thick $Si_3N_4$ membrane while the dehydration was prevented by embedding the crystals in Paratone-N. The authors demonstrated that a complete dataset could be collected using ~38 μL of protein sample. Furthermore, in 2015, Roedig et al.[90] developed a fixed-target sample holder from single-crystalline silicon, featuring an array of micropores designed to accommodate thousands of microcrystals. The authors claimed that the incorporation of single-crystalline silicon, combined with the effective removal of mother liquor through blotting, resulted in exceptionally low background scattering from the chip. In the same year, Murray et al.[91] developed a Si mesh fixed-target device achieving a reduction of background scattering from the chip material by sandwiching in between an 8 μm-thick Kapton foil on one side and 150 nm $Si_3N_4$ on the other. Another patterned Si chip was developed by Oghbaey et al.[79] in 2016 that could collect diffraction data from 11,664 different positions. Each of these positions was tailored to hold one crystal. The authors mapped the chip with in situ spectroscopy to determine the positions of the crystals before the experiment. With the predetermined positions, they could reach a ~100% hit rate with an approximate indexing rate of 50%. This same approach was used by Owen et al.[92] in 2017 to develop a fixed-target system where the patterned Si array was enclosed with two 6 μm Mylar films. Using this system, the authors could obtain myoglobin and MetMb structures at 1.8 Å and 1.7 Å, respectively.

Further, in 2019, Lieske et al.[93] demonstrated a new micropatterned Si chip where protein crystals could be grown on-chip and which could be used for in situ diffraction data collection. Crystallography experiments were performed under cryogenic conditions at synchrotron sources and at room temperature at XFEL sources. Furthermore, these chips were capable of performing on-chip TR-SX experiments. In the same year, this system was used by Tolstikova et al.[94] to demonstrate high-speed (1 kHz) sample delivery and structure determination at a synchrotron facility. In addition, Ebrahim et al.[95] modified a previously developed Si chip[79,89] that enabled separation and structure determination of crystal polymorphs as well as tracking of radiation damage in microcrystals. Using up to 200 μL crystal suspension, the authors could differentiate between two polymorphs with different unit cell sizes to determine two independent structures within a population of copper nitrite reductase crystals.

The main advantage of Si-based fixed-target systems is the low background contribution from the material, as in most cases, Si was etched away completely to create windows where crystals are held in place for data collection. If ideally centered, the X-ray beam hits only the thin layer of mother liquor and the crystal itself. Fixed-target devices for experimental chambers at atmospheric pressure are now widely used, but efficient approaches to prevent dehydration are still under development and optimization for vacuum experimental chambers, including the minimization of background scattering from materials used to prevent dehydration. In addition, Si-based fixed target devices pose the inherent risk of hitting the Si material if the crystal-containing window is not aligned properly to the X-ray beam, which can severely damage the detector[41].

The fixed-target realizations described above show that various research groups have taken advantage of Si or $Si_3N_4$ to develop fixed-target systems to overcome different issues related to SX. As outlined in Table 1, the sample volume required for these Si fixed-target devices varies from ~1 μL up to 200 μL, which corresponds to protein amounts consumed in the range of a few mg (1–5 mg), with some reporting sub-mg to μg amounts, in some cases as low as several tens of micrograms[88,89,91,93]. We note that the sample consumption for the works listed in Table 1 was estimated based on reported protein concentrations in the mother liquor (assuming most protein molecules form crystals) and volumes corresponding to the reported device dimension. While a comparison with the ideal sample consumption as outlined above may only hold considering protein crystals of similar size and originating from similar protein concentrations in the mother

**Table 2 | Background scattering of different polymers used in fixed-target sample delivery devices**

| Polymeric material | Background scattering ring position at maximum (Å) |
|---|---|
| Nylon[208] | 3.4 and 4.5 |
| COC[118] | 5–6 |
| Kapton[109] | 15.3 |
| SU-8[114] | 10 |
| Mylar[113] | 4.7 |
| PDMS[117] | 7.5 |

liquor, the devices reported in the literature still require at least two orders of magnitude higher protein quantities than ideally estimated.

## Polymer-based fixed-target systems

Polymer materials have gained interest as the material of choice for fixed-target microfluidic devices due to their cost-effectiveness and ease of fabrication compared to Si-based devices[96]. A wide variety of polymeric substances, including Teflon, Nylon, polydimethylsiloxane (PDMS), polycarbonate (PC), polymethylmethacrylate (PMMA), cyclic olefin copolymer (COC), cyclic olefin polymer (COP), SU-8, polyethylene terephthalate (Mylar) or polyimide (known under the trade name Kapton), have been used to develop fixed-target sample delivery systems[4,97]. Each of these materials exhibits characteristic background scattering in the diffraction image, which appears in a diffraction ring (see Table 2). These polymeric materials have been implemented to develop fixed-target systems either exclusively or in combination with other polymeric substances, and we review them below with information on sample consumption (listed in Table 1).

**Single polymer-based fixed-target systems.** COC is a popular choice of material for fixed-target sample delivery systems due to its compatibility with a wide variety of solvents, low vapor permeability, and high X-ray transparency. In 2009, Ng et al. employed COC to develop a 1 mm-thick device in a microscope slide format, including nine individual channels dedicated to protein crystallization. The authors demonstrated the applicability of this chip for in situ structure determination by determining the crystal structure of bacteriorhodopsin at 2 Å resolution[86,96]. In 2013, Pinker et al.[98] developed a three-layer microfluidic device with 8 individual channels for on-chip crystallization and structure determination. This 3-layer device consisted of a 600 μm-thick fluidic layer, a 600 μm-thick cover slide, and a 3 mm-thick COC frame for the rigidity of the entire assembly. This work was further improved by de Wijn et al.[99] in 2019 by changing the device structure from three layers to two layers. In this case, the fluidic layer was 1 mm thick, and the cover slide was 100 μm thick. The authors performed on-chip crystallization and structure determination at four different beamlines to demonstrate the applicability of the device. Furthermore, Karpik et al.[100] developed an ultrathin perforated COC film (2–3 μm) on a polymer frame in 2020 to reduce the background scattering from the material. This same system was used by Martiel et al.[101] in 2021 to demonstrate the ability of the devices to deliver membrane protein crystals in the sponge phase. The schematic of this system is shown in Fig. 2a. In 2024, Manna et al.[72] developed a COC-based fixed-target device for capturing crystals in predefined positions for precise positioning on the X-ray beam path for diffraction data collection. Mainly targeted towards the newly developed compact X-ray light source (CXLS)[102–104], this device was equipped with compatibility for data collection inside vacuum experimental chambers as high as $10^{-5}$ mbar. Additionally, the device was used at the European Synchrotron Radiation Facility (ESRF, France) to obtain a high-resolution (1.6 Å) lysozyme structure. Moreover, the authors improved the design further through numerical simulation studies to increase the number of predefined positions up to 18,000 per device.

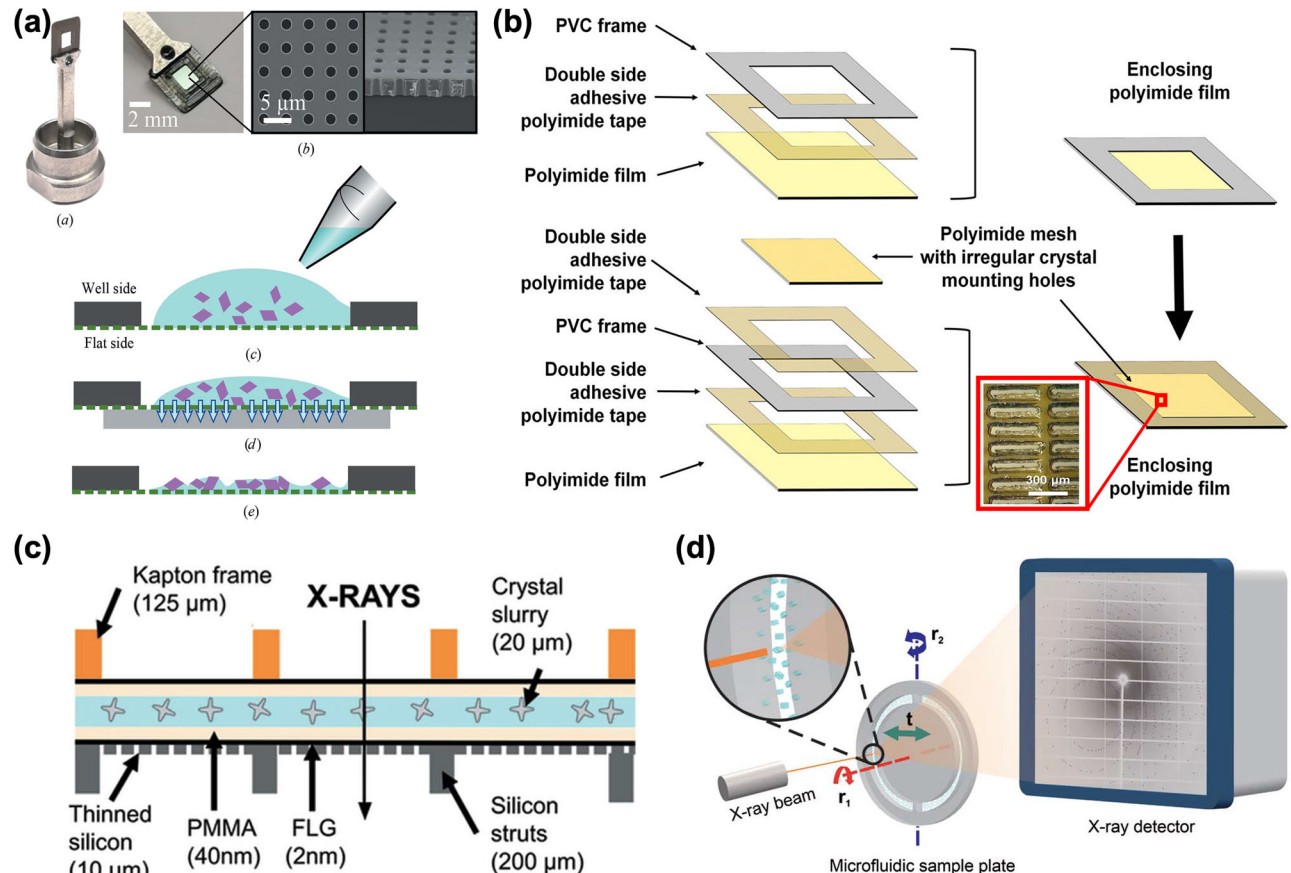

**Fig. 2 | Examples of polymer-based fixed-target devices. a** Fixed-target system developed by Martiel et al.[101] where a microporous COC membrane (3 μm) was developed as a sample support. This panel shows an image of the device mounted on a holder and a scanning electron microscope (SEM) image of the pores on the COC membrane. Schematics of the filling process are also shown, where devices were filled from the well-side and excess mother liquor was removed from the bottom using blotting paper. Reprinted with permission from IUCrJ. **b** Polyimide-based fixed-target system developed by Nam et al.[111], which contained a polyimide mesh with irregularly shaped holes (shown in red rectangular inset). A mesh was enclosed between two polyimide films to prevent crystal dehydration. **c** A fixed-target system developed by Shelby et al.[122] in combination with graphene to prevent the dehydration of samples. This panel shows the schematics of a cross-sectional view of the assembled device. Reprinted with permission from IUCrJ. **d** A rotation-based fixed-target system developed by Zhao et al.[124]. This panel demonstrates the overall design of the device. Reprinted with permission from IUCrJ.

Other polymer materials have also been employed for fixed-target devices. Emamzadah et al.[105] developed a COP-based fixed-target device in 2009, which was used for on-chip crystallization and structure determination. The authors used COP to eliminate the issue of water-vapor permeability in devices made from more commonly used silicon elastomers. Moreover, COP could be used to obtain in situ diffraction data from crystals grown on-chip. Furthermore, in 2014, Cohen et al.[106] reported a fixed-target grid developed at the Stanford Synchrotron Radiation Light Source (SSRL) using PC. The fixed-target grid contained holes of 125 μm, 200 μm, and 400 μm diameter to accommodate different-sized crystals. This simple fixed-target device could be used for both filling with pre-grown crystals as well as to perform in situ crystallization.

Kapton is another widely used material for fixed-target systems owing to the low X-ray background scattering from the material[107]. Additionally, this material has been used to prevent dehydration of the crystals during crystallography experiments. In 2020, Lee et al.[108] used two 25 μm-thick Kapton foils to enclose crystals embedded in agarose and gelatin. The crystal suspension was spread over an area of 20 × 20 mm from which up to 440,000 diffraction images could be collected. In the same year, Lee et al.[109] introduced a fixed-target holder that can hold four polyimide tubes with 100 μm ID and 13 μm thickness. Although the authors claimed these thin tubes to be one-dimensional and the capacity to hold a 3.9 μL sample volume to be

lower compared to a two-dimensional fixed-target system, the overall sample requirement of ~15 μL is more than reported on other fixed-target systems developed earlier[89,91,98,99,106,110]. In 2021, Nam et al.[111] developed an irregularly shaped polyimide mesh to facilitate random crystal orientation (device layer schematics shown in Fig. 2b). Using this system, they obtained the lysozyme structure at 1.65 Å at a synchrotron source using ~8 μL of crystal suspension. In the same year, Illava et al.[112] developed a Kapton-based fixed-target device to be employed at room temperature and cryogenic temperatures. The device was integrated into a device holder for easy sample loading and removal of excess liquid to reduce the background scattering further.

In 2018, Doak et al.[113] obtained a structure for lysozyme at an XFEL source at 2.1 Å resolution using a "chipless" design. The system was developed by sandwiching 14 μL of lysozyme crystal slurry between two 2.5 μm Mylar films. Two sheets of Mylar were sealed together at the edges by pressing them together between two O-rings. Furthermore, the photoresist SU-8 was used by Feld et al.[110] in 2015 to develop a plastic transmission electron microscope-like mesh to be used as a fixed-target system for crystal structure determination, requiring only 2 μL of sample. The same photoresist was used by Saha et al.[114] in 2023 to develop a centrifugal actuation-based microfluidic fixed-target device for on-chip crystallization and in situ diffraction data collection. This device had eight identical sections for crystallization, and the X-ray windows had an effective thickness of ~20 μm. The same group

reported another SU-8-based device in 2025, employing a grid design to hold crystals for delivery to an X-ray source[115]. While the overall design of the device had been previously described, the novelty of this work lies in demonstrating the ease of fabrication, cost-effectiveness, and suitability for large-scale production.

The major motivation for using polymeric materials is the reduction of background scattering, ease of use and fabrication, low sample consumption, and flexibility in design, such as adjustments of the device depth to account for particular crystal sizes. From Table 2, it can be concluded that all employed polymeric materials show characteristic background scattering at lower resolution (>3 Å), allowing high-resolution data collection. As summarized in Table 1, some of the fixed-target devices developed with a single polymer have consumed sample amounts as low as 0.01 mg[99,110,115], but generally not more than a few tens of mg were consumed. This is similar to the Si-based fixed-target devices and is correspondingly two or more orders of magnitude higher than the ideal sample consumption.

**Fixed-target devices based on multiple polymers.** Several fixed-target sample delivery devices were reported, where either a combination of multiple polymers or a combination of polymers and graphene was employed. The benefits of using these combinations will be discussed in the next few paragraphs. In 2009, Dhouib et al.[86] explored a simple design for on-chip crystallization for different chip materials like PDMS, PMMA, SU-8, COC, and COP. From the diffraction data analysis, they concluded that COC and PMMA were among the materials to have the least background scattering, which resulted in high-resolution structures of proteins using amounts ranging from only 0.04–0.1 mg.

Guha et al.[87] developed a combined COC and PDMS-based microfluidic system as a fixed-target in 2012, which could screen up to 100 crystallization conditions using ~1.4 μL crystal sample. The device was comprised of two layers of COC with a total of 120 μm thickness and a single 20 μm PDMS layer in between. In 2014, a similar device was developed by the same group[116]. This 200 μm thick device containing two COC layers and two PDMS layers demonstrated Lipidic Cubic Phase (LCP) crystallization of membrane proteins and on-chip diffraction data collection at room temperature, and allowed the structure determination of the photosynthetic reaction center at 2.5 Å resolution. Lyubimov et al.[117] developed a PDMS and PMMA-based microfluidic device targeted to capture crystals in predefined positions in a hydrodynamic trap array in 2015. The authors explored different trap and bypass designs to optimize for crystal capture. This device consumed ~5 μL of crystal sample to obtain a lysozyme structure at 2.5 Å at synchrotron and XFEL sources.

Furthermore, Gilbile et al.[118] and Liu et al.[119] developed COC and PMMA-based fixed-target sample delivery devices with a focus on minimizing background scattering. In these seven-layer devices, two layers of PMMA and two layers of COC were used as supporting layers, whereas two layers of 2–5 μm-thick COC layers were used as X-ray windows. Similarly, Sui et al.[120] explored the use of thin-layer graphene inside COC and PMMA-based microfluidic devices, where the use of a graphene layer was introduced as a diffusion barrier to reduce evaporation. In continuation of this work, thin-layer graphene electrodes were introduced to perform electro-crystallization on-chip, which showed faster nucleation as well as a high signal-to-noise ratio in the diffraction data[121]. Shelby et al.[122] in 2020 used graphene layers to encapsulate PMMA film layers containing crystals to maintain hydration conditions under vacuum (see Fig. 2c). The polymer/graphene sandwich was placed on a patterned Si substrate with ~100,000 X-ray accessible windows and required ~20 μL of crystal sample to fill the device windows.

A multilayer PMMA and Mylar fixed-target device was introduced by Zhao et al.[123] in 2020 and modified further in 2023[124]. This multilayer fixed-target device was unique for its delivery of the crystals to the synchrotron source through rotational motion. The device configurations for this approach are shown in Fig. 2d. In 2023, Carrillo et al.[125] developed a fixed-target design with precise arrays of cavities similar to the work by Manna et al.[72] and fiducial markers for loading the device and capturing the crystals in the predefined positions. They created different sizes of the cavities to successfully capture crystals of 5 μm, 10 μm, and 25 μm. Moreover, to perform an on-chip optical pump-probe experiment, the authors manufactured a black version of the same devices to minimize light contamination to the neighboring cavities[126].

Several multiple polymer-based systems were integrated with PDMS. Although Guha et al.[87] and Khvostichenko et al.[116] have used a very thin layer of PDMS in their systems to reduce background scattering from this material, there is still a need to integrate COC and PDMS, which requires complicated chemical modifications of the polymers for adequate bonding strategies. Lyubimov et al.[117] have developed a PDMS device to capture crystals in specific positions, but their system suffered from high background contribution from the siloxane. Table 2 shows that the system developed by Gilbile et al.[118] and Liu et al.[119] consumes a higher volume of sample compared to most of the systems in this section, but their approach offers very low background contribution from the material itself. Moreover, the fabrication strategy developed by these authors is relatively simple despite the multilayer approach. Integration of graphene into the fixed-target systems was introduced to prevent liquid evaporation from devices and potentially make them compatible in a vacuum experiment, which remains a cumbersome task. Among the graphene-integrated systems, Sui et al. improved their system to consume only 3.2 μL protein solution compared to their earlier development requiring 16 μL of samples (see Table 1).

Comparing the various fixed-target device realizations listed in Table 1 shows that almost any type of material employed for fixed-target devices could be realized, requiring about 1 μL of sample and consequently less than 1 mg of protein, sometimes down to tens of μg. Although this consumption is still about two orders of magnitude higher than that of the ideal scenario, it is crucial to recognize the limitations associated with the assumptions underlying this situation. These include, but are not limited to, the size distribution of the protein crystals in experimental situations, the arrangement of the crystals on a fixed-target device, specifically when placing them on predefined positions or arrays, and the ability to collect diffraction data from each crystal on the device, depending on the instrument used for data collection. We also point out that larger variations in the consumed sample volume may arise in Table 1, which are not necessarily only related to the volume capacity of the fixed-target device, but also to manual sample loading procedures requiring pipetting several μL of "dead volume" into the device (as reported previously[101,115], and as per the author's own experience)[72]. Such volume may be applied to the device, but removed prior to the measurement, which then leads to larger sample waste. However, in conclusion, the protein amounts required to obtain a dataset in SX are moderate, making fixed-target devices an excellent choice for experimenters when other experimental conditions allow for their use.

## Liquid delivery methods

Liquid delivery systems have been among the first ones realized to deliver protein crystals into the path of an X-ray beam for SX[127,128]. There have been numerous successful structure determination experiments arising from this injection method, and it is compatible with both pump-probe and mix-and-inject time-resolved crystallography. From the early realizations in low viscosity liquids and high sample consumption rates, liquid injectors have been further developed to consume less sample and have been realized in various materials and through various fabrication methods to suit the needs of the ever-changing experimental design and samples. This section

outlines the major developments in liquid injection systems, including gas dynamic virtual nozzles (GDVNs), droplet generators, electrospinning, and viscous injection techniques, each offering unique advantages for SX experiments. Liquid injection methods remain a main sample injection approach, specifically in XFEL SX, with about 70% of all protein structures deposited originating from XFEL experiments being liquid-based[129]. The progress toward minimizing sample consumption in the variations in sample consumption of the presented liquid injection techniques is discussed, paving the way for more efficient data collection in SX in future applications. Tables 3 and 4 highlight the sample delivery conditions and expand on additional critical parameters, such as protein concentration, crystal size, and sample consumption, across the manuscripts referenced in this section. These parameters illustrate the interplay between sample properties, delivery systems, and experimental conditions, highlighting the inherent challenges in accurately assessing and achieving effective sample consumption.

## Gas dynamic virtual nozzle

The first method used to deliver protein crystals into the path of an XFEL was through a microfluidic device called a GDVN[26]. A key advantage of this continuously flowing sample system, Fig. 3, was its ability to deliver crystals rapidly into the interaction region, enabling SX through the fast replenishment of samples into the x-ray interaction region. To assess sample consumption rates of current liquid delivery realizations, we first discuss an ideal sample injection scenario. We consider cubic protein crystals suspended in mother liquor with an average of 4 μm edge length delivered in a 5 μm thick jet produced by a GDVN and jetting with a velocity of 35 m/s, which is considered to be fast enough to accommodate MHz repetition rates[130]. Further assuming that the crystals were to be injected evenly spaced in the jet stream and in such a manner that each crystal is probed by an X-ray pulse (of similar diameter as the crystal edge length), an ideal sample consumption to obtain a dataset can be calculated. For example, at the various XFEL facilities, the repetition rate varies from 30 to 120 Hz, and assuming that 10,000 diffraction patterns would be needed to obtain a high-resolution structure, the jet operation with the above-mentioned thickness and velocity would require ~60–230 μL of sample with a crystal particle density of $4 \times 10^{10}$ to $1.7 \times 10^{11}$ crystals/mL, for 120 and 30 Hz respectively, representing the highest and lowest amount of sample required. Similar estimates can be carried out for the EuXFEL, which has a unique pulse structure, where pulse trains (currently delivering ~350 pulses)[131,132] at a MHz repetition rate are separated by 100 ms. A smaller sample volume with a lower crystal density would be required for the same full dataset since more pulses are delivered in a faster time: 2.0 μL of sample with a density of $5 \times 10^6$ crystals/mL. In these ideal scenarios, the amount of protein required to obtain a structure would then amount to 450 ng, as outlined in the section "Fixed-target sample delivery methods".

While utilizing protein crystals optimally is crucial, achieving the ideal case where every crystal injected is hit by an x-ray pulse is practically impossible as this synchronization is extremely challenging; in a typical GDVN, there is no control for the crystal distribution within the jet, such that optimized hit rates are typically achieved by high crystal densities in the solution ($>10^7$ crystals/mL), which leads to large amounts of "wasted" sample which in most cases cannot be recovered. Furthermore, variations in jet flow or sample clogging can result in some or most pulses not hitting the crystals, additionally wasting the sample. These factors, combined with the dead volume contained in most sample reservoirs as well as sample consumption during the optimization of the X-ray beam while injecting, make it difficult to achieve the ideal sample consumption. Not surprisingly, as outlined below, all liquid injection techniques reported protein crystal consumption orders of magnitude higher than the ideal case outlined above.

Despite these challenges, the GDVN continues to be one of the most popular methods of liquid delivery[26,127,128,133–135]. Published in 2008[127], the early iterations of the GDVN consisted of a nesting glass-in-glass design. Constructing these nozzles involved grinding a Kapton-coated glass capillary to create a beveled end, followed by precise insertion into a hand-formed outer glass casing shaped by rotating a borosilicate glass tube over a propane torch. Alignment accuracy was achieved by adjusting the spacing between the inner capillary and outer channel through the meticulous placement of a PTFE sleeve by hand, occasionally hand-shaved for a better fit. Operation during SX experiments typically required liquid flow rates around or above 20 μL/min at crystal concentrations of $10^{10}$ crystals/mL. Under these conditions, operating at 120 Hz at the LCLS, for example, the majority of the protein crystals are not exposed to the X-rays, resulting in considerable waste. In this scenario, to generate 10,000 diffraction patterns, several milliliters of the sample may be consumed, far exceeding the theoretical minimum of ~60 μL.

However, GDVNs remain preferred injectors for SX experiments to date, specifically for experiments in vacuum and time-resolved applications. Several improved fabrication strategies have been suggested to reduce the manual labor involved in their fabrication and to improve or add functionality. Beyerlein et al.[136] suggested an alternative method, employing micro-injection molding of ceramic as an outer casing, simplifying the assembly of a GDVN, and improving chemical compatibility. Other design variations of the GDVN consisted of low-pressure versions, as proposed by Doak et al.[133] in 2012, then further developed by Trebbin et al.[137] in 2014. In this chip-based iteration of the GDVN, the channels for liquid and gas flow are fabricated through photolithography from PDMS and require flow rates similar to glass GDVNs, producing jets a few micrometers in diameter. Due to the complexity and required geometrical detail, it was soon realized that high-resolution 3D printing could provide a suitable alternative to manual or chip-based assembly procedures[138,139]. The early realizations of 3D-printed nozzles were made by 2-photon polymerization (2PP) using a suitable polymer as printing resin. 2PP constitutes a high-resolution printing technique capable of generating micro-structures down to a few hundred nm resolution. The 3D-printed nozzles were glued onto a glass capillary with good reproducibility and adhesion strength to facilitate liquid delivery. Flow rates for the 3D-printed GDVNs were similar to the glass GDVNs and ranged from 5 to 30 μL/min for liquids[139]. Though the assembly and reproducibility of nozzle fabrication were optimized with the introduction of polymer-based GDVNs, the flow rates required by these injectors were still higher than desired for efficient sample delivery.

Further development has pushed GDVNs into the regime of making sub-micrometer jets (on the order of a few hundred nanometers) by altering the inner geometry of the nozzle. Nazari et al.[140] and Karpos et al.[141] developed asymmetric 2PP 3D-printed GDVN nozzles capable of sustaining liquid jets as low as 1 μL/min with diameters as small as 325 nm. While these nozzles are primarily useful in small-angle X-ray scattering/wide-angle X-ray scattering (SAXS/WAXS) experiments conducted on single particles or protein solutions (as opposed to protein crystal slurries), their potential applicability extends to nano-focus X-ray beams available at certain XFEL facilities. This suggests potential utilization for probing sub-micron protein crystals in thinner jets while consuming an order of magnitude less protein sample based on flow rates possible with these designs compared to previous GDVN realizations.

In addition to the improved fabrication methods, GDVNs have evolved to address a multitude of sample delivery goals, including mixing experiments, easing the delivery of more viscous samples, and further optimizing sample conservation. Among the initial modifications to GDVNs was the incorporation of mixing elements. The use of a GDVN also allowed for the activation of light-driven protein reactions, enabling the investigation of "pump-probe" SX experiments[78].

**Table 3 | Overview of GDVN, electrospinning, and high viscosity liquid injection systems in the literature, including details on X-ray source, flow rates, analyte, protein, and particle concentration, crystal size, and reported sample consumption**

| Authors | TR (PP, MISC, N) | X-ray source [freq.] | End-station [environment] | Flow rate (µL/min) | Analyte | Protein conc. (mg/mL) | Particle conc. (part/mL) | Crystal size | Sample consumption (volume [µL]/amount [mg]) | Comment |
|---|---|---|---|---|---|---|---|---|---|---|
| **Glass and ceramic GDVN** | | | | | | | | | | |
| Chapman et al.[26] | N | LCLS (30) | CXI (vacuum) | $Q_{exp}$:10 | PSI | 1 | $10^9$ | <2 | NI | Glass GDVN |
| Beyerlein et al.[136] | N | LCLS (120‡) | CXI (vacuum) | $Q_j$:12–40 $Q_{exp}$:40–60 | HEWL | 100^a | $5 \times 10^{7a}$ | $3 \times 6$ µm^a | NI | Ceramic GDVN |
| Calvey et al.[144] | MISC | N/A | N/A | $Q_j$:10–250 | PS beads, rhodamine | N/A | N/A | N/A | NA | Glass GDVN |
| Gisriel et al.[209] | N | EuXFEL (10 Hz, 30 pulses/train) | SPB/SFX (vacuum) | $Q_{exp}$: 20 | PSI | 100 | $1.7 \times 10^8$ | $5 \times 5 \times 15$ µm | NI | Glass GDVN |
| Grunbein et al.[210] | N | EuXFEL (10 Hz, 60 pulses/train) | SPB/SFX (vacuum‡) | $Q_j$:30–40 | (1) Urease (2) Concanavalin A (3) Concanavalin B | NI | NI | <20 µm | NI | Glass GDVN |
| Pandey et al.[211] | MISC | EuXFEL (10 Hz, 202 pulses/train) | SPB/SFX (vacuum‡) | $Q_{exp}$: 3.3–11.6 | β-lactamase mixed with ceftriaxone | NI | NI | $10 \times 10 \times 2$ µm | NI | Glass GDVN |
| **Polymer GDVN** | | | | | | | | | | |
| Trebbin et al.[137] | MISC | N/A | N/A | $Q_j$: 2.25–150 | (1) water, (2) fluorescein (3) rhodamine B | N/A | N/A | N/A | N/A | PDMS-Chip |
| Nelson et al.[139] | N | N/A | N/A | >5 µL/min | (1) water (2) cytochrome c oxidase | N/A | N/A | N/A | N/A | 3D-printed |
| Brehm et al.[212] | MISC | LCLS (120‡) | MFX (He‡) | $Q_j$: 2.5–15 | KDO8PS | NI | NI | 10 µm | NI | 3D-printed mixer |
| Vakili et al.[213] | N | N/A | N/A | $Q_j$: 8–167 | water | N/A | N/A | N/A | N/A | Kapton Foil-Chip |
| Knoska et al.[138] | MISC | LCLS (120) | MFX (He) | GDVN: $Q_j$: 2.4–20 $Q_{exp}$: 5 | Equine hemoglobin | 8–10 | NA | 1 µm | NA | 3D-printed and DFFN |
| Nazari et al.[140] | N | N/A | N/A | $Q_j$: 1–20 | water | N/A | N/A | N/A | N/A | 3D-printed |
| **Modified GDVNs** | | | | | | | | | | |
| Wang et al.[143] | MISC | N/A | N/A | 0.05 | (1) water (2) sulforhodamine 101 | N/A | N/A | N/A | N/A | Glass mixing |
| Stagno et al.[61] | MISC | LCLS (120) | CXI (vacuum‡) | $Q_{exp}$: 300 nL (LCP) 15–29 (GDVN) | adenine riboswitch | 40 | NI | 1–10 µm | NI | Glass GDVN and LCP injector with Agarose media |
| Oberthur et al.[214] | MISC | LCLS (120) | CXI (vacuum‡) | $Q_j$: 3–10, $Q_{exp}$: (1) 5 (2) 3–5 (3) 8.5 (4) 7 | (1) RNA polymerase II (2) granulovirus (3) NiFe-hydrogenase (4) dioxygenase | (1) 24 (3) 10 (4) 5–7 | 1–3: $10^{11}$ 4: $10^8$–$10^9$ | (1) 0.8 µm (2) 0.4×0.2×0.2 µm (3) 5–10 µm (4) 1–2×1–2×10–30 µm | (1) 450 µL*/ 10.8 mg (3) 765 µL*/ 7.65 mg (4) 344 µL*/2.41 mg | Glass DFFN |

**Table 3 (continued) | Overview of GDVN, electrospinning, and high viscosity liquid injection systems in the literature, including details on X-ray source, flow rates, analyte, protein, and particle concentration, crystal size, and reported sample consumption**

| Authors | TR (PP, MISC, N) | X-ray source [freq.] | End-station [environment] | Flow rate (µL/min) | Analyte | Protein conc. (mg/mL) | Particle conc. (part/mL) | Crystal size | Sample consumption (volume [µL]/ amount [mg]) | Comment |
|---|---|---|---|---|---|---|---|---|---|---|
| Doppler et al.[146] | PP | EuXFEL (10 Hz, 125 pulses/ train) | SPB/SFX (vacuum) | $Q_j$: 1–10 $Q_{exp}$: 5 | PSII | NI | NI | 10–30 µm | NI | 3D-printed Co-flow |
| Jernigan et al.[149] | N | LCLS (120) | MFX (GDVN-He, coMESH-air) | $Q_{exp}$: 3–5 (coMESH) 25 (GDVN) | NendoU | 40–80 | NI | 2 × 2 × 2 µm, 10–15 µm | NI | CoMESH and GDVN-misc |
| **Electrospinning injectors** | | | | | | | | | | |
| Kern et al.[215] | PP | LCLS (120) | CXI (vacuum) | $Q_{exp}$:2.5–3.1 | PSII | 7.4 | NI | 5–10 µm | NI | MESH |
| Sierra et al.[148] | N | LCLS (120) | CXI (vacuum) | $Q_j$: 0.14–3.1 $Q_{exp}$: 0.17 | Thermolysin | NI | $2 × 10^{10}$ | 2.5 × 4–3.5 × 7 µm | NI/0.14 mg | MESH |
| Sierra et al.[153] | MISC | LCLS (120) | CXI (vacuum) | $Q_{exp}$: 0.75–3 | (1) PSII (2) paromomycin complex (3) Thermolysin | (1) 70, (3) 25 | 2: $10^{10}$–$10^{11}$ | (1) 5–15 µm (2) 3–10 × 3–10 × 20–30 µm (3) 3 × 5 µm | (2) 360 µL/NI | CoMESH |
| Dao et al.[154] | MISC | LCLS (120) | CXI (vacuum) | NI | Thermus thermophilus | NI | $10^{10}$–$10^{11}$ | 2 × 2 × 4 µm | 500 µL/NI | CoMESH |
| Ciftci et al.[155] | MISC | LCLS (120) | CXI (vacuum) | $Q_{exp}$:1–3 | HIV-1 | NI | $10^{10}$–$10^{11}$ | 1 × 1 × 5 µm to 5 × 5 × 15 µm | 500 µL/NI | CoMESH |
| Ishigami et al.[152] | N | LCLS (30) | MFX (He) | $Q_{exp}$:3 | Cytochrome c oxidase | 80–90 | $10^{10b}$ | 20 × 20 × 4 µm | NI | MESH |
| **Viscous injection** | | | | | | | | | | |
| Weierstall et al.[157] | N | LCLS (60–120) | CXI (vacuum) | $Q_j$: 0.001–0.3 $Q_{exp}$:0.17 | (1) β2 adrenergic receptor (2) A$_{2A}$AR (3) glucagon receptor (4) 5-HT$_{2B}$ (5) SMO (6) DgkA | (1–3) 20–50 (6)12 | NI | (5) >5 µm | <100 µL/ <0.5 mg | LCP Injector with LCP |
| Liu et al.[158] | N | LCLS (120) | CXI (vacuum) | $Q_j$: 0.003–0.3 | 5-HT$_{2B}$ | NI | NI | 5 × 5 × 5 µm | NI/0.3 mg | LCP Injector with LCP |
| Sugahara et al.[162] | N | SACLA (30) | BL3-EH4 (He) | $Q_j$: 0.12–0.48 $Q_{exp}$: (1, 2, 4) 0.48 (3) 0.46 | (1) HEWL (2) glucose isomerase (3) thaumatin (4) fatty acid-binding protein type 3 | NI | (1) $6 × 10^7$ (2) $2 × 10^7$ (3) $1 × 10^7$ (4) $0.9 × 10^7$ | (1) 7–10 µm (2) 10–30 µm (3) 10–30 µm (4) 10–20 µm | (1) NI/2.4 mg, (2) NI/0.7 mg (3) NI/0.28 mg (4) NI/0.15 mg | Needle Extrusion with mineral oil-grease media |
| Botha et al.[168] | N | SLS (10) | PXII (He‡) | $Q_{exp}$: 0.021 | HEWL | 30–60 | NI | 10 × 10 × 30 µm 15 × 15 × 60 µm | NI | HVE with LCP and Vaso-line media |
| Conrad et al.[165] | N | LCLS (120‡) | CXI (vacuum, He) | $Q_{exp}$: 1.6 | (1) Phycocyanin (2) PS I (3) PS II | (1) 15 | (1–3) $2 × 10^{10}$ | (1) 1–5 µm | (1) 0.02 µL*/0.3 mg | LCP Injector with Agarose media |
| Sugahara et al.[55] | N | SACLA (30) | BL3-EH4 (He) | $Q_{exp}$: 0.48 | (1) Proteinase K (2) HEWL | 40 | (1, 2) $6.7 × 10^7$ | (1, 2) 5–10 µm | 30 µL/1.2 mg* | Needle extruder with Super Lube grease and hyaluronic acid media |

**Table 3 (continued) | Overview of GDVN, electrospinning, and high viscosity liquid injection systems in the literature, including details on X-ray source, flow rates, analyte, protein, and particle concentration, crystal size, and reported sample consumption**

| Authors | TR (PP, MISC, N) | X-ray source [freq.] | End-station [environment] | Flow rate (µL/min) | Analyte | Protein conc. (mg/mL) | Particle conc. (part/mL) | Crystal size | Sample consumption (volume [µL]/amount [mg]) | Comment |
|---|---|---|---|---|---|---|---|---|---|---|
| Kovacsova et al.[166] | N | SLS (10) | PXII (He‡) | $Q_i$: (1,2) 0.06–0.15 (3) 0.3–0.15 (4) 0.09–0.15 | (1) Lyophilized thermolysin (2) Glucose isomerase (3) HEWL (4) Bacteriorhodopsin | (1) 25 (2) 80 (3) 30–60 (4) 35–50 | NI | (1) 50–130×5–10×5–10 µm (2) 10–15×10–15×10–15 µm (3) 30×20×20 µm (4) 20×50×2 µm | (1) 20 µL*/0.5 mg (2) 6 µL*/0.5 mg (3) 8 µL*/0.5 mg (4) 10 µL*/0.5 mg | HVE with sodium carboxymethyl cellulose and Pluronic media |
| Sugahara et al.[170] | N | SACLA (30) | BL3-EH4 (He) | $Q_{exp}$: (1) 0.42–0.75 (2) 0.47 (3) 0.38–0.47 | (1) HEWL (2) thaumatin (3) proteinase K | (1) 20 (2, 3) 40 | (1) $1.7\times10^7$–$5.8\times10^8$ (2) $4.3\times10^8$ (3) $4.9$–$9.3\times10^7$ | (1) 1×1×1 µm, 5×5×5 µm, 20×20×30 µm (2) 2×2×4 µm (3) 4×4–5×5×7 µm, 8×8–12×12×12 µm | (1) 105 µL*/2.1 mg (2) 12.5 µL*/0.5 mg (3) 35.8 µL*/1.4 mg | HVC injector with nuclear grease |
| Shimazu et al.[171] | N | SACLA (30) | BL3-EH4 (He) | $Q_i$: 0.1–5.6 $Q_{exp}$: (1) 0.24 (2) 0.42 | (1) $A_{2A}AR$ (2) HEWL | (1) 50 (2) 20 | (2) $2.3\times10^8$ | (1) 20×3×3 µm (2) ~5 µm | (1) 30 µL/1.5 mg* (2) 40 µL/0.8 mg* | HVC injector with LCP and nuclear-grade grease |
| Sugahara et al.[163] | N | SACLA (30) | BL2 (He) | $Q_{exp}$: 0.24, 0.11 | proteinase K | | (i, v) $9.4\times10^7$ (ii, iii, iv) $1.4\times10^8$ (vi, vii) $7.2\times10^8$ | (i–v) 2×2 µm (vi–vii) 0.8×0.8 µm | (i–v) 20 µL/0.8 mg* (vi–vii) 8 µL/0.32 mg* | HVC injector with (i) Paraffin grease (ii) DATPE grease (iii) Nuclear grease (iv) Cellulose (v) Frozen Paraffin grease (vi) Paraffin grease (native) (vii) DATPE grease (pr-derivative) |
| Vakili et al.[216] | MISC | EuXFEL (10 Hz, 1 pulse/train) | SPB/SFX (vacuum) | $Q_i$: 0.11–0.36 | iq-mEmerald protein mixed with $CuCl_2$ | 50 | NI | 5×15 µm | NI | 3D-printed injector with LCP media |
| Wolff et al.[167] | PP | SACLA (30) | BL2 (He‡) | $Q_{exp}$: 2.5 | HEWL | 20 | NI | NI | NI | HVC with hydroxyethyl cellulose media |

Sample consumption is recorded in volume and protein amount in milligrams. Time-resolved experiments in the second column are classified into pump-probe (PP) for light-induced, mix and inject (MISC), and N for not time-resolved. †The conditions in which the experiment was conducted are not explicitly stated, and the recorded condition is assumed based on the X-ray instrument and end-station used. $Q_{exp}$: flow rates listed were used for data collection. $Q_i$: flow rates recorded were given in a range. *The sample consumption value was estimated from the reported protein concentration in the mother liquor, considering that all protein molecules were converted to crystals, considering the reported volumes consumed, or the amount of protein consumed. NI refers to data not included in the publication. N/A refers to data not applicable to the experimental type reported in the publication. The values listed in the table (symbol a) are derived from a citation within the manuscript, which provided the crystallization conditions[218]. The values listed in the table (symbol b) are derived from a citation within the manuscript providing the crystallization conditions[217].

**Table 4 | Overview of droplet injection systems in the literature, including details on X-ray source, flow rates, analyte, protein, and particle concentration, crystal size, and reported sample consumption**

| Authors | TR* (PP, MISC, N) | X-ray source (freq.) | End-station (environment) | Flow rate (µL/min) | Analyte(s) | Protein conc. (mg/mL) | Particle conc. (part/mL) | Crystal size | Sample consumption (volume [µL]/ amount [mg]) | Injector comment/ droplet Size |
|---|---|---|---|---|---|---|---|---|---|---|
| Echelmeier et al.[172] | N | LCLS (120#) | CXI (vacuum) | Qexp: 5.5 | (1)Fluorescein droplets (2) granulovirus | NI | NI | NI | NI | glass GDVN + PDMS segmented drop/NI |
| Mafune et al.[147] | N | SACLA (30) | BL3-EH4 (He) | Qexp: 0.5 | HEWL | NI | $3.2\times10^7$ to $3.2\times10^8$ | 5 | NI/0.3 mg | Piezo-driven/ 0.268 nL |
| Roessler et al.[177] | N | LCLS (60) | XPP (air) | N/A | (1) HEWL, (2) thermolysin, (3) stachydrine demethylase, (4) MauG mixed with MADH (5) hemoglobin (6) PSII | (1) 70 (2) 30 (5) 18 (6) 20–40 | NI | (1) 5–10 µm (2)10–100 µm, (3) 25–50 µm, 50–100 µm, 200–300 µm (4) 10–50 µm (5) 50–250 µm (6) 50–100 µm, 150–400 µm | NI | ADE and levitation/ 0.1–2.5 nL |
| Awel et al.[181] | N | LCLS (120#) | CXI (vacuum) | Qexp: 2.7–3.5 | granulovirus | N/A | $3\times10^{11}$ | $0.2\times0.2\times0.37$ µm | NI | Ceramic aerosol/0.03 pL |
| Echelmeier et al.[173] | N | LCLS (120) | MFX (He) | Qr: 0.5–20 | PSI | 1[a] | $10^{9a}$ | 0.2–1 µm | NI | Segmented Drop/ <1 nL |
| Echelmeier et al.[174] | N | EuXFEL (10 Hz, 15 pulses/ train) | SPB/SFX (vacuum) | Qr: 3–20 | KDO8PS | 21 | $5\times10^9$ | 8–10 µm | 962 µL/20 mg* | Segmented Drop/ 70–800 pL |
| Sonker et al.[70] | N | LCLS (120) | MFX (He) | Qexp:4.1–5.1 | (1) KDO8PS (2) HEWL | (1) 20 (2) 126 | (1) $2\times10^4$ | (2) 5–10 µm | NI | Segmented Drop/NI |
| Doppler et al.[176] | N | LCLS (120) | MFX (He) | Qexp: 3–4 | (1) NQO1 (2) Phycocyanin | (1) 25 (2) 50 | NI | (1) $10\times2\times2$ µm (2) 5–15 µm | NI | Segmented Drop/2.3 nL |
| Perrett et al.[68] | N | EuXFEL (10 Hz, 16 pulses/train) | FXE (He) | NI | HEWL | NI | $3–6\times10^7$ | NI | NI | DoD/NI |
| Doppler et al.[80] | MISC | EuXFEL (10 Hz, 300 pulses/ train) | SPB/SFX (vacuum) | Qexp: 0.5–4.9 | NQO1+NADH | 18–26.5 | $2\times10^7$ | $5\times40$ µm | 228 µL/4.9 mg | Segmented Drop/2.5 nL |

Sample consumption is recorded in volume and protein amount in milligrams. Time-resolved experiments in the second column are classified into pump-probe (PP) for light-induced, mix and inject (MISC), and N for not time-resolved. #The conditions in which the experiment was conducted are not explicitly stated, and the recorded condition is assumed based on the X-ray instrument and end-station used. Qexp: The flow rates listed were used for data collection. Qr: The flow rates recorded were given in a range. *The sample consumption value was estimated from the reported protein concentration in the mother liquor, considering that all protein molecules were converted to crystals, considering the reported volumes consumed, or the amount of protein consumed. NI refers to data not included in the publication. N/A refers to data not applicable to the experimental type reported in the manuscript. The values listed in the table (symbol a) are derived from a citation within the manuscript providing the crystallization conditions[27].

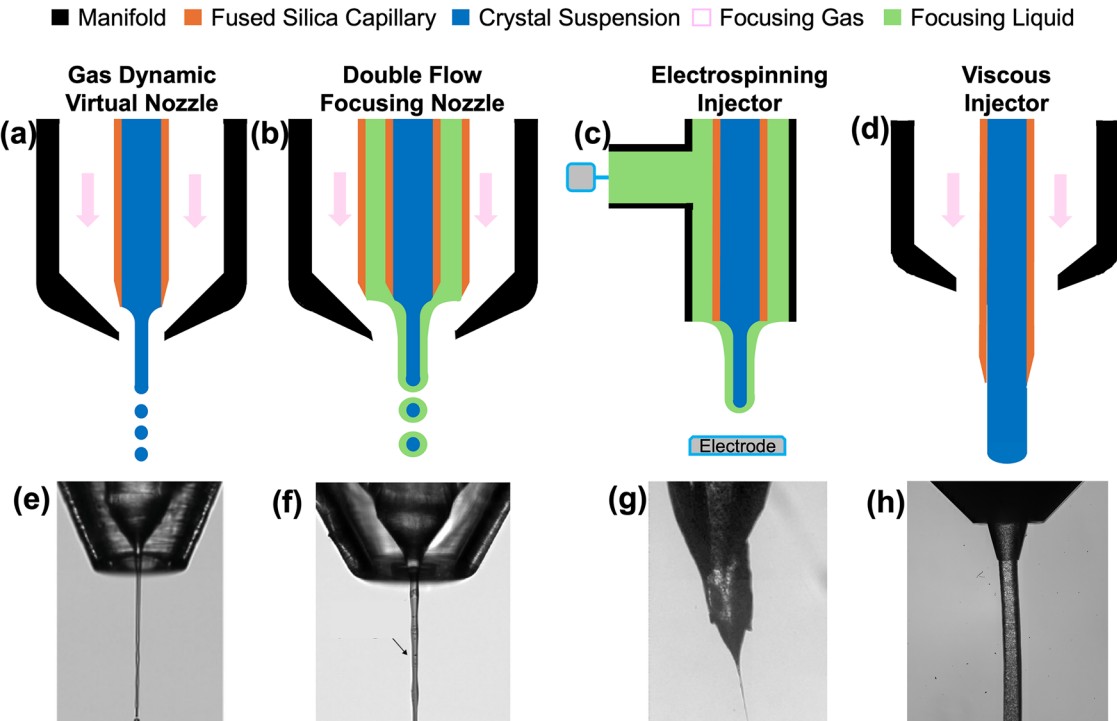

**Fig. 3 | Examples of liquid delivery methods.** Schematics adapted from Echelmeier et al.[4] of **a** a 3D-printed gas dynamic virtual nozzle. **b** a double-flow-focusing gas dynamic virtual nozzle, **c** an electrospinning jet, and **d** a high viscosity extrusion. **e**, **f** Images of nozzles showing sample injection for each case in (**a**–**g**) adapted and reprinted from Knoska et al.[138] and Sierra et al.[153], respectively. Image (**h**) provided by H. Hu (Arizona State University, personal communication).

However, numerous proteins are involved in non-light-driven reactions, which in turn require the inclusion of mixing elements with GDVNs[78,142], as introduced by Wang et al.[143]. While the inner channel delivered the crystal sample, an additional outer channel delivered the substrate for the reaction with the protein crystals. After the two solutions are combined into a single stream where mixing takes place, the sample is transported downstream to the GDVN, where the jet is generated for delivery to the XFEL beam. This early design faced challenges, such as operating at high substrate-to-crystal flow rates (2000:1), which decreases sample consumption, but also dilutes the protein crystal concentration, which is offset by a reduction in hit rates, consequently requiring longer injection times. The improved version of Calvey et al.[144] partially addresses this issue by achieving rapid mixing of protein and substrate within sub-millisecond to several hundred-millisecond timescales with substrate-to-crystal flow rate ratios around 40:1. This enhancement offers higher hit rates compared to standard glass GDVNs and more efficient sample use, though the trade-off between dilution and flow consistency remains a consideration.

To further the applicability of liquid mixers to time points up to several seconds, a simple T-junction was employed to combine two liquids before entering the liquid line of a glass GDVN[61]. However, the switch to high-resolution 3D printing provided enhanced design options, allowing researchers to customize GDVN configurations for specific experimental needs in TR-SX. This was realized in a 3D-printed hybrid mixer and nozzle[145], combining the mixing element with the injection nozzle in a single 3D-printed piece, enabling reaction time points between 30 and 140 ms with crystal sample flow rates between 2.5 and 15 μL/min and substrate flows ranging from 6.5 to 37.5 μL/min, representing an 8-fold improvement in the efficiency of sample delivery compared to typical GDVN flow rates.

While GDVNs enabled sample delivery for many protein crystals and were able to be used for TR-SX, the protein amounts required for these experiments remained limited for most proteins not accessible in high quantities. Therefore, researchers began optimizing sample consumption while still maintaining the high-quality data collection facilitated through GDVNs. The double-flow-focusing nozzle (DFFN), proposed by Wang et al., is one such realization that stems from the nested glass capillary design of a GDVN, aimed to reduce sample consumption, enhance jet quality, and facilitate rapid mixing experiments[143]. Then converted into a 3D-printed version by Knoska et al., the proximity of the two inner capillaries to the gas-focusing region allows the use of an organic solvent or a binding substrate in mother liquor as a sheathing liquid (see Fig. 3b, f)[138]. This ability reduces sample consumption by volume replacement from the sheath while ensuring the generation of stable jets[138]. Sample operation flow rates in the DFFNs range from 0.05 to 10 μL/min, with flows of 3–8.5 μL/min reported in SX experiments, similarly resulting in an 8-fold decrease in sample required at the lowest flow rate.

It is also important to mention that flow rate is not the only factor contributing to sample consumption in these experiments. Protein crystal concentration within the sample solution plays a critical role; higher crystal count per volume generally leads to increased hit rates and better data quality, and this parameter is often highly variable for SX experiments with different proteins. Thus, comparing flow rates and crystal consumption amounts must be carefully assessed when comparing sample consumption between experiments. In addition, the geometry and running parameters of the nozzle affect sample consumption. By discussing these parameters alongside flow rate, we can better contextualize the multifaceted factors that influence sample consumption and guide optimization strategies for future experiments.

Another approach to reduce sample consumption and concomitantly improve nozzle lifetime and jetting conditions was introduced by Doppler et al.[146], featuring a junction upstream of the GDVN, where the sample stream is brought in contact with an immiscible liquid oil termed co-flow injection. Uniquely, the addition of the oil improved the injection of more viscous samples by minimizing device

clogging and generating several hundred micrometer-long jets that allowed for probing the sample far from the nozzle. The latter improves the GDVN lifetime by minimizing the backsplash to the nozzle. Membrane proteins such as Photosystem II could be injected with the co-flow injector in an aqueous buffer containing 35% poly-ethylene glycol (PEG) 5000 with a viscosity of ~90 mPa. The co-flow injector flow rates range from 1 to 10 μL/min, with 5 μL/min utilized at the EuXFEL for SX with PSII.

Assessing sample conservation is challenging when comparing GDVN injection methods due to the limited availability of detailed data in many studies, as key parameters such as protein concentration and total sample volume consumed are not consistently reported (and are thus not listed in Table 3). Due to this gap in reporting, the flow rate remains one of the few available parameters for comparison, which can be misleading as sample consumption depends heavily on factors such as protein concentration and the time spent delivering the sample. Nonetheless, significant advancements have been made over the years, with the sample flow requirements of the original glass GDVN, which demanded upwards of 20 μL/min, being substantially reduced. Modern innovations such as the DFFN have brought flow rates down to 3 μL/min, while the Co-flow injector has further improved efficiency, achieving rates as low as 1 μL/min.

The evolution of GDVN-based sample delivery methods—from the original glass and ceramic nozzles to more advanced designs like polymer-based nozzles, DFFNs, mixing, and co-flow injectors—has led to significant strides in improving sample efficiency. The early glass and ceramic GDVNs, while groundbreaking, posed challenges in terms of high sample consumption, particularly due to the continuous flow nature and incomplete synchronization with the XFEL pulses. Subsequent developments, such as the microfluidic DFFNs and the introduction of mixing elements, have addressed some of these limitations by reducing flow rates and improving the precision of crystal delivery[4,64,147].

### Electrospinning sample delivery systems

Further decreasing the sample required for injection, Sierra et al. combined the principles of electrospinning and microjets to create a new sample injection method that decreased sample flow rates while eliminating the need for any complex nozzle fabrication. By imposing a potential difference between a diverging liquid stream from the end of a capillary and an electrode positioned 5–8 mm from the capillary exit, the stream will converge and form a liquid jet via electrospinning. The cone formed between the outlet of the liquid supply line and the electrode resembles that of a GDVN jet and can serve as a continuous injection method for SX, Fig. 3c, g[148–150]. The microfluidic electrokinetic sample holder (MESH) injector is a flexible injection system with a simple design that can deliver crystals from an Eppendorf tube, avoiding the need for high delivery pressures while being compatible with more complex pump-probe or X-ray emission spectroscopy experiments[151,152]. Flow rates of 0.17–5 μL/min reduce the amount of sample required for SX studies compared to GDVNs.

When the MESH injector was initially designed, it required the use of cryo-protectants, like glycerol and PEG, to prevent freezing after the protein-liquid solution exited the outlet of the liquid supply line during delivery in a vacuum. These additional chemicals have the benefit of keeping the crystals suspended in the sample vial. However, not all proteins are compatible with these dehydrating chemicals. To avoid adding cryo-protectants to the buffer of the delivered protein crystals, the concentric MESH (coMESH) injector was developed[153]. The coMESH injector works similarly to the DFFN, where the crystal solution is surrounded by a sheathing liquid. This sheathing liquid can contain the cryo-protectant, preventing the delivered solution from freezing in vacuum, and because it is introduced around the unaltered sample just before it leaves the opening of the injector, there is minimal interaction[154,155]. This configuration can also be used to perform mixing experiments where the substrate is delivered in the sheathing solution, and where both solutions may contain a cryo-protectant[63]. Although the sheathing solution could compensate for sample flows in theory, in practice, the sample flow rates in the coMESH injector are likewise around 0.7–3 μL/min, still a large improvement over GDVN injection methods. Moreover, it is important for this delivery method that the electrical potential applied to the sample may influence the reconstructed protein structure.

To assess sample consumption rates of MESH injectors, we again consider cubic protein crystals suspended in mother liquor with an average edge length of 4 μm, delivered in a cone with a radius and height of 50 μm. Assuming that the cone is replaced after every X-ray pulse, the ideal sample consumption to obtain a dataset can be calculated by multiplying the volume of the cone by 10,000 X-ray pulses required to generate the full dataset. With these assumptions, the ideal sample consumption would be 0.33 μL, with a crystal concentration of $3 \times 10^7$ crystals/mL. The idealized sample consumption estimate assumes a perfectly uniform (evenly spaced) delivery of crystals within the jet, which is challenging to achieve in practice. Furthermore, handling such a small volume of sample and ensuring its consistent delivery throughout the experiment is difficult, as even slight variations in the injector performance, non-ideal behavior of crystal slurries in the liquid lines, or sample handling can lead to inconsistencies. Maintaining a stable crystal concentration and avoiding loss during the injection process adds further complexity. As outlined in Table 3, sample consumption rates are often reported to be much higher, typically around 400-500 μL, due to these challenges[153–155]. This higher consumption accounts for inefficiencies in the delivery system, variations in crystal quality, and the need for additional samples to compensate for factors like evaporation and handling losses.

### High viscosity extruders (HVEs)

Protein crystallization conditions vary vastly among classes of proteins. Membrane proteins are particularly difficult to crystallize and, in most scenarios, need to be injected in high-viscosity buffers sustaining their membrane-like environment. LCP was developed to aid in the crystallization of membrane proteins in a lipidic environment[46,156,157], facilitating protein crystallography of this important class of proteins. Once grown in LCP, proteins cannot be removed from the lipid environment, and thus, injectors compatible with highly viscous media had to be developed.

The LCP injector, developed by Weierstall et al.[157], extrudes the viscous solution in a thin string that can be aligned with the path of an X-ray beam, Fig. 3d, h. The high viscosity demands high pressures of 2000 to 10,000 psi to extrude the viscous medium, requiring a sample reservoir that uses two pistons to magnify the force applied by the HPLC pump. The ejected LCP string tends to curl on itself. Therefore, a focusing gas, typically nitrogen or helium, is used to stabilize the extruding LCP. LCP injection requires significantly less sample as it typically operates at 0.01–1 μL/min extrusion rates, and it is compatible with pump-probe time-resolved experiments as the viscous media can be pumped a short distance after being extruded and then probed by the XFEL immediately after[157–161]. In addition to the LCP injector, other groups have simply used a syringe injector to achieve the same thin strand of crystal-containing viscous media[55,162–164]. Flow rates for the syringe needle extrusion are likewise around 0.1 to 1 μL/min. However, the extruded stream from either injection method has a large diameter (50–100 μm) compared to the liquid jets (0.5–10 μm), which increases the background scattering and can make structure determination challenging.

Since many proteins do not grow in LCP or simply do not require it for their growth, other viscous media have been explored. Agarose was injected using the LCP injector[165], hydrogels like sodium carboxymethyl cellulose[166], various greases (paraffin, nuclear, DATPE)[163], poly-acrylamide[164], and 18% hydroxyethyl cellulose[167] injected using a high

viscosity extruding sample injector modeled after the LCP injector[168]. These additional extrusion media operate at similar flow rates as LCP. However, they can exhibit lower background scattering, which is favorable for data analysis.

High-viscosity extrusion methods have been instrumental in addressing the challenge of sample conservation in experiments involving difficult-to-crystallize proteins or proteins that prefer a lipidic environment. By operating at flow rates as low as 0.01–1 μL/min, these methods significantly reduce the amount of sample required, typically below a milligram of protein, compared to more traditional liquid jet systems. Additionally, the exploration of alternative viscous media, such as hydrogels and greases, further contributes to efficient sample use by allowing more flexibility in matching the crystallization conditions to the protein delivery media.

However, a notable challenge of high-viscosity injection is that it is largely restricted to use at low repetition rate X-ray sources. The slower flow rates and larger extruded stream diameter are not well-suited to the MHz repetition, and the large diameter of the extruded stream can increase background scattering; this challenge is often offset by the reduction in sample waste. Despite these limitations, high-viscosity extrusion remains a critical method for optimizing sample consumption in XFEL-based crystallography, particularly for specific experimental setups where low-rep-rate sources are sufficient. It is worth noting, however, that viscous injection systems are effective delivery methods at synchrotron sources under certain conditions, particularly when using larger crystals, slower flow rates, and micro-focused beamlines, which enable multiple hits on the same crystal, reducing sample consumption[168,169].

To assess sample consumption rates of viscous injectors, we consider cubic protein crystals suspended in the viscous medium 20 μm apart, delivered in an extruded jet of 100 μm diameter. The ideal sample consumption to obtain a dataset can be calculated by multiplying the extruded volume for one shot by the 10,000 X-ray pulses required to generate a full dataset. With these assumptions, the ideal sample consumption would be ~1.5 μL, with a crystal concentration of $6.4 \times 10^6$ crystals/mL for the estimated 450 ng of protein. The idealized sample consumption estimate for viscous injectors assumes uniform crystal distribution within the viscous media and efficient jet extrusion. However, achieving consistent flow and uniform delivery is challenging in viscous systems. Variations in extrusion pressure, injector performance, and sample handling can lead to inconsistencies, making it difficult to maintain the idealized consumption rate. In practice, handling small volumes of viscous samples can result in losses during injection, such as clogging or uneven flow. As summarized in Table 3, sample consumption rates for viscous injectors are typically in the range of 10–100 μL, which is one to two orders of magnitude higher than ideally required. This is likely due to injector variability and sample handling losses leading to a need for additional sample to ensure reliable data collection under practical conditions[163,166,170,171].

## Droplet-based injectors

Droplet injectors are a promising approach for sample conservation, capable of delivering the sample to the X-ray interaction region only when the laser pulses. Early approaches for droplet injection with protein crystals focused on enhancing control over droplet formation and transport. In 2015, Echelmeier et al.[172] introduced a chip-based droplet injector generating protein crystal droplets segmented by an immiscible oil phase, transporting the two phases intermittently to the orifice of a GDVN for jetting. This innovative approach reduced the sample flow rate from 20 μL/min to 5.5 μL/min, delivering granulo virus to the LCLS XFEL while conserving about 70% of the sample compared to a jetting GDVN. The segmented droplet injection principle was further improved by generating a microfluidic droplet generator with 2PP 3D printing interfaced to a GDVN downstream, Fig. 4a, d[173]. This new injector was employed for SX at the EuXFEL in the early user-assisted commissioning experiments[174], where the team could demonstrate a reduction of sample consumption by up to 60% in comparison to continuous injection with a GDVN using similar flow rates. Droplets were generated at a frequency of 10 Hz, matching the MHz train repetition frequency, where the droplets were large enough to span the length of the train containing 32 X-ray pulses spaced ~900 ns (1.1 MHz). Moreover, the room temperature structure of the enzyme KDO8PS was obtained for the first time, revealing additional structural details compared to cryo-crystallography studies.

Subsequent improvements of the segmented droplet approach included strategies to synchronize the droplet arrival with the X-ray pulses. The principle of synchronization employed non-contact electrodes placed on either side of the droplet generation region to influence the release of the protein crystal-laden droplets into the segmented oil phase through electrowetting effects[175]. Using a feedback mechanism, the electrical triggering synchronization principle could further be improved to position the generated droplets in time relative to the XFEL to maximize synchronization. This principle was demonstrated at 120 Hz at the MFX instrument at LCLS[70,176]. Two injector types were developed for this purpose. The capillary coupled droplet injection device[129] and the modular droplet injector[70,176] both proved successful in delivering microcrystals of 4 different proteins, achieving up to 75% reduction in sample consumption. More recently, segmented droplet injection was applied at 10 Hz at the SPB/SFX instrument at the EuXFEL, where time-resolved mixing in droplets led to the elucidation of key structural insights into the reaction mechanism of NQO1 with its coenzyme NADH. Sample consumption savings reached up to 97% utilizing 2.3 nL droplets and total protein amounts of less than 4 mg to complete a full dataset[80].

Another droplet injection concept emerged in 2016 when Mafune et al.[147] proposed a different interface with a GDVN. Instead of using a carrier oil liquid and a gas-driven nozzle, they employed a piezoelectric element to issue an electric pulse, pulling a droplet out of the nozzle at the same repetition rate as the XFEL in use, Fig. 4b, e. With this injector, lysozyme crystals were successfully delivered into the path of the beam at the SACLA XFEL at 30 Hz, utilizing flow rates around 0.5 μL to inject 0.3 nL droplets, consuming only 0.3 mg of protein to complete a dataset. This was likewise applied at EuXFEL in 2024 by Perrett et al., where long sausages of protein crystal sample were ejected to cover 16 pulses of the MHz repetition rate train, ensuring efficient sample delivery while minimizing consumption[68]. Their approach demonstrated the ability to achieve high-quality diffraction patterns with minimal material usage, further highlighting the injector's effectiveness in reducing sample consumption in high-throughput X-ray crystallography.

Acoustic droplet ejection (ADE) is yet another technology of choice to deliver small-volume elements of crystal-laden liquid into the path of an XFEL. This technology uses short bursts of acoustic energy focused on the surface of a liquid reservoir to eject crystal-containing droplets. In 2016, Roessler et al.[177] developed this method by using acoustic injectors to create focused sound waves ejecting picoliter to nanoliter-sized droplets containing crystals, Fig. 4c, f. The authors demonstrated that droplets could be either injected from a liquid reservoir situated below the interaction region upwards into the path of an XFEL or, with an alternative assembly, downwards into the interaction region. The droplets created from these acoustic generators varied in size from 0.1 to 2.5 nL and delivered crystals in sizes ranging from 5 to 150 μm. This injector largely depends on droplet trajectory and thus can be challenging when larger crystals may alter the droplet path, or the additives contained in the crystallization buffers (e.g., PEG, surfactants) can hinder the formation of the droplet. Additionally, this injection system is not suited for in-vacuum experiments primarily due to sample freezing after injection.

A further development was made by Tsujino et al.[178], who expanded the suspended droplet approach to incorporate an acoustic

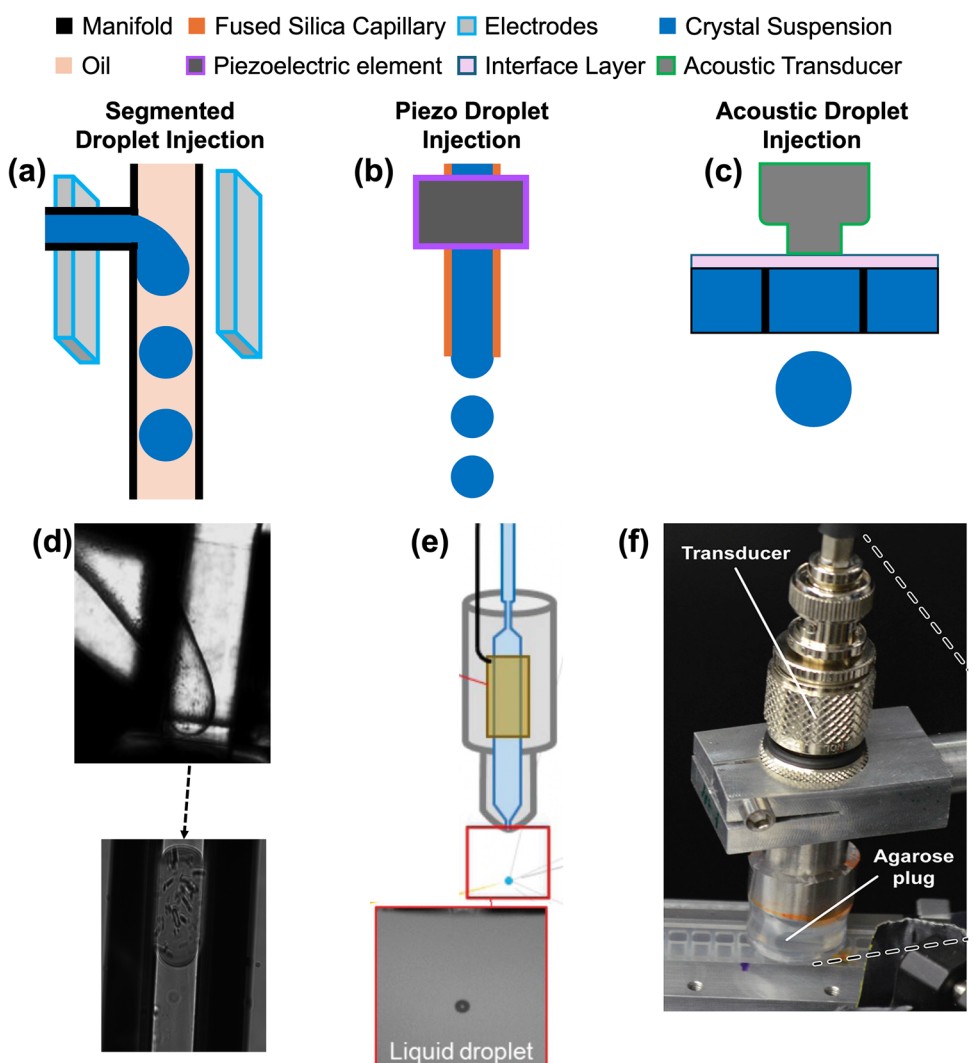

**Fig. 4 | Examples of droplet-generating delivery methods.** Schematic representations of **a** segmented droplet injection, **b** piezo droplet injection, and **c** acoustic droplet injection, adapted and reprinted from Echelmeier et al.[4]. Corresponding experimental images of **d** the segmented droplet generation in oil with a protein crystal (needle-shaped) containing droplet, **e** piezo-actuated droplet formation, and **f** an acoustic droplet ejection (ADE) injector. Adapted and reprinted from Sonker et al.[70], Doppler et al.[80], and reprinted with permission from Roessler et al.[177] and Mafune et al.[147].

levitator. This approach, known as droplet-on-demand (DoD), involved suspending the piezo-ejected droplet within a single-axis acoustic levitator. The droplets included $100–200\,\mu m$ lysozyme crystals into the X06SA beamline at the Swiss Light Source. Despite the large size of the delivered crystals, a single droplet of 31 nL could be irradiated several thousand times before requiring replacement. This approach represents a promising direction in balancing sample conservation with efficient data collection. By allowing multiple exposures within the same droplet, this method not only reduces the overall sample consumption dramatically but also maximizes the utility of each droplet.

In a similar scenario as outlined above to reach ideal sample consumption, a $4 \times 4 \times 4\,\mu m$ cubic crystal encased in a $5\,\mu m$ diameter liquid droplet (65 fL) and with 10,000 of these droplets each intersecting the X-ray pulses to produce a diffraction pattern would only require 650 pL of sample. This volume and number of crystals result in a crystal concentration of $1.5 \times 10^{10}$ crystals/mL. Interestingly, such concentration is reported for liquid injection (see Table 3) only once, where typically, the protein concentrations are several orders of magnitude lower. This is not surprising, since such high crystal concentrations with small enough crystals are experimentally challenging to reach, and crystal slurries of such density are difficult to handle. Additionally, loading tens of picoliters of sample is unrealistic, at least in a manual fashion, thus requiring automated sample loading procedures for future droplet injectors with optimized sample consumption.

It is further interesting to note that sample conservation would nominally be the same for each X-ray source, assuming droplet injection can be carried out in a synchronized manner matching the XFEL repetition rate. However, the time scale to reach a full dataset may vary. For example, in the ideal droplet injection case as outlined above, it would take 1.4 min to generate a full dataset at the LCLS XFEL at 120 Hz. At the EuXFEL, where ~350 pulses are delivered at 10 Hz repeating trains at MHz intra-train repetition rate[131,132], it would only take 3 s to complete a dataset. However, there are various factors influencing droplet injection that prevent this ideal scenario. Factors such as pressure fluctuations, nozzle clogging, or temperature variations can affect the consistency of the droplet formation and delivery. Additionally, the process of generating droplets at a consistent size and containing only a single crystal is inherently challenging, as variations in crystal size, morphology, and concentration can lead to droplets with multiple crystals or none at all. Moreover, achieving

perfect synchronization between the droplets and the X-ray pulses is technically demanding. Even minor deviations in timing or droplet trajectory can result in missed opportunities for data collection, reducing the overall efficiency of the experiment.

While the droplet generation methods discussed offer significant improvements in reducing sample consumption compared to continuous injectors, they still differ from the ideal scenario. For instance, the ADE, levitator, and segmented droplet injector generate droplets on the order of a few nanoliters, which are several orders of magnitude larger than the 65 pL droplets described above. The larger droplet volumes translate to increased sample requirements. These differences in droplet generation methods also introduce varying limitations. For example, droplet injection systems require synchronization with high-repetition-rate X-ray sources, dictating specialized technology. In contrast, ADE levitated droplets offer unique advantages for extended residence times in the beam path, exploring multiple pulses for data analysis; however, it is restricted to in-air studies and is constrained by the difficulty of scaling down droplet size. It is also worth noting that electrospray injection is well-suited for generating very small droplets; however, applications for crystals in electrospray methods have not been reported so far[179–182].

The impact of these differences is also notable when comparing sample exposure efficiency. Larger droplets may allow for more than one exposure from X-rays, but can also increase the likelihood of not hitting a crystal in the droplet, particularly if the X-ray pulse duration or beam size does not match the speed or size of the droplet. Despite this, the innovations in droplet generation continue to push the boundaries, bringing us closer to achieving the optimal balance between sample conservation and effective data collection.

## Hybrid sample delivery systems

A promising sample delivery technique for both static and TR-SX combines features of liquid delivery and fixed target systems into a hybrid delivery approach. These systems leverage solid supports with minimal background scattering and liquid injectors for precise sample deposition. Hybrid methods have been developed to match the specific needs of TR-SX experiments, including studies enabled by pump-probe excitation, MISC, and gas reactant exposure techniques. While these systems can reduce sample consumption compared to continuous liquid jets, their more complex and sometimes bulkier designs can pose challenges when adapting to specific X-ray sources and end stations. Table 5 highlights the sample delivery conditions and expands on additional critical parameters, such as protein concentration, crystal size, and sample consumption, across the manuscripts referenced in this section.

### Drop-on-tape (DOT)

Early approaches to integrate liquid delivery and fixed targets were realized with a tape-drive system. In this system, a thin liquid stream is deposited onto a thin, X-ray-compatible film, which is then transported to the X-ray interaction region. This setup is versatile with respect to crystal size and mother liquor and has been implemented at both XFEL and synchrotron sources. One way these tape-drive systems can be utilized is by combining them with droplet delivery devices. Building on ADE systems developed by Roessler et al.[177,183,184], Fuller et al.[185] introduced the acoustically ejected droplets onto a tape drive (ADE-DOT) approach in 2017. This method integrates on-demand ADE onto a tape drive. To facilitate protein slurry and substrate mixing, the ADE-DOT approach employs a microfluidic mixing system before a nozzle deposits the small droplets onto Kapton or Mylar tape[186]. The tape then transports the droplets into the X-ray interaction region. This setup is compatible with both mixing reactions and light-induced reactions[185,187]. The ADE-DOT system's flexibility in adjusting droplet size allows for the analysis of a wide range of crystal sizes, resulting in moderate sample consumption. In Fuller's study[185], droplet volumes ranged from 0.6 to 8 nL, with crystal sizes between 20 and 100 μm. This achieved a 100% droplet hit rate and a sample consumption rate of 3.3 μL/min at 10 Hz, comparable to the low flow rates of continuously injecting GDVN systems.

Although Fuller's study was published in 2017[185], Young et al. had already demonstrated the ADE-DOT approach in 2016 to detect diffraction patterns of dark-adapted Photosystem II structures[188]. In their study, X-ray diffraction data were obtained from larger crystals (20–50 μm) than those typically used in serial crystallography with XFELs (5–15 μm). ADE-DOT delivery methods[189] are often favored for their high probing efficiency and compatibility with larger crystal sizes (>20 μm in droplets)[190], which is advantageous for X-ray emission spectroscopy and contributes to reduced sample consumption[129]. Furthermore, the combination with a tape delivery allows sampling on longer time scales relevant to biological reaction mechanisms[191]. Examples of studies employing the ADE-DOT approach at XFELs include investigations of gas-phase substrates delivered to aqueous media on millisecond time scales or above[192–195]. While tape-drive systems are versatile with respect to crystal size, this flexibility can affect the penetration depth of optical triggers and the diffusion time in mixing and gas permeation approaches[186]. Reaction times can be adjusted by modifying the tape-drive speed, but extended delay times may lead to dehydration challenges, particularly for aqueous samples[129]. Future work could explore strategies to mitigate these limitations, such as enhancing tape material properties to reduce evaporation or integrating humidity control systems in the sample chamber to maintain sample hydration.

An alternative mixing method for time-resolved experiments is the Drop-on-drop method, an extension of ADE-DOT. By modifying the ADE-DOT design to include two droplet dispensing heads, Butryn et al.[196,197] introduced an on-demand sample delivery system (see Fig. 5a) that delivers ligand solution in bursts of multiple drops onto a pre-dispensed, larger crystal-containing drop. This Drop-on-drop method uses a piezoelectric injector to dispense pL-sized ligand droplets while the ADE-DOT system dispenses nanoliter-sized protein crystal-containing droplets. The Drop-on-drop method was employed at the SACLA XFEL with crystals of up to 3–5 μm in size, though it is adaptable to a broader range of crystal sizes (5–100 μm) due to its flexibility in accommodating various droplet volumes. This approach offers high temporal resolution and maintains high hit rates while reducing sample consumption, requiring only 84–258 μL of ligand solution and 162–420 μL of microcrystal slurry[196]. Nguyen et al.[198] utilized the Drop-on-drop method to determine the SX crystal structure of the CYP121 reaction in situ, demonstrating its applicability to enzymatic studies. Additionally, the method's precise control over droplet size and timing allows for efficient use of reagents and minimizes sample waste.

When considering an ideal case for comparisons of delivery efficiency, the consumption of drop-on-drop approaches mirrors that of the droplet delivery system discussed in the section "Droplet-based injectors." In this ideal scenario, the total volume required for a complete dataset would be approximately 650 pL with a crystal density of $1.5 \times 10^{10}$ crystals/mL. When comparing these idealized values to the amounts reported in ADE-DOT studies, where crystal densities are typically on the order of $10^7$ crystals/mL (Table 5), the differences become apparent. These discrepancies underscore the ongoing challenges in aligning experimental realities with theoretical ideals for sample consumption. While ADE-DOT and Drop-on-drop methods represent significant advancements in flexibility and efficiency, their practical implementations often require higher sample volumes due to factors such as variability in crystal density, droplet alignment precision, and the need to optimize hit rates under experimental conditions. Furthermore, these systems are technically challenging to operate and sometimes too bulky to be integrated into an existing end-station.

**Table 5 | Overview of Hybrid Sample Delivery Methods in the literature, including details on X-ray source, flow rates, tape speeds, analyte, protein, and particle concentration, crystal size, and reported sample consumption**

| Authors | TR[a] (PP, MIX, O₂, N) | X-ray source (freq.) | End-station (environment) | Flow rate (µL/min) | Analyte(s) | Protein conc. (mg/mL) | Particle conc. (part/mL) | Crystal size | Sample consumption (volume [µL]/amount [mg]) | Tape speed (mm/s) |
|---|---|---|---|---|---|---|---|---|---|---|
| **Tape-drive** | | | | | | | | | | |
| Beyerlein et al.[71] | MIX | PETRA III (25) | P11 (He[b]) | $Q_{exp}=0.6$ | HEWL | 126 | NI | 6–8 µm | 300 µL/18.9 mg | 0.6 |
| Zielinski et al.[201] | N | PETRA III (25) | P11 (He[b]) | $Q_{exp}=$ (1) 2 (2, 3) 1 | (1) β-lactamase (2) GH11 xylanase (3) Urate oxidase | (1) 22 (2) 15 (3) 20 | NI | (1) 11–15 µm (2) 10–20 µm (3) 3–20 µm, 400 × 400 × 300 µm | (1) 91.4 µL/1.76 mg[a] | 1 |
| Lee et al.[202] | N | PAL (30) | NCI (He[b]) | $Q_{exp}=0.05–0.1$ $Q_r=0.05–10$ | HEWL | 50 | NI | 10 × 10 × 10 µm | NI | 1.5 |
| Henkel et al.[204] | MIX | PETRA III (130) | P11 (He[b]) | 1 | HEWL | 126 | NI | 38 × 38 µm | NI | 1 |
| **Drop-on-tape** | | | | | | | | | | |
| Young et al.[188] | PP | LCLS (10, 120) | CXI (vacuum[b]) XPP (He[b]) MFX (He[b]) | NI | Photosystem II | NI | NI | 20–50 µm | NI | NI |
| Fuller et al.[185] | PP/O₂ | LCLS (10, 30, 60) | XPP (He) MFX (He) | N/A | (1) Photosystem II (2) Phytochrome (PSM) (3) Phytochrome (PAS-GAF) (4) RNR | (1) NI (2) 40 (3) 40 (4) NI | NI | (1) 20–50 µm (2) 100 µm (3) 50 µm (4) 20–30 µm | (1) NI/2.4 mg (2) NI/6.1 mg (3) NI/2.9 mg (4) NI/6.9 mg | NI |
| Kern et al.[151] | PP | LCLS (10) | MFX (He[b]) | NI | Photosystem II | NI | NI | 20–60 µm | NI | NI |
| Burgie et al.[219] | N | LCLS (10) | MFX (He) | $Q_{exp}=2.5–3$ | TePixJ(GAF) | 25 | $1.2 \times 10^7$ | NI | NI | 300 |
| Ibrahim et al.[220] | PP | LCLS (20) SACLA (30) | MFX (He[b]) BL2-EH3 (He[b]) | NI | Photosystem II | 40–50 | NI | 20–60 µm | NI | NI |
| Srinivas et al.[195] | O₂ | SACLA (30) LCLS (30) PAL (15) | NI (NI) MFX (He) NCI (He) | $Q_{exp}$ (grease) = 20–80 $Q_{exp}$ (DOT) = 8 | sMMOH:MMOB | NI | NI | 20–30 µm | NI | 300 |
| Rabe et al.[194] | O₂ | LCLS (30) SACLA (30) | MFX (He) BL2 (He[b]) | $Q_{exp}$ (ADE-DOT) = 7 (v-ext) = 1–1.5 | isopenicillin N synthase | 50–52 | $5 \times 10^7$ | 3 × 3 × 40–60 µm | NI | 50, 93.75, 187.5, 300, 375 |
| Ohmer et al.[192] | N | LCLS (30[b]) | MFX (He[b]) | NI | MCRred1-silent | 40 | NI | 40–80 µm | NI | NI |
| Hussein et al.[221] | PP | LCLS (20) | MFX (He[b]) | NI | Photosystem II | NI | NI | 20–60 µm | NI | NI |
| Bhowmick et al.[222] | PP | LCLS (10, 20) SALCA (20) | MFX (He[b]) BL2 (He[b]) | NI | Photosystem II | NI | NI | 20–60 µm | NI | NI |
| Matika et al.[223] | PP, O₂ | LCLS (30) | MFX (He[b]) | $Q_r=7–9$ | (1) Photosystem II, (2) isopenicillin N synthase | (2) 50–52 | (1) $2 \times 10^6$ (2) $2 \times 10^7$ | (1) 20–50 µm (2) 40–60 µm | NI | 200–300 |
| Lebrette et al.[193] | N | LCLS (30) | MFX (He[b]) | $Q_{exp}=9$ | Class Ie R2 Protein Radical | 13 | NI | 10 × 10 × 5 µm | 316.8 µL[a]/4.1mg[a] | 300 |

**Table 5 (continued) | Overview of Hybrid Sample Delivery Methods in the literature, including details on X-ray source, flow rates, tape speeds, analyte, protein, and particle concentration, crystal size, and reported sample consumption**

| Authors | X-ray source (freq.) | End-station (environment) | TR[a] (PP, MIX, O₂, N) | Flow rate (µL/min) | Analyte(s) | Protein conc. (mg/mL) | Particle conc. (part/mL) | Crystal size | Sample consumption (volume [µL]/ amount [mg]) | Tape speed (mm/s) |
|---|---|---|---|---|---|---|---|---|---|---|
| Butryn et al.[196] | SACLA (30) | BL2 (He[b]) | MIX | N/A | (1) HEWL mixed with GlcNAc (2) CTX-M-15 mixed with ertapenem | (1) 50 (2) 20 | (1) ~10⁷ (2) 8 × 10⁷ | (1) 3–5 µm (2) 5×10–20 µm | (1) 932 µL/42.6 mg (2) 995 µL/8 mg | (1) 300, 100, 30 (2) 100, 30 |
| Nguyen et al.[198] | LCLS (30) | MFX[b] (He[b]) | MIX | $Q_{exp}$ = 3.6 | CYP121 | 10–14 | NI | 30 µm | NI | 200 |
| **Other hybrid methods** | | | | | | | | | | |
| Mathew et al.[207] | LCLS (10) | XPP | N | N/A | Polyketide synthases | NI | NI | 50 × 10 × 2 µm | 300 µL/3.3 mg | 2.5 |
| Mehrabi et al.[191] | PETRA III (25[b]) | P14 (air[b]) P14-2 (He[b]) | MIX | NI | (1) xylose isomerase (2) HEWL mixed with GlcNAc | (1) 80 (2) 67 | NI | NI | NI | NI |

Sample consumption is recorded in volume and protein amount in milligrams. Time-resolved experiments in the second column are classified into pump-probe (PP) for light-induced, mix and inject (misc). oxygen evolution reactions (O₂), and N for not time-resolved.

[a]The sample consumption value was estimated from the reported protein concentration in the mother liquor, considering that all protein molecules were converted to crystals considering the reported volumes consumed or the amount of protein consumed. NI refers to data not included in the manuscript. N/A refers to data not applicable to the experimental type reported in the manuscript.

[b]The conditions in which the experiment was conducted are not explicitly stated, and the recorded condition is assumed based on the X-ray instrument and end-station used. Q_exp: The flow rates recorded were given in a range.

Future improvements could be focused on increasing droplet throughput to match higher repetition rate X-ray sources and refining microfluidic mixing designs that reduce ligand waste. Further innovation in the area of tape-drive materials or droplet deposition techniques might also reduce splashing during droplet delivery or adverse evaporation during transport to the X-ray interaction region. Adaptive feedback systems that dynamically adjust droplet size, deposition timing, or mixing parameters could optimize sample usage in real time. Finally, combination approaches, where ADE-DOT systems are combined with other approaches, such as advanced optical monitoring of droplets, might permit the real-time observation of droplet integrity, positioning, and reaction progress, reducing excess preparation of samples. This increases the throughput of the systems as well as allows the application of such methods toward more complex biological systems and for longer time-resolved studies. The performance will only approach the theoretical case for delivery efficiency, along with the high temporal resolution and hit rates needed for XFEL studies, by addressing these challenges in the development of future ADE-DOTs and Drop-on-drop configurations.

**Liquid stream on tape-drive**

Tape drive systems are also capable of time-resolved studies by synchronization of liquid deposition, subsequent mixing, and laser excitation with controlled crystal transportation on the tape[187,199,200]. This synchronization allows for precise timing of the sample arrival and replenishment in the interaction region, facilitating investigations of dynamic processes on ultrafast timescales. In 2017, Beyerlein et al.[71] introduced a tape-drive method using a polyimide tape for mix and diffuse serial crystallography. This method demonstrated the structure of lysozyme bound to chitotriose by depositing a protein crystal suspension mixed with a ligand solution onto a continuously moving tape drawn from a feeder roll to a collector roll. Realized at a Synchrotron source, the system's normal angle of incidence facilitated data collection while maintaining precision. By controlling the sample flow rate and tape speed to avoid multiple crystal hits, approximately 19 mg of protein were consumed per dataset, demonstrating the potential for high-throughput drug screening and rapid structural enzymology[71].

In 2022, Zielinski et al.[201] refined the tape-drive system, termed the CFEL Tape-Drive, to achieve faster and more efficient data collection (see Fig. 5b). This method enabled the high-yield production of homogenous microcrystals by eliminating the tedious reservoir loading step and allowed uninterrupted data collection without requiring experimental hutch access between sample acquisitions. With a sample consumption rate of 0.2–2 µL/min enabled with a low-pressure pump system, this approach significantly reduced sample consumption compared to the Beyerlein et al.[71] study.

A related approach, the Beamline Integrated Tape System, was developed by Park et al. and implemented at PAL-XFEL[164,202,203]. This system combines inject and transfer mechanisms, where a crystal suspension is deposited on a polyimide film via an injection needle and transported to the X-ray integration region using a translation stage. Notably, the sample stream can be scanned both horizontally and vertically, enhancing the data collection rate while maintaining low sample consumption. Using this approach, only 1.4 µL was required to solve the structure of Lysozyme.

More recent advancements in tape-drive technology were demonstrated by Henkel et al.[204] in 2023 with the development of the Just-In-Time-Crystallization for Easy Structure Determination (JINXED). This method adapted a 3D-printed GDVN to mix protein and precipitant solutions immediately before deposition on the tape, enabling crystallization on the fly. This approach circumvents the need to handle sensitive microcrystalline solutions between crystallization and data collection, which is advantageous for fragile crystal systems. However, its application to a broader range of proteins remains limited

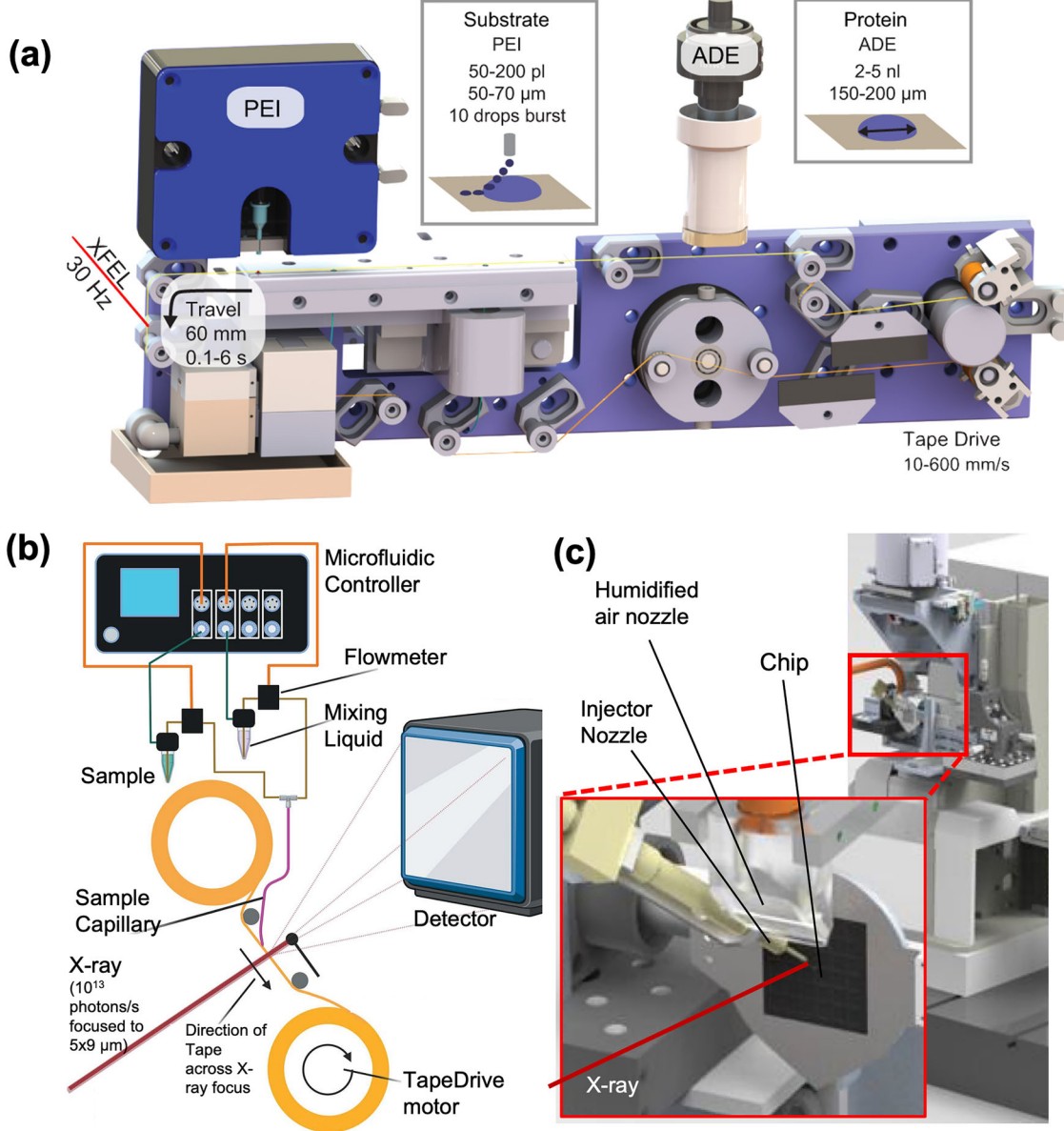

**Fig. 5 | Examples of hybrid sample delivery methods. a** An on-demand drop-on-drop method developed by Butryn et al.[196] to perform TR-SX. In this method, substrate droplets were ejected onto protein crystal droplets using a piezoelectric injector (PEI), while the protein crystal droplets themselves were deposited onto the tape by an acoustic droplet ejector (ADE). The mixed droplets were then carried to the XFEL interaction region through the tape drive. **b** A tape drive sample delivery method described by Zielinski et al.[201] where crystal suspension is deposited on a polyimide tape that is mounted on motors for exposure to an X-ray beam. **c** A liquid application method for time-resolved crystallography developed by Mehrabi et al.[191] where a fixed target SX approach is combined with a piezo-driven droplet injector. The device mount is shown with an inset showing the piezoelectric droplet ejector in front of an open fixed-target device. Reprinted with permission from Zielinski et al.[201] and adapted with permission from Butryn et al.[196] and Mehrabi et al.[191].

due to the challenge of optimizing crystallization conditions. While lysozyme crystallizes within seconds, many proteins typically require hours or days to form suitable crystals, making it difficult to adapt JINXED to proteins with more complex crystallization requirements.

While the sample requirements for liquid-stream-on-tape methods enable a more thorough analysis of the solutions deposited, they are still far from ideal. In an ideal case, assuming a $4 \times 4 \times 4\ \mu m^3$ cubic crystal is contained within a liquid stream just wide enough to keep it properly hydrated (5 μm) and the crystals are spaced 20 μm apart, the volume required for a full dataset (10,000 patterns) would be ~4 nL. Given that there are 10,000 crystals in that volume, the ideal crystal concentration would be $2.5 \times 10^9$ crystals/mL. While this number is roughly the concentration of crystals typically targeted in a liquid injection experiment, handling 4 nL is not feasible. This limitation arises not only from the technical challenges of managing such a minuscule volume but also from the difficulty in maintaining consistent crystal dispersion within the stream of deposited solution. While handling volumes as low as 4 nL presents technical challenges, it is important to emphasize that one of the key advantages of serial techniques is the ability to work with microliter volumes of crystal suspensions rather than requiring milliliter quantities, making them inherently more sample-efficient despite some acceptable wastage. Achieving a balance between minimizing sample consumption and maintaining data quality remains a critical challenge and may be alleviated with future automated sample-loading approaches.

**Other hybrid methods**

Another notable hybrid method for in situ mixing in SX is the Liquid Application Method for Time-Resolved Analyses (LAMA), developed by Mehrabi et al.[191] in 2019. LAMA combines a piezo-driven droplet injector with a fixed-target SX approach to achieve sample-conserving, time-resolved studies (see Fig. 5c). In this method, pL-sized droplets are ejected from the piezo-injector onto chip-mounted protein crystals, allowing for the mixing of substrates with the protein crystals. To prevent crystal dehydration, the chip is kept in a constant stream of humidified air, an innovation that overcomes the limitations of traditional tape-based methods.

This hybrid on-chip mixing technique enabled data acquisition across a wide range of time scales from milliseconds to several minutes, without significantly extending the data collection times. By incorporating the piezo-based droplet injector, the technique also enables the use of the Hit-and-Return (HARE) approach, which minimizes background scattering, further improving data quality[205,206]. Moreover, LAMA is efficient in sample use, requiring only 1.5–3 μL ligand solution, while a single chip can accommodate around 1–2 mg of protein and approximately 20,000 crystals (Table 5). The ideal sample consumption for LAMA would be based on the fixed target theoretical minimum outlined in section A of 29 nL, containing a crystal density of $3.5 \times 10^8$ crystals/mL. The LAMA method, using the chip by Owen et al., used 60–80 μL of sample and up to 5 mg of protein[92]. This discrepancy underscores the inherent challenge of optimizing sample efficiency in LAMA, as the practical application of this method currently requires significantly more material than the theoretical ideal. Continued advancements in chip design, integrated with efforts to decrease volumes of ejected droplets, could bridge this gap, allowing for further reductions in sample consumption while maintaining data quality and throughput.

A different hybrid approach, the Crystal Extractor, developed by Mathews et al.[207], introduces a way to deliver microcrystals directly from the native crystallization solution into the X-ray beam under controlled environmental conditions. This method is compatible with XFEL and synchrotron sources and can handle an assortment of crystal sizes[75]. The Crystal Extractor employs a solenoid driver to insert a crystal support device, such as a mesh or thin film, into a solution of crystals in their mother liquor. This substrate is then quickly removed, extracting a thin liquid film on the support device that contains a random distribution of crystals, which is then brought into the path of an X-ray beam for data collection. After each exposure, the support device is repositioned to expose previously unexposed crystals. This process is repeated, allowing for fresh batches of crystals to be extracted and exposed sequentially.

One of the key advantages of the Crystal Extractor is its ability to minimize sample consumption and waste. Unexposed crystals are

quickly returned to the solutions, providing rehydration and another opportunity to be exposed during subsequent extractions. However, this method also has a downside: crystals that are damaged during exposure are also returned to the solution. These damaged crystals could potentially be re-imaged in future extractions, compromising data quality. In comparison with the ideal sample consumption for a typical SX experiment requiring 650 pL, the reported 300 μL used by the Crystal Extractor system is significantly higher. While the Crystal Extractor's ability to recycle unexposed crystals provides a way to mitigate waste, the relatively large sample volumes employed could be a limiting factor for high-throughput experiments or those using limited or expensive samples. Additionally, because the sample is being recycled, it is not applicable for time-resolved studies.

## Summary and future trends

The delivery of samples has long been a pivotal area of research in X-ray crystallography, and its significance has grown substantially over the past decade, driven by advancements in next-generation synchrotron radiation sources and the development of XFELs. With the above sections, we have summarized existing protein crystal delivery approaches employed in SX for synchrotron and XFEL crystallography. It is evident that all sample delivery approaches provide much larger quantities than theoretically required to obtain a full dataset. We had established an ideal amount of 450 ng based on the crystal structure of the 31 kDa NQO1 protein and 10,000 indexable patterns required to obtain a full dataset (see above sections). For the discussed techniques, we have established volume and crystal concentration values that would comply with these assumptions. It is clear that none of the reported methods provides such efficiency in sample consumption, and we caution that the ideal sample consumption greatly depends on the protein to be studied by SFX or SMX techniques. However, such an ideal estimate is a good guideline to point out the current bottleneck in sample consumption in the SX field and to steer the field into future sample consumption efficiency improvements. Figure 6 is thus an interesting overview of the reported sample consumption by different delivery methods under the three sample injection categories broadly classified in this review as listed in Tables 1 and 3–5. Clearly, the sample usage efficiency of fixed-target methods typically reports the least sample consumption (below 1 mg); meanwhile, hybrid methods like DOT have reported the highest sample consumption (well above 10 mg), even higher than liquid GDVN injections in some studies.

However, the actual sample consumption in the studies reviewed here is often not reported; thus, we suspect that in some cases, consumed sample amounts may have been much higher, potentially up to gram amounts of protein for a single structure. While our comparisons only take into account crystal size and the placement or alignment in a

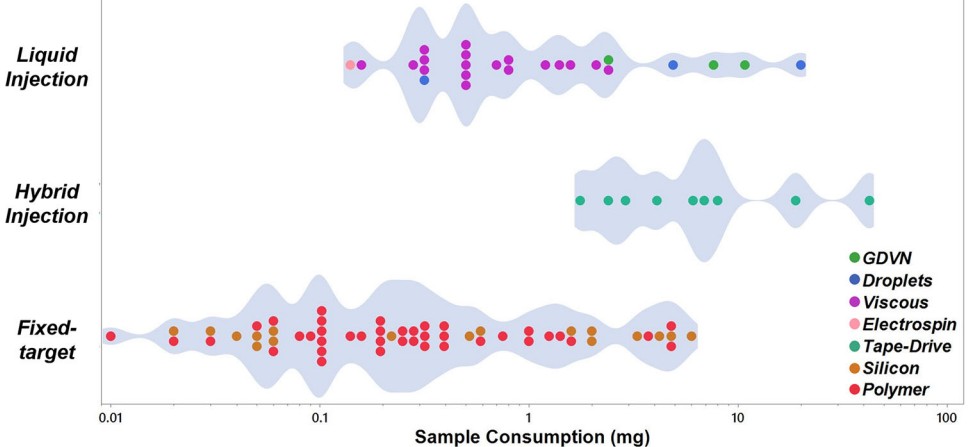

**Fig. 6 | Overview of the sample consumption.** The different injection methods discussed in this review are summarized from over 90 reports enlisted in Tables 1 and 3–5.

particular sample delivery strategy, other experimental parameters related to the particular X-ray instrument and end-station, as well as the optimization of the crystal sample itself, such as the crystal size, may lead to a further increase in the sample consumption requirement. Nonetheless, this compilation of methods and their sample consumption comparison gives a comprehensive overview of the capabilities of existing sample injection methods and is further summarized below. It needs to be noted that efforts to find suitable crystallization conditions may also account for sample consumption needs; however, due to the large variability in protein types and required conditions, this aspect has not been considered in our review. The summarized and discussed sample delivery methods may, however, be useful for researchers aiming to perform a particular SX experiment, knowing the amounts of protein crystals available for their particular sample.

As outlined in this review, the fixed-target sample delivery method, a cornerstone of crystallography since its inception, has undergone considerable innovation in terms of design and materials, particularly in response to improvements in X-ray sources. However, reducing sample consumption has not been a major focus in the development of fixed-target systems, which typically require sample volumes of ≤10 μL and 1 mg or less protein amounts, which are much smaller, sometimes orders of magnitude, than the liquid injection methods as shown above in Fig. 6. Most of the developed fixed-target devices have employed model proteins as proof of concept. Thus, for future device development, it will be important to demonstrate the capability of using non-standard proteins along with model proteins. Furthermore, one key advantage of fixed-target methods is the ability to grow crystals on-chip, enabling direct delivery to the X-ray beam for data collection. A notable limitation of the fixed-target approach, however, is its limited applicability to TR-SX, where liquid sample delivery has traditionally been the preferred method for non-light-driven reactions. The future effort thus needs to be directed to the development of fixed-target devices allowing mixing experiments prior to X-ray exposure. This could be achieved by the integration of miniaturized elements to allow mixing, including the precise positioning of crystals with respect to the corresponding solutions to be mixed. Another future improvement in fixed-target approaches could originate from automated sample delivery, reducing the "dead volume" introduced through manual pipetting steps currently required in most fixed-target device applications.

Liquid delivery systems are more commonly used in XFEL experiments, particularly for pump-probe and mix-and-inject experiments that are designed to probe protein dynamics in TR-SX. While liquid delivery offers clear advantages for time-resolved studies, it has been hindered by significant challenges related to sample consumption, where the amount of protein needed for a full dataset is multiplied by the number of time points probed, requiring up to gram amounts of protein. In response, considerable efforts have been made to reduce the amount of precious protein required for these experiments, leading to the development of more efficient systems. Innovations such as synchronized droplet generators and HVEs have successfully minimized sample consumption compared to earlier methods like GDVNs, as detailed in this review and evident in Fig. 6. Recent advancements have sought to combine the strengths of both fixed-target and liquid delivery methods, opening the possibility of achieving ideal sample consumption. One notable example is the LAMA system, which generates picoliter-sized droplets and delivers them to the X-ray beam via a fixed-target system. However, it is important to note that sample consumption is influenced not only by the delivery method but also by the parameters of the X-ray source itself. As X-ray sources vary widely in their characteristics, designing a universal delivery system that performs optimally across all platforms remains a significant challenge. The process of optimizing these systems requires time and extensive iteration, which is often difficult to achieve given the rapidly evolving needs of X-ray facilities.

Furthermore, the limited time availability at these facilities makes the optimization process even more challenging for experimenters. A promising solution to this challenge may come with the development of CXLSs and compact X-ray free-electron lasers, such as the one being developed at Arizona State University[102–104]. These pioneering X-ray sources could make it possible to increase access to X-ray facilities, thereby enabling more efficient optimization of sample delivery systems. With the greater availability of such compact sources, the process of refining and optimizing delivery methods could be accelerated significantly to further reduce sample consumption. Looking ahead, the combination of fixed-target and liquid delivery systems is likely to provide a path toward approaching the ideal sample consumption while enabling the execution of critical X-ray crystallography experiments for protein structure determination. Future advancements through the integration of innovative microfluidic tools could improve control over microfluidic systems to handle smaller liquid volumes with high precision. Innovations in crystal growth techniques that produce more uniform crystals and strategies to reduce sample wastage during injection could also significantly improve the efficiency of these methods. As the field progresses, combining machine learning and artificial intelligence approaches to design but also optimize injection methods while operating will contribute to optimizing sample delivery and reducing the disparity between theoretical and practical sample requirements.

### Reporting summary
Further information on research design is available in the Nature Portfolio Reporting Summary linked to this article.

## Data availability
Data availability is assured through the published works referenced in this review article. Data points in Fig. 6 are obtained from the tables provided with this article.

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

## Acknowledgements

The authors greatly acknowledge funding from the National Science Foundation through the BioXFEL Science and Technology Center (agreement #1231306, A.R., D.D., M.Sr.), as well as under Mid-scale Research Infrastructure-2 award No. 2153503 (A.R., A.M.) and Mid-Scale RI-1 award No. 1935994 (A.R., A.M.).

## Author contributions

A.R. conceived the original idea of this review. D.D., A.M., M.So., and A.R. contributed to organizing the structure of the manuscript. D.D., A.M., M.So., M.Sr., and A.R. contributed to the literature review and wrote sections of the manuscript.

## Competing interests

Authors A.M., M.So., and A.R. hold a provisional patent on fixed-target devices for serial crystallography applications. Authors D.D., A.M., M.So., and A.R. hold a patent on sample delivery with segmented droplet formation. D.D. and A.R. hold a patent on co-flow injection.
