## [Transparent Peer Review file · Nature Communications]

Sample Delivery Methods for Protein X-ray Crystallography with a Special Focus on Sample Consumption

Corresponding Author: Dr Alexandra Ros

Version 0:

Reviewer comments:

Reviewer #1

(Remarks to the Author)

The article "Sample Delivery Methods for Protein X-ray Crystallography with a Special Focus on Sample Consumption" by Abhik Manna et al. is an extended review on one of the most challenging aspects of serial protein crystallography, namely sample consumption. Obviously, high sample consumption implies higher costs for the experiments, both in terms of resources and time. Additionally, it can discourage new groups or users from performing serial crystallography experiments, preventing the method from being widely applied. In parallel, the question of the best sample delivery method in terms of protein consumption is frequently discussed within the community of expert users of serial crystallography. Hence, the presented review article is timely and highly relevant. The article covers a very wide range of sample delivery methods, including lesser-known techniques, and explains their original application as well as further improvements and iterations. In particular, it summarizes the published works in a very useful set of tables and attempts to quantify the amount of sample protein consumed in each experiment, which is a very non-trivial and time-consuming task. An expert in the field would follow the article easily but could be troubled by some aspects summarized in the general comments below.

I would recommend it for publication after the general and detail comments are addressed.

General comments:

1. XFEL and synchrotron experiments are placed under the umbrella of new generation X-ray sources. Historically, serial techniques were developed for XFELs and then "transferred" to synchrotron sources, both 3rd and 4th generations. A clearer and more in-depth discussion of this could be presented in the introduction for non-experts.

2. The same comment applies to the distinction between static and time-resolved measurements. The authors could elaborate a bit more in the introduction rather than just saying, "Eventually, the technique was further evolved from static structures to obtaining molecular movies..." on page 3. Furthermore, in the literature, two terms are used to distinguish time-resolved experiments at XFELs and synchrotrons, namely TR-SFX and TR-SMX, i.e., time-resolved serial femto- and millisecond crystallography, respectively. Again, a comment for non-experts would be useful.

3. Figures presented in the article look rushed and unprofessional, resembling simple "screenshots" from published work. Examples include: Fig 1(a) – image being highly distorted; Fig 2(c) – text and details very small and almost unreadable; Fig 2(d) too small to read; Fig 3 very "crowded" with no scales of dimensions and almost no labels; Fig 4 – no labels explaining different parts of the instruments. This is very concerning, especially for a review article. The article by Barends TRM, Stauch B, Cherezov V, Schlichting I, "Serial femtosecond crystallography" (Nature Reviews Methods Primers 2, 59, 2022) is a good example with high-quality figures that are easy to follow, especially for non-experts.

4. A schematic figure or chart should be included with a summary of different delivery methods. This can be done in various ways. A hierarchical chart would be most applicable. Information on which sample delivery method is the most widespread would be useful and could be part of such a chart.

5. The "Ideal situation," i.e., the ideal case of sample consumption, is introduced on page 5 and is used as a metric throughout the article. When comparing this metric to the reviewed published work, there is a discrepancy of two orders of

magnitude or more, e.g., on pages 10, 11, 13, 16, 20, and 22. Given that this discrepancy is so consistent and large, it implies that the metric might not be the most useful tool for comparison between different delivery methods. In short, the metric does not help the reader decide which method is better. The authors should address this concern.

6. The section summarizing different delivery methods often quantifies sample consumption in different units, e.g., 1-5 mg on page 8, 3.2-16 μL of sample solution on page 13, 1-10 $\mu\text{L}/\text{min}$ of sample solution on page 19, and 0.7-3 μL of sample volume on page 20. While these refer to different techniques (hence volumes instead of mass or volume rates), they should nonetheless be quantified in the same way, so the reader can easily compare the numbers.

7. In the opinion of the referee, the article could be written more concisely, as some sections are repetitive or redundant (e.g., the ideal consumption requirement is stated several times throughout the article, i.e., 10,000 patterns with $4 \times 4 \times 4 \mu\text{m}^3$ crystals). See also detailed comments annotated in the PDF attached to this review.

8. Finally, the discussion section could be improved by writing it in the style of a guidebook for a new user. Serial crystallography experiments are complex, as already explained, so it would be useful for non-experts to have some guidance not only in terms of sample consumption (the topic of the review) but also regarding other challenges associated with executing these experiments. An attempt could be made to speculate on which method will consume the least amount of sample with the lowest "activation barrier" in terms of user expertise.

Reviewer #2

(Remarks to the Author)

General comments:

This review provides an good overview of serial crystallography over the past 10 yrs with a particular focus on sample consumption. Overall, it is written well and was a joy to read. It covers a many different delivery systems and assesses their sample volume and relates it back to the 'ideal' sample volume for crystallographers to generate a data set. Sample consumption is a hot and debatable topic in this field. While the authors have presented a standardized way to compare each sample delivery system, the challenge lies with what you consider is the 'ideal sample volume' and what is acceptable in the field. For example, to setup and grow large well diffracting crystals of an unknown protein for traditional crystallography may take 100s of crystallization screens. If you estimate a protein concentration of minimum 20mg/ml in a nano droplet (50nl) and set up a 1x96 well screen with duplicate wells. Once crystallization plate would require ~0.192 mg of protein for the one screen, if you multiple this by just 10 screens it now becomes 1.92 mg. It may take 100's of screening condition to get a few large crystals. Therefore the amount of protein for optimizing a large good quality crystals maybe in the order of 5-10 mg for example. SX offers an opportunity to use shower of good quality crystals which you may hit in the first few screens and then go straight into SX, the amount of protein you saved in screen you can justify for SX data collection. I think term sample consumption is relative and one must make it clear in this review what it should be really compared to and justify what is denoted as high sample consumption. This would be a valuable comment to make in this review.

Fix-targets- Has there been consideration of crystal orientation? Is there an issue associated with preferential orientations of the crystal on a surface which can be noted. Does this lead to higher number of crystals required to collect a full data set? Do the authors see this problematic for testing of future protein targets (non-standards) and are there any efforts in the literature to help combat this? My point here is that Table 1 shows a nice summary of fixed-target structures where the protocols have worked well, and structures have been solved, however, ~65% of the target structures solved in the table are standard proteins used for crystallography which are easily crystallized and well characterized. For unknown crystal samples the complexity lies within the crystallization environment and crystal surface/shape/charge etc. So, what maybe straight forward for lysozyme maybe challenging for non-standard crystals grown. Any comment in this area the authors could make and can be guided by the literature would be valuable to the readers.

- One of the first silicon chip filtration was tested by Alke Meents group publication not mentioned here, Roedig, I Vartiainen, R Duman, S Panneerselvam, N Stübe, O Lorbeer, M Warmer, G Sutton, DI Stuart, E Weckert, C David, A Wagner, A Meents, Scientific 2015, p10451 publication not cited

HVE section- although the authors primary focus on HVE in the context of XFELs, synchrotron source are also important. One might argue more ideal for HVE. Syringe type injectors have less wastage compared to viscous injector and depending on crystal size (larger crystals), there is a possibility for multiple hits on the same crystals at a slow flow rate and micro focus beam line. A couple of papers which demonstrate this but not limited to are Botha S, et.al. Acta Crystallogr D Biol Crystallogr. 2015 Feb;71(Pt 2):387-97 and Hadian-Jazi, M.; et.al. Analysis of Multi-Hit Crystals in Serial Synchrotron Crystallography Experiments Using High-Viscosity Injectors. Crystals 2021, 11, 49. This is not mentioned in this review.

P29, para 2 line While the authors make the comment 'handling 4 nL is not feasible. This limitation arises not only from the technical challenges of managing such a miniscule volume but also from the difficulty in maintaining consistent crystal dispersion within the stream of deposited solution.' I tend to disagree with this statement. Isn't the whole to aim is to lower sample consumption as much as possible? Having stock solutions of 4nl is a challenge for any application. Even nano drop crystallization protocols where minimum dispensing volumes for proteins are as low as 50 nl. Even this setup requires micro liter stock volumes of protein. The real advantage here is that we don't need to use mls of sample and can realistically get away with ul stock volumes of crystals for this technique. Automated sample loading will help but the question I ask here is do when then have a limit on the minimum volume for SX as of all techniques there is a reasonable amount of wastage that

occurs which is acceptable. This is something that the authors do mention later in the article but maybe worth mentioning here too.

P31, last paragraph 'Continued advancements in chip design, sample delivery systems, and crystal handling techniques could bridge this gap, allowing for further reductions in sample consumption while maintaining data quality and throughput.' Can't this statement be said for all techniques that have been developed not just LAMA?

Specific comments:

P4, para 3, line 1:

'This challenge of sample consumption along with well-timed advances in microfabrication and 3D printing techniques in the last decade led to a wildfire of new sample injection techniques allowing SX as well as approaches to conserving precious crystal samples.

- This sentence doesn't make sense. I think there is a word missing "other"

P10, para 1, line 1 'Kapton is another widely used material for fixed-target systems owing to the low X-ray background scattering from the material.'

-References are need here. In the context of SFX some of the first experiments were conducted using Kapton fixed targets eg. Ryan RA et.al J Vis Exp. 2017 and more recently Bosman et al. Cell Reports Physical Science, 2024, 101987 provided a good comparison study not mentioned in this review.

P16 para 1 'A smaller sample volume with a higher crystal density would be required for the same full dataset since more pulses are delivered in a faster time: 2.0 μL of sample with a density of 5×10^6 crystals/mL.'

- shouldn't this be lower crystal density not higher if you are comparing it to 4×10^{10} ????

Fig 2b) label 'Enclosing polyimide film' is cut off.

Fig3j. Labels in insert is too small and unclear.

Fig 4. Labeling of different components in figure would be helpful.

Version 1:

Reviewer comments:

Reviewer #1

(Remarks to the Author)

The reviewer appreciates the authors' rebuttal comments and acknowledges the overall improvement of the manuscript, particularly the enhanced quality of the figures. The reviewer specifically welcomes the inclusion of a new figure (Figure 6) and the revisions made to the Summary section.

With Figure 6 now incorporated into the main text, it may be appropriate to move the corresponding table to the supplementary information. The reviewer suggests that the authors and the editor discuss the best course of action to avoid making the manuscript unnecessarily lengthy.

A PDF containing minor editorial changes as well some open questions has been attached. The reviewer strongly recommends a thorough re-reading of the manuscript, as it contains numerous typographical and grammatical errors—particularly in the sections added during the rebuttal.

In the Summary section, the authors state: "However, such an ideal estimate is a good guideline to point out the current bottleneck in sample consumption in the SX field and to steer the field into future sample consumption efficiency improvements." Does this imply a direction toward the application of fixed targets? While the authors thoroughly discuss the nuances of various delivery methods, they do not address their user-friendliness. It would be valuable to hear the authors' perspective on which methods are most suitable for non-experts. The reviewer is also inclined toward the use of fixed targets, while fully acknowledging their limitations.

The reviewer would recommend the manuscript for publication once the points in the attached PDF document are addressed.

Reviewer #2

(Remarks to the Author)

I am satisfied with the updated manuscript provided by the authors which have improved the review. It provides a very nice review of the sample delivery methods and sample consumption in the field. I recommend this manuscript for publication.

Version 2:

Reviewer comments:

Reviewer #1

(Remarks to the Author)

The reviewer is satisfied with the revised manuscript and the amendments made during the revision process. In the reviewer's opinion, the manuscript is ready for publication.

Reviewers' comments:

Reviewer #1 (Remarks to the Author):

The article “Sample Delivery Methods for Protein X-ray Crystallography with a Special Focus on Sample Consumption” by Abhik Manna et al. is an extended review on one of the most challenging aspects of serial protein crystallography, namely sample consumption. Obviously, high sample consumption implies higher costs for the experiments, both in terms of resources and time. Additionally, it can discourage new groups or users from performing serial crystallography experiments, preventing the method from being widely applied. In parallel, the question of the best sample delivery method in terms of protein consumption is frequently discussed within the community of expert users of serial crystallography. Hence, the presented review article is timely and highly relevant. The article covers a very wide range of sample delivery methods, including lesser-known techniques, and explains their original application as well as further improvements and iterations. In particular, it summarizes the published works in a very useful set of tables and attempts to quantify the amount of sample protein consumed in each experiment, which is a very non-trivial and time-consuming task. An expert in the field would follow the article easily but could be troubled by some aspects summarized in the general comments below.

I would recommend it for publication after the general and detail comments are addressed.

Response: We thank the reviewer for judging our review to be timely and highly relevant. Below, we provide detailed comments on each critique mentioned.

General comments:

1. XFEL and synchrotron experiments are placed under the umbrella of new generation X-ray sources. Historically, serial techniques were developed for XFELs and then “transferred” to synchrotron sources, both 3rd and 4th generations. A clearer and more in-depth discussion of this could be presented in the introduction for non-experts.

Response: To satisfy this request, we have rephrased the last two sentences of the first paragraph in the introduction.

2. The same comment applies to the distinction between static and time-resolved measurements. The authors could elaborate a bit more in the introduction rather than just saying, “Eventually, the technique was further evolved from static structures to obtaining molecular movies...” on page 3. Furthermore, in the literature, two terms are used to distinguish time-resolved experiments at XFELs and synchrotrons, namely TR-SFX and TR-SMX, i.e., time-resolved serial femto- and millisecond crystallography, respectively. Again, a comment for non-experts would be useful.

Response: We thank the reviewer for this comment. We have changed the text in the paragraph on page 3, first explaining SFX and then SMX to make this point clear to the reader. Due to the large information content on injection methods, we have then opted to define all serial crystallography methods (regardless of X-ray source used) SX, to make the flow of the review and discussion of sample delivery techniques easier to follow. In our opinion, it would simply be too complex to keep the distinction in the remainder of the manuscript. However, our detailed tables list the x-ray source and instrument such that the interested reader will find the information in our manuscript.

3. Figures presented in the article look rushed and unprofessional, resembling simple “screenshots” from published work. Examples include: Fig 1(a) – image being highly distorted; Fig 2(c) – text and details very small and almost unreadable; Fig 2(d) too small to read; Fig 3 very “crowded” with no scales of dimensions and almost no labels; Fig 4 – no labels explaining different parts of the instruments. This is very concerning, especially for a review article. The article by Barends TRM, Stauch B, Cherezov V, Schlichting I, "Serial femtosecond crystallography" (Nature Reviews Methods Primers 2, 59, 2022) is a good example with high-quality figures that are easy to follow, especially for non-experts.

Response: We thank the reviewer for this comment and agree. We have carefully reworked the figures improving their quality in content and resolution. Specifically, we updated Figures 1a, 2b, 2c, 2d. Furthermore, Figures 3 and 4 have been revised for clarity and visual coherence. The layout was reorganized to reduce clutter and represent the injection systems more prominently. Labels have been added above each injection type to aid identification, and the legend has been moved and rearranged to increase visibility. However, scale bars could not be included, as the original published sources for the injection systems did not report dimensions with sufficient accuracy. We also revised Figure 5 for easier visibility and included labels for improved comprehension. This also prompted us to change the order of sections C1 and C2 to match the order in Figure 5.

4. A schematic figure or chart should be included with a summary of different delivery methods. This can be done in various ways. A hierarchical chart would be most applicable. Information on which sample delivery method is the most widespread would be useful and could be part of such a chart.

Response: We agree that a summary figure would improve our manuscript. Due to the complexity of the provided information we decided to represent the sample consumption in the form of a violin graph. We have now included Figure 6 showing overall sample consumption by different delivery systems under the three broad injection types described in this review. This new Figure 6 is included and discussed in the conclusion section.

5. The “Ideal situation,” i.e., the ideal case of sample consumption, is introduced on page 5 and is used as a metric throughout the article. When comparing this metric to the

reviewed published work, there is a discrepancy of two orders of magnitude or more, e.g., on pages 10, 11, 13, 16, 20, and 22. Given that this discrepancy is so consistent and large, it implies that the metric might not be the most useful tool for comparison between different delivery methods. In short, the metric does not help the reader decide which method is better. The authors should address this concern.

Response: We thank the reviewer for this concern, but we believe the ideal sample consumption as we define it, is a metric that helps point out that sample consumption is indeed a major bottleneck and that there is “plenty of room at the bottom” where future sample consumption reduction can make SX techniques more significant in the field. We agree that the current and reviewed sample consumption methods are mostly orders of magnitude higher, but we list the currently achievable sample consumption in detail in the tables. In addition, we discuss the best cases in relation to the ideal sample consumption, such that the reader can learn about the available methods, but also learns that more samples could be conserved. We have added a sentence in the conclusion section (first paragraph) and have added a summary figure to further emphasize the existing bottleneck.

6. The section summarizing different delivery methods often quantifies sample consumption in different units, e.g., 1-5 mg on page 8, 3.2-16 μL of sample solution on page 13, 1-10 $\mu\text{L}/\text{min}$ of sample solution on page 19, and 0.7-3 μL of sample volume on page 20. While these refer to different techniques (hence volumes instead of mass or volume rates), they should nonetheless be quantified in the same way, so the reader can easily compare the numbers.

Response: We fully realize this fact, but we found that the way we are representing the consumption (in either volume or mass) is an adequate way of representation. In fact, most publications cite only one or the other and we made assumptions to be able to compare all cases. We are providing both volume of sample and mass of protein, so the reader can get a better idea of what the particular tabulated value and unit means. We strongly believe this is an adequate way of representing sample consumption, keeping in mind that our ideal case is based on protein mass realizing that protein crystals are suspended in their mother liquor and can only be used for serial crystallography techniques when starting from a suspension.

7. In the opinion of the referee, the article could be written more concisely, as some sections are repetitive or redundant (e.g., the ideal consumption requirement is stated several times throughout the article, i.e., 10,000 patterns with $4 \times 4 \times 4 \mu\text{m}^3$ crystals). See also detailed comments annotated in the PDF attached to this review.

Response: We agree with some degree of redundancy, but we wanted to make sure a reader is aware of the definition when reading through only a portion of our long review, such as for example only the fixed target device section. In addition, for each sample delivery method, we bring this ideal sample consumption in context to details of the

method, such as the need to jet for liquid injection or the formation of droplets for the droplet methods and their implication on sample consumption.

Related to the additional pdf document with reviewer 2 remarks, we believe we caught all comments and made changes accordingly in the manuscript. Several comments were repeated with the text version of the reviewer 1 comments and were already addressed in this response document.

8. Finally, the discussion section could be improved by writing it in the style of a guidebook for a new user. Serial crystallography experiments are complex, as already explained, so it would be useful for non-experts to have some guidance not only in terms of sample consumption (the topic of the review) but also regarding other challenges associated with executing these experiments. An attempt could be made to speculate on which method will consume the least amount of sample with the lowest “activation barrier” in terms of user expertise.

Response: We thank the reviewer for this comment. We have added Figure 6 to the conclusion section (as already mentioned above), which should help guide the interested reader. The added text changes should also further help with bringing the sample amounts in relation to researchers' needs. Our major motivation for using the ideal sample consumption as outlined in our review was to point out the bottleneck in SX techniques and the need for future studies in the field.

Reviewer #2 (Remarks to the Author):

General comments:

This review provides an good overview of serial crystallography over the past 10 yrs with a particular focus on sample consumption. Overall, it is written well and was a joy to read. It covers a many different delivery systems and assesses their sample volume and relates it back to the ‘ideal’ sample volume for crystallographers to generate a data set. Sample consumption is a hot and debatable topic in this field. While the authors have presented a standardized way to compare each sample delivery system, the challenge lies with what you consider is the ‘ideal sample volume’ and what is acceptable in the field. For example, to setup and grow large well diffracting crystals of an unknown protein for traditional crystallography may take 100s of crystallization screens. If you estimate a protein concentration of minimum 20mg/ml in a nano droplet (50nl) and set up a 1x96 well screen with duplicate wells. Once crystallization plate would require ~0.192 mg of protein for the one screen, if you multiple this by just 10 screens it now becomes 1.92 mg. It may take 100’s of screening condition to get a few large crystals . Therefore the amount of protein for optimizing a large good quality crystals maybe in the order of 5-10 mg for example. SX offers an opportunity to use shower of good quality crystals which you may hit in the first few screens and then go straight into SX, the amount of protein you saved in screen you can justify for SX data collection. I think term sample consumption is relative and one must make it clear in this review what it should be really compared to and justify what is

denoted as high sample consumption. This would be a valuable comment to make in this review.

Response: We thank the reviewer for their general positive assessment of our review article. We agree that defining the ideal sample consumption is a challenge on its own. We decided to go very aggressive with our assumptions and - as the reviewer points out – do not consider crystallization screens. We simply believe that crystallization itself is a very challenging topic dependent on the protein type and can possibly not be included in a discussion of sample consumption in an SX experiment. We assume that the crystallization conditions are found and researchers determined that SX approaches are feasible. We have therefore included a statement in the definition of our ideal sample consumption strategy in the last paragraph of the introduction and further points this out in the conclusion.

1. Fix-targets- Has there been consideration of crystal orientation? Is there an issue associated with preferential orientations of the crystal on a surface which can be noted. Does this lead to higher number of crystals required to collect a full data set? Do the authors see this problematic for testing of future protein targets (non-standards) and are there any efforts in the literature to help combat this? My point here is that Table 1 shows a nice summary of fixed-target structures where the protocols have worked well, and structures have been solved, however, ~65% of the target structures solved in the table are standard proteins used for crystallography which are easily crystallized and well characterized. For unknown crystal samples the complexity lies within the crystallization environment and crystal surface/shape/charge etc. So, what maybe straight forward for lysozyme maybe challenging for non-standard crystals grown. Any comment in this area the authors could make and can be guided by the literature would be valuable to the readers.

Response: We acknowledge the reviewer's concern. What we reviewed in our manuscript is the current status in the field. We can not provide more information on "non-standard" proteins, but believe that the advances in SX will allow to study a larger variety of proteins. By offering a review on sample consumption we hope to contribute to this future improvement in SX techniques for a wider class of proteins as well. To include this issue, we have added two sentences in the conclusion section summarizing fixed-target devices pointing out that there is a need for further studies with more challenging proteins.

2. - One of the first silicon chip filtration was tested by Alke Meents group publication not mentioned here, Roedig, I Vartiainen, R Duman, S Panneerselvam, N Stübe, O Lorbeer, M Warmer, G Sutton, DI Stuart, E Weckert, C David, A Wagner, A Meents, Scientific 2015, p10451 publication not cited

Response: We thank the reviewer for this comment and have added this reference in the fixed target section under A1. Silicon-based fixed-target methods. This reference has also been added to Table 1 under Silicon-based fixed-target devices.

3. HVE section- although the authors primary focus on HVE in the context of XFELs, synchrotron source are also important. One might argue more ideal for HVE. Syringe type injectors have less wastage compared to viscous injector and depending on crystal size (larger crystals), there is a possibility for multiple hits on the same crystals at a slow flow rate and micro focus beam line. A couple of papers which demonstrate this but not limited to are Botha S, et.al. Acta Crystallogr D Biol Crystallogr. 2015 Feb;71(Pt 2):387-97 and Hadian-Jazi, M.; et.al. Analysis of Multi-Hit Crystals in Serial Synchrotron Crystallography Experiments Using High-Viscosity Injectors. Crystals 2021, 11, 49. This is not mentioned in this review.

Response: We thank the reviewer for this comment. While our primary focus was on high-viscosity extrusion (HVE) in the context of XFELs, we agree that synchrotron sources play an important and complementary role. We have added the following note in the HVE section as well as the two suggested citations: “It is worth noting, however, that viscous injection systems are effective delivery methods at synchrotron sources under certain conditions—particularly when using larger crystals, slower flow rates, and micro-focused beamlines—which enable multiple hits on the same crystal reducing sample consumption.”

4. P29, para 2 line While the authors make the comment ‘handling 4 nL is not feasible. This limitation arises not only from the technical challenges of managing such a miniscule volume but also from the difficulty in maintaining consistent crystal dispersion within the stream of deposited solution.’ I tend to disagree with this statement. Isn’t the whole to aim is to lower sample consumption as much as possible? Having stock solutions of 4nl is a challenge for any application. Even nano drop crystallization protocols where minimum dispensing volumes for proteins are as low as 50 nl. Even this setup requires micro liter stock volumes of protein. The real advantage here is that we don’t need to use mls of sample and can realistically get away with ul stock volumes of crystals for this technique. Automated sample loading will help but the question I ask here is do when then have a limit on the minimum volume for SX as of all techniques there is a reasonable amount of wastage that occurs which is acceptable. This is something that the authors do mention later in the article but maybe worth mentioning here too.

Response: We thank the reviewer for highlighting this once more. To make it more evident in the Liquid Stream on Tape section, we added the following: “While handling volumes as low as 4 nL presents technical challenges, it is important to emphasize that one of the key advantages of serial techniques is the ability to work with microliter-scale crystal suspensions rather than requiring milliliter quantities, making them inherently more sample-efficient despite some acceptable wastage.”

5. P31, last paragraph ‘Continued advancements in chip design, sample delivery systems, and crystal handling techniques could bridge this gap, allowing for further reductions in sample consumption while maintaining data quality and throughput.’ Can’t this statement be said for all techniques that have been developed not just LAMA?

Response: We have changed this sentence to “Continued advancements in chip design integrated with optimizing lower droplet volumes could bridge this gap, allowing for further reductions in sample consumption while maintaining data quality and throughput.”

Specific comments:

6. P4, para 3, line 1: ‘This challenge of sample consumption along with well-timed advances in microfabrication and 3D printing techniques in the last decade led to a wildfire of new sample injection techniques allowing SX as well as approaches to conserving precious crystal samples.

- This sentence doesn’t make sense. I think there is a word missing "other"

Response: Thank you, the sentence has been updated with the missing word “other” as suggested by the reviewer.

7. P10, para 1, line 1 ‘Kapton is another widely used material for fixed-target systems owing to the low X-ray background scattering from the material.’ -References are need here. In the context of SFX some of the first experiments were conducted using Kapton fixed targets eg. Ryan RA et.al J Vis Exp. 2017 and more recently Bosman et al. Cell Reports Physical Science, 2024, 101987 provided a good comparison study not mentioned in this review.

Response: We thank the reviewer for this comment. The following reference is now included: Bosman et al. Cell Reports Physical Science 5, 101987, June 19, 2024 . In addition, we checked the article by Ryan et al. J Via Exp. 2017 which had used the Kapton based fixed-target device for structure determination of Buckminster fullerene which does not match with the application of protein X-ray crystallography in our review, thus opted to not cite it in our review.

8. P16 para 1 ‘A smaller sample volume with a higher crystal density would be required for the same full dataset since more pulses are delivered in a faster time: 2.0 µL of sample with a density of 5×10^6 crystals/mL.’- shouldn’t this be lower crystal density not higher if you are comparing it to 4×10^{10} ????

Response: We appreciate the reviewer’s question. The mention of a "higher crystal density" was a misstatement in this context and has been corrected to "lower crystal density" to accurately reflect the comparison with the previous ideal case.

9. Fig 2b) label ‘Enclosing polyimide film’ is cut off.

Response: We have made changes to this figure as described in responses to reviewer 1 comments above and this issue is fixed.

10. Fig3j. Labels in insert is too small and unclear.

Response: We have made changes to this figure and this issue is fixed.

11. Fig 4. Labeling of different components in figure would be helpful.

Response: We agree with this comment and have improved the Figure accordingly as mentioned above in responses to reviewer 1.

Response to Reviewers

Reviewer 1:

The reviewer appreciates the authors' rebuttal comments and acknowledges the overall improvement of the manuscript, particularly the enhanced quality of the figures. The reviewer specifically welcomes the inclusion of a new figure (Figure 6) and the revisions made to the Summary section.

With Figure 6 now incorporated into the main text, it may be appropriate to move the corresponding table to the supplementary information. The reviewer suggests that the authors and the editor discuss the best course of action to avoid making the manuscript unnecessarily lengthy. *[A comment about this suggestion was added to our cover letter to the editor.]*

A PDF containing minor editorial changes as well some open questions has been attached. The reviewer strongly recommends a thorough re-reading of the manuscript, as it contains numerous typographical and grammatical errors—particularly in the sections added during the rebuttal.

In the Summary section, the authors state: “However, such an ideal estimate is a good guideline to point out the current bottleneck in sample consumption in the SX field and to steer the field into future sample consumption efficiency improvements.” Does this imply a direction toward the application of fixed targets? While the authors thoroughly discuss the nuances of various delivery methods, they do not address their user-friendliness. It would be valuable to hear the authors' perspective on which methods are most suitable for non-experts. The reviewer is also inclined toward the use of fixed targets, while fully acknowledging their limitations. *[Our response to this point can be found below for the comment on page 33].*

The reviewer would recommend the manuscript for publication once the points in the attached PDF document are addressed.

Response: We thank the reviewer for the detailed review of our manuscript and the additional suggestions. We have made the suggested editorial changes and respond with additional comments made by the reviewer in the provided pdf document below:

Comment on page 4: “this should be TR-SX, correct?”

Response: We have corrected the term to “TR-SFX” as per the text.

Comment on page 14: “would it possible to provide a citation here (even if it is a self citation)?”

Response: We thank the reviewer for this suggestion. We found two citations appropriate for this statement from other groups and included one of our own works as well.

Comment on page 22: “from the experience of the reviewer, the same injector is often used for XFELs and synchrotron e.g. for LCP-grown crystals one often uses the same 100um jet diameter which is exposed to X-ray beam. The crystals size in such case is often very hard to optimize. Maybe you can comment on this?”

Regarding slower flow rates and micro-focus beam, the reviewers agrees fully.”

Response: We thank the reviewer for pointing out his/her agreement with our statement of flow rates. For the first comment, we are not sure what exactly the reviewer asks us to comment about. In our opinion, optimizing crystal size is a general problem that applies to all injection techniques, and we have therefore opted to not change the text on page 22. However, we have now amended a sentence in the summary section on page 33 (last two lines) referring to crystal size as a means of optimizing sample consumption.

Comment #1 on page 23: “What are the small elongated features within the droplet?”

Response: These elongated features are needle like crystals. We have changed the figure caption to explain the features.

Comment #2 on page 23: “a scale bar on the liquid droplet panel would be very helpful”

Response: Unfortunately, the manuscript we cite here, does not provide information that allows us to include a scale bar. We have thus opted to not include one.

Comment on page 30: “maybe a term "mixing" would be more appropriate here?”

Response: We have changed this sentence to: “Tape drive systems are also capable of time-resolved studies by synchronization of liquid deposition, subsequent mixing, and laser excitation with controlled crystal transportation on the tape.^{187, 201, 202” and added an additional reference at the end of the sentence.}

Comment on page 31: “should this be ref 203?”

Response: The citation at the end of the sentence was a review so we have moved the citation at the end of first sentence of this section, to avoid confusion.

Comment on page 33 (including reviewer 1’s comment above about this paragraph): “what would be that direction? Would the authors suggest using fixed-targets?”

Response: In this sentence, we generally point out that improvements in sample consumption efficiency are required, summarizing the vast information provided in our review. We don’t believe that a statement on a preference for fixed-target devices in this paragraph would be useful to the reader, as we are still establishing the argument about the complexity of sample consumption reduction in this paragraph. Further below in the summary section, we discuss fixed-target approaches and their limitations pointing out suggested improvements for the future. We have also discussed and summarized the other sample injection methods outlined in this review in a similar manner. We believe that our current summary does justice to describe a more complex situation, as fixed-target devices (while generally reported to use the least sample amount) are not the solution for each SFX or SMX experiment.

Comment on page 34: “Maybe worth adding how many publications were examined/checked in order to make this plot. Showing how many data points there are would be useful.”

Response: We thank the reviewer for this suggestion and have added how many references were used to establish this figure.

Reviewer 2:

We thank the reviewer for his/her assessment of our revised version and are pleased that this reviewer agrees with publishing our work.

[revised manuscript text omitted]

**Figure 6:** Overview summary of the sample consumption by different injection methods discussed in this
review and as reported in Tables 1 and 3-5.

As outlined in this review, the fixed-target sample delivery method, a cornerstone of
crystallography since its inception, has undergone considerable innovation in terms of design and
materials, particularly in response to improvements in X-ray sources. However, reducing sample
consumption has not been a major focus in the development of fixed-target systems, which
typically require sample volumes of $\leq 10 \mu\text{L}$ and 1 mg or less protein amounts, which are much
smaller – sometimes orders of magnitude - than the liquid injection methods as shown above in
Figure 6. Most of the developed fixed-target devices have employed model proteins as proof of
concept. Thus, for future device development, it will be important to demonstrate the capability
of using non-standard proteins along with model proteins. Furthermore, one key advantage of
fixed-target methods is the ability to grow crystals on-chip, enabling direct delivery to the X-ray
beam for data collection. A notable limitation of the fixed-target approach, however, is its limited
applicability to TR-SX, where liquid sample delivery has traditionally been the preferred method
for non-light-driven reactions. The future effort thus needs to be directed to the development of
fixed-target devices allowing mixing experiments prior to X-ray exposure. This could be achieved
by the integration of miniaturized elements to allow mixing including precise positioning of

crystals with respect to the corresponding solutions to be mixed. Another future improvement
in fixed-target approaches could originate from automated sample delivery, reducing the 'dead
volume' introduced through manual pipetting steps currently required in most fixed-target
device applications.

Liquid delivery systems are more commonly used in XFEL experiments, particularly for pump-
probe and mix-and-inject experiments that are designed to probe protein dynamics in TR-SX.
While liquid delivery offers clear advantages for time-resolved studies, it has been hindered by
significant challenges related to sample consumption where the amount of protein needed for a
full data set is multiplied **by** the number of time points probed requiring up to gram amounts of
protein. In response, considerable efforts have been made to reduce the amount of precious
protein required for these experiments, leading to the development of more efficient systems.
Innovations such as synchronized droplet generators and HVEs have successfully minimized
sample consumption compared to earlier methods like GDVNs, as detailed in this review **and**
**evident in Figure 6**. Recent advancements have sought to combine the strengths of both fixed-
target and liquid delivery methods, opening the possibility of achieving ideal sample
consumption. One notable example is the LAMA system, which generates picoliter-sized droplets
and delivers them to the X-ray beam via a fixed-target system. However, it is important to note
that sample consumption is influenced not only by the delivery method but also by the
parameters of the X-ray source itself. As X-ray sources vary widely in their characteristics,
designing a universal delivery system that performs optimally across all platforms remains a
significant challenge. The process of optimizing these systems requires time and extensive
iteration, which is often difficult to achieve given the rapidly evolving needs of X-ray facilities.

Furthermore, the limited time availability at these facilities makes the optimization process
even more challenging for experimenters. A promising solution to this challenge may come with
the development of compact X-ray light sources (CXLS) and compact X-ray free-electron lasers

[revised manuscript text omitted]
 (4) dioxxygenase	(1) 24 (3) 10 (4) 5-7	1-3: 10 ¹¹ 4: 10 ⁸ -10 ⁹	(1) 0.8 μm (2) 0.4x0.2x0.2 μm (3) 5-10 μm (4)	(1) 450μL*/ 10.8mg (3) 765μL*/ 7.65mg (4) 344μL*/ 2.41mg	Glass DFFN

Doppler et al. ¹⁴⁶	PP	EuXFEL (10Hz, 125 pulses/train)	SPB/SFX (vacuum)	Q _r : 10-1 Q _{exp} :5	PSII	NI	NI	1-2×1-2×10-30 μm 10-30 μm	NI	3D-printed Co-flow
Jernigan et al. ¹⁴⁹	N	LCLS (120)	MFX (GDVN-He, coMESH-air)	Q _{exp} : 3-5 (coMESH) 25 (GDVN)	NendoU	40-80	NI	2x2x2 μm, 10-15 μm	NI	CoMESH & GDVN-misc
Electrospinning Injectors										
Kern et al. ²¹⁶	PP	LCLS (120)	CXI (vacuum)	Q _{exp} :2.5-3.1	PSII	7.4	NI	5-10 μm	NI	MESH
Sierra et al. ¹⁴⁸	N	LCLS (120)	CXI (vacuum)	Q _r : 0.14-3.1 Q _{exp} : 0.17	Thermolysin	NI	2x10 ¹⁰	2.5x4-3.5x7 μm	NI/0.14mg	MESH
Sierra et al. ¹³⁰	misc	LCLS (120)	CXI (vacuum)	Q _{exp} : 0.75-3	(1) PSII (2) paromomycin complex (3) Thermolysin	(1)70, (3) 25	2: 10 ¹⁰ -10 ¹¹	(1) 5-15 μm (2) 3-10x3-10x20-30 μm (3) 3x5 μm	(2) 360μL/NI	CoMESH
Dao et al. ¹⁵³	misc	LCLS (120)	CXI (vacuum)	NI	Thermus thermophilus	NI	10 ¹⁰ -10 ¹¹	2x2x4 μm	500μL/NI	CoMESH
Ciftci et al. ¹⁵⁴	misc	LCLS (120)	CXI (vacuum)	Q _{exp} :1-3	HIV-1	NI	10 ¹⁰ -10 ¹¹	1x1x5 μm to 5x5x15 μm	500μL/NI	CoMESH
Ishigami et al. ¹⁵²	N	LCLS (30)	MFX (He)	Q _{exp} :3	Cytochrome c oxidase	80–90	10 ^{10b}	20x20x4 μm	NI	MESH
Viscous Injection										
Weierstall et al. ¹⁵⁶	N	LCLS (60-120)	CXI (vacuum)	Q _r : 0.001-0.3 Q _{exp} :0.17	(1) β2 adrenergic receptor (2) A _{2A} AR (3) glucagon receptor	(1-3) 20-50 (6) 12	NI	(5) >5 μm	<100μL/ <0.5mg	LCP Injector with LCP

					(4) 5-HT _{2B} (5) SMO (6) DgkA					
Liu et al. ¹⁵⁷	N	LCLS (120)	CXI (vacuum)	Q _r : 0.003- 0.3	5-HT _{2B}	NI	NI	5x5x5 μm	NI/0.3mg	LCP Injector with LCP
Sugahara et al. ¹⁶¹	N	SACLA (30)	BL3 - EH4 (He)	Q _r : 0.12- 0.48 Q _{exp} : (1,2,4) 0.48 (3) 0.46	(1) HEWL (2) glucose isomerase (3) thaumatin (4) fatty acid- binding protein type 3	NI	(1) 6x10 ⁷ (2) 2x10 ⁷ (3) 1x10 ⁷ (4) 0.9x10 ⁷	(1) 7-10 μm (2) 10-30 μm (3) 10-30 μm (4) 10-20 μm 10x10x30 μm	(1) NI/2.4mg, (2) NI/0.7mg (3) NI/0.28mg (4) NI/0.15mg	Needle Extrusion with mineral oil-grease media
Botha et al. ¹⁶⁷	N	SLS (10)	PXII (He‡)	Q _{exp} : 0.021	HEWL	30-60	NI	15x15x60 μm	NI	HVE with LCP and Vaseline media
Conrad et al. ¹⁶⁴	N	LCLS (120‡)	CXI (vacuum, He)	Q _{exp} : 1.6	(1) Phycocyanin (2) PS I (3) PS II	(1) 15	(1-3) 2x10 ¹⁰	(1) 1-5 μm	(1) 0.02μL* / 0.3mg	LCP Injector with Agarose media Needle extruder with
Sugahara et al. ⁵⁵	N	SACLA (30)	BL3 - EH4 (He)	Q _{exp} : 0.48	(1) Proteinase K (2) HEWL	40	(1,2) 6.7x10 ⁷	(1,2) 5-10 μm	30μL/1.2mg*	Super Lube grease and hyaluronic acid media
Kovacsova et al. ¹⁶⁵	N	SLS (10)	PXII (He‡)	Q _r : (1,2) 0.06- 0.15 (3) 0.3-0.15 (4) 0.09-0.15	(1) Lyophilized thermolysin (2) Glucose isomerase (3) HEWL (4) Bacteriorhodopsin	(1) 25 (2) 80 (3) 30-60 (4) 35-50	NI	(1) 50- 130x5- 10x5-10 μm (2) 10- 15x10- 15x10-15 μm (3) 30x20x20 μm (4) 20x50x2 μm	(1) 20μL* / 0.5mg (2) 6μL* / 0.5mg (3) 8μL* / 0.5mg (4) 10μL* / 0.5mg	HVE with sodium carboxy- methyl cellulose and Pluronic media

Sugahara et al. ¹⁶⁹	N	SACLA (30)	BL3 - EH4 (He)	Q _{exp} : (1) 0.42-0.75 (2) 0.47 (3) 0.38-0.47	(1) HEWL (2) thaumatin (3) proteinase K	(1) 20 (2,3) 40	(1) 1.7x10 ⁷ - 5.8x10 ⁸ (2) 4.3x10 ⁸ (3) 4.9-9.3x10 ⁷	(1) 1x1x1 μm, 5x5x5 μm, 20x20x30 μm (2) 2x2x4 μm (3) 4x4x4 - 5x5x7 μm, 8x8x8 - 12x12x12 μm	(1) 105μL*/2.1mg (2) 12.5μL*/0.5mg (3) 35.8μL*/1.4mg	HVC injector with nuclear grease
Shimazu et al. ¹⁷⁰	N	SACLA (30)	BL3 - EH4 (He)	Q _r : 0.1-5.6 Q _{exp} : (1) 0.24 (2) 0.42	(1) A _{2A} AR (2) HEWL	(1) 50 (2) 20	(2) 2.3x10 ⁸	(1) 20x3x3 μm (2) ~5 μm	(1) 30μL/1.5mg* (2) 40μL/0.8mg*	HVC injector with LCP and nuclear grade grease HVC injector with (i) Paraffin grease (ii) DATPE grease (iii) Nuclear grease (iv) Cellulose (v) Frozen Paraffin grease (vi) Paraffin grease (native) (vii) DATPE grease (pr-derivative) 3D-printed injector with LCP media HVC with hydroxyethyl
Sugahara et al. ¹⁶²	N	SACLA (30)	BL2 (He)	Q _{exp} : 0.24, 0.11	proteinase K		(i,v) 9.4x10 ⁷ (ii, iii, iv) 1.4x10 ⁸ (vi, vii) 7.2x10 ⁸	(i-v) 2x2 μm (vi-vii) 0.8x0.8 μm	(i-v) 20μL/0.8mg* (vi-vii) 8μL/0.32mg*	
Vakili et al. ²¹⁷	MISC	EuXFEL (10Hz, 1 pulse/ train)	SPB/SFX (vacuum)	Q _r : 0.11-0.36	iq-mEmerald protein mixed with CuCl ₂	50	NI	5x15 μm	NI	
Wolff et al. ¹⁶⁶	PP	SACLA (30)	BL2 (He†)	Q _{exp} : 2.5	HEWL	20	NI	NI	NI	

cellulose
media

**Table 4:** Overview of droplet injection systems in the literature including details on X-ray source, flow rates, analyte, protein and
 particle concentration, crystal size, and reported sample consumption. Sample consumption is recorded in volume and protein
 amount in milligrams. Time-resolved experiments in the second column are classified into pump-probe (PP) for light-induced, mix and
 inject (misc), and N for not time-resolved. ‡The conditions in which the experiment was conducted are not explicitly stated and the
 recorded condition is assumed based on the X-ray instrument and end-station used. Q_{exp} : The flow rates listed were used for data
 collection. Q_r : The flow rates recorded were given in a range. *The sample consumption value was estimated from the reported
 protein concentration in the mother liquor considering that all protein molecules were converted to crystals considering reported
 volumes consumed or amount of protein consumed. NI refers to data not included in the publication. N/A refers to data not
 applicable to the experimental type reported in the manuscript. The values listed in the table (symbol a) are derived from a citation
 within the manuscript providing the crystallization conditions.²⁷

Authors	TR* (pp, misc , N)	X-ray Source (freq)	End-station (environme nt)	Flow Rate ($\mu\text{L}/\text{min}$)	Analyte(s)	Protein Conc. (mg/mL)	Particle Conc. (part/mL)	Crystal Size	Sample Consumption (volume [μL]/ Amount [mg])	Injector Comment/Dro plet Size
Echelmeier et al. ¹⁷²	N	LCLS (120†)	CXI (vacuum)	Q_{exp} : 5.5	(1)Fluorescein droplets (2) granulovirus	NI	NI	NI	NI	glass GDVN + PDMS segmented drop/NI
Mafune et al. ¹⁴⁷	N	SACLA (30)	BL3 - EH4 (He)	Q_{exp} : 0.5	HEWL	NI	3.2×10^7 to 3.2×10^8	5	NI/0.3mg	Piezo-driven/ 0.268 nL
Roessler et al. ¹⁷¹	N	LCLS (60)	XPP (air)	N/A	(1) HEWL, (2) thermolysin, (3) stachydrine demethylase, (4) MauG mixed with MADH (5) hemoglobin (6) PSII	(1) 70 (2) 30 (5) 18 (6) 20-40	NI	(1) 5-10 μm (2) 10-100 μm (3) 25-50 μm , 50-100 μm , 200-300 μm (4) 10-50 μm (5) 50-250 μm (6) 50-100 μm , 150-400 μm	NI	ADE and levitation/ 0.1-2.5 nL

Awel et al. ¹⁸⁰	N	LCLS (120±)	CXI (vacuum)	Q _{exp} : 2.7-3.5	granulovirus	N/A	3x10 ¹¹	0.2 x 0.2 x 0.37 μm	NI	Ceramic aerosol/0.03 pL
Echelmeier et al. ¹⁷³	N	LCLS (120)	MFX (He)	Q _r : 0.5-20	PSI	1 ^a	10 ^{9 a}	0.2-1 μm	NI	Segmented Drop/ <1 nL
Echelmeier et al. ¹⁷⁴	N	EuXFEL (10Hz, 15 pulses/train)	SPB/SFX (vacuum)	Q _r : 3-20	KDO8PS	21	5x10 ⁹	8-10 μm	962μL/20mg*	Segmented Drop/70-800 pL
Sonker et al. ⁷⁰	N	LCLS (120)	MFX (He)	Q _{exp} : 4.1 - 5.1	(1) KDO8PS (2) HEWL	(1) 20 (2) 126	(1) 2x10 ⁴	(2) 5-10 μm	NI	Segmented Drop/NI
Doppler et al. ¹⁷⁶	N	LCLS (120)	MFX (He)	Q _{exp} : 3-4	(1) NQO1 (2) Phycocyanin	(1) 25 (2) 50	NI	(1) 10x2x2 μm (2) 5-15 μm	NI	Segmented Drop/2.3 nL
Perrett et al. ⁶⁸	N	EuXFEL (10Hz, 16 pulses/train)	FXE (He)	NI	HEWL	NI	3-6x10 ⁷	NI	NI	DoD/NI
Doppler et al. ⁸⁰	MISC	EuXFEL (10Hz, 300 pulses/train)	SPB/SFX (vacuum)	Q _{exp} : 0.5 - 4.9	NQO1+NADH	18 to 26.5	2x10 ⁷	5 x 40 μm	228μL/4.9mg	Segmented Drop/2.5 nL

**Table 5:** Overview of Hybrid Sample Delivery Methods in the literature including details on X-ray source, flow rates, tape speeds,
 analyte, protein and particle concentration, crystal size, and reported sample consumption. Sample consumption is recorded in
 volume and protein amount in milligrams. Time-resolved experiments in the second column are classified into pump-probe (PP) for
 light-induced, mix and inject (misc), oxygen evolution reactions (O₂), and N for not time-resolved. ‡The conditions in which the
 experiment was conducted are not explicitly stated and the recorded condition is assumed based on the X-ray instrument and end-
 station used. Q_{exp}: The flow rates listed were used for data collection. Q_r: The flow rates recorded were given in a range. *The sample
 consumption value was estimated from the reported protein concentration in the mother liquor considering that all protein molecules
 were converted to crystals considering reported volumes consumed or amount of protein consumed. NI refers to data not included in
 the manuscript. N/A refers to data not applicable to the experimental type reported in the manuscript.

Authors	TR* (pp, mix, O ₂ , N)	X-ray Source (freq)	End- station (environm ent)	Flow Rate (μ L/min)	Analyte(s)	Protein Conc. (mg/mL)	Particle Conc. (part/mL)	Crystal Size	Sample Consumption (volume [μ L]/ Amount [mg])	Tape Speed (mm/s)
Tape-Drive										
Beyerlein et al. ⁷¹	MIX	PETRA III (25)	P11 (He‡)	Q _{exp} = 0.6	HEWL	126	NI	6-8 μ m	300 μ L/18.9mg	0.6
Zielinski et al. ¹⁸⁸	N	PETRA III (25)	P11 (He‡)	Q _{exp} = (1) 2 (2,3) 1	(1) β -lactamase (2) GH11 xylanase (3) Urate oxidase	(1) 22 (2) 15 (3) 20	NI	(1) 11-15 μ m (2) 10-20 μ m (3) 3-20 μ m, 400x400x300 μ m	(1) 91.4 μ L/ 1.76mg*	1
Lee et al. ²⁰¹	N	PAL (30)	NCI (He‡)	Q _{exp} = 0.05–0.1 Q _r = 0.05 - 10	HEWL	50	NI	10x10x10 μ m	NI	1.5
Henkel et al. ²⁰³	MIX	PETRA III (130)	P11 (He‡)	1	HEWL	126	NI	38x38 μ m	NI	1

Drop-on-Tape

Young et al. ¹⁹⁰	PP	LCLS (10, 120)	CXI (vacuum‡) XPP (He‡) MFX (He‡)	NI	Photosystem II	NI	NI	20-50 μm	NI	NI
Fuller et al. ¹⁸⁴	PP/O ₂	LCLS (10, 30, 60)	XPP (He) MFX (He)	N/A	(1) Photosystem II (2) Phytochrome (PSM) (3) Phytochrome (PAS-GAF) (4) RNR	(1) NI (2) 40 (3) 40 (4) NI	NI	(1) 20-50 μm (2) 100μm (3) 50μm (4) 20-30μm	(1) NI/2.4mg (2) NI/6.1mg (3) NI/2.9mg (4) NI/6.9mg	NI
Kern et al. ¹⁵¹	PP	LCLS (10)	MFX (He‡)	NI	Photosystem II	NI	NI	20-60μm	NI	NI
Burgie et al. ²¹⁸	N	LCLS (10)	MFX (He)	Q _{exp} = 2.5-3	TePixJ(GAF)	25	1.2x10 ⁷	NI	NI	300
Ibrahim et al. ²¹⁹	PP	LCLS (20) SACLA (30)	MFX (He‡) BL2-EH3 (He‡)	NI	Photosystem II	40-50	NI	20-60μm	NI	NI
Srinivas et al. ¹⁹⁷	O ₂	SACLA (30) LCLS (20) PAL (15)	NI (NI) MFX (He) NCI (He)	Q _{exp} (grease) = 20-80 Q _{exp} (DOT) = 8	sMMOH:MMOB	NI	NI	20-30μm	NI	300
Rabe et al. ¹⁹⁶	O ₂	LCLS (30) SACLA (30)	MFX (He) BL2 (He‡)	Q _{exp} (ADE- DOT) = 7 (v-ext) = 1-1.5	isopenicillin N synthase	50-52	5 × 10 ⁷	3x3x40-60μm	NI	50, 93.75, 187.5, 300, 375
Ohmer et al. ¹⁹⁴	N	LCLS (30‡)	MFX (He‡)	NI	MCRred1-silent	40	NI	40-80μm	NI	NI
Hussein et al. ²²⁰	PP	LCLS (20)	MFX (He‡)	NI	Photosystem II	NI	NI	20-60μm	NI	NI

Bhowmick et al. ²²¹	PP	LCLS (10,20) SALCA (20)	MFX (He†) BL2 (He†)	NI	Photosystem II	NI	NI	20-60µm	NI	NI
Matika et al. ²²²	PP, O ₂	LCLS (30)	MFX (He†)	Q _r = 7-9	(1) Photosystem II, (2) isopenicillin N synthase	2) 50-52	(1) 2x10 ⁶ (2) 2x10 ⁷	(1) 20-50µm (2) 40-60µm	NI	200-300
Lebrette et al. ¹⁹⁵	N	LCLS (30)	MFX (He†)	Q _{exp} = 9	Class Ie R2 Protein Radical	13	NI	10x10x5 µm	316.8µL*/ 4.1mg*	300
Butryn et al. ¹⁸⁷	MIX	SACLA (30)	BL2 (He†)	N/A	(1) HEWL mixed with GlcNAc (2) CTX-M-15 mixed with ertapenem	(1) 50 (2) 20	(1) ~10 ⁷ (2) 8x10 ⁷	(1) 3-5µm (2) 5x10-20µm	(1) 932µL/ 42.6mg (2) 995µL/8mg	(1) 300, 100, 30 (2) 100, 30
Nguyen et al. ¹⁹⁹	MIX	LCLS (30)	MFX† (He†)	Q _{exp} = 3.6	CYP121	10-14	NI	30µm	NI	200
Other Hybrid Methods										
Mathew et al. ²⁰⁶	N	LCLS (10)	XPP	N/A	Polyketide synthases	NI	NI	50x10x2 µm	300µL/3.3mg	2.5
Mehrabi et al. ¹⁹³	MIX	PETRA III (25†)	P14 (air†) P14-2 (He†)	NI	(1) xylose isomerase (2) HEWL mixed with GlcNAc	(1) 80 (2) 67	NI	NI	NI	NI

**Reference:**

1. Shi Y. A Glimpse of Structural Biology through X-Ray Crystallography. *Cell* **159**, 995-1014
(2014).

2. Authier A. *Early days of X-ray crystallography*. OUP Oxford (2013).

3. Botha S, Fromme P. Review of serial femtosecond crystallography including the COVID-
19 pandemic impact and future outlook. *Structure* **31**, 1306-1319 (2023).

4. Echelmeier A, Sonker M, Ros A. Microfluidic sample delivery for serial crystallography
using XFELs. *Anal Bioanal Chem* **411**, 6535-6547 (2019).

5. Hejazian M, Balaur E, Abbey B. Recent Advances and Future Perspectives on Microfluidic
Mix-and-Jet Sample Delivery Devices. *Micromachines* **12**, 531 (2021).

6. Drenth J, Haas C. Protein crystals and their stability. *Journal of Crystal Growth* **122**, 107-
109 (1992).

7. Bilderback DH, Hoffman SA, Thiel DJ. Nanometer Spatial Resolution Achieved in Hard X-
Ray Imaging and Laue Diffraction Experiments. *Science* **263**, 201-203 (1994).

8. Stevens RC. High-throughput protein crystallization. *Current Opinion in Structural*
*Biology* **10**, 558-563 (2000).

9. Bugg CE. The future of protein crystal growth. *Journal of Crystal Growth* **76**, 535-544
(1986).

10. Weber PC. [2] Overview of protein crystallization methods. In: *Methods in Enzymology*.
Academic Press (1997).

11. Smith JL, Fischetti RF, Yamamoto M. Micro-crystallography comes of age. *Current*
*Opinion in Structural Biology* **22**, 602-612 (2012).

12. Riekel C. Recent developments in microdiffraction on protein crystals. *Journal of*
*synchrotron radiation* **11**, 4-6 (2004).

13. Evans G, Alianelli L, Burt M, Wagner A, Sawhney KJS. Diamond Beamline 124: A Flexible
Instrument for Macromolecular Micro-crystallography. *AIP Conference Proceedings* **879**,
836-839 (2007).

14. Fischetti RF, *et al.* Mini-beam collimator enables microcrystallography experiments on
standard beamlines. *Journal of synchrotron radiation* **16**, 217-225 (2009).

- 15. Moukhametzianov R, *et al.* Protein crystallography with a micrometre-sized
synchrotron-radiation beam. *Acta Crystallographica Section D: Biological*
*Crystallography* **64**, 158-166 (2008).
- 16. Neutze R, Wouts R, van der Spoel D, Weckert E, Hajdu J. Potential for biomolecular
imaging with femtosecond X-ray pulses. *Nature* **406**, 752-757 (2000).
- 17. Saldin E, Schneidmiller E, Yurkov MV. *The physics of free electron lasers*. Springer
Science & Business Media (1999).
- 18. Fromme P. XFELs open a new era in structural chemical biology. *Nature Chemical*
*Biology* **11**, 895-899 (2015).
- 19. Cheng RK. Towards an Optimal Sample Delivery Method for Serial Crystallography at
XFEL. *Crystals* **10**, 215 (2020).
- 20. Georgescu I. The first decade of XFELs. *Nature Reviews Physics* **2**, 345-345 (2020).
- 21. Chapman HN. X-Ray Free-Electron Lasers for the Structure and Dynamics of
Macromolecules. *Annual Review of Biochemistry* **88**, 35-58 (2019).
- 22. Liu H, Lee W. The XFEL Protein Crystallography: Developments and Perspectives.
*International Journal of Molecular Sciences* **20**, 3421 (2019).
- 23. Gavira JA. Current trends in protein crystallization. *Archives of Biochemistry and*
*Biophysics* **602**, 3-11 (2016).
- 24. Coleman C, *et al.* Modeling of XFEL induced ionization and atomic displacement in
protein nanocrystals. In: *X-Ray Free-Electron Lasers: Beam Diagnostics, Beamline*
*Instrumentation, and Applications*. Proc. SPIE (2012).
- 25. Obara Y, *et al.* Femtosecond time-resolved X-ray absorption spectroscopy of anatase
TiO₂ nanoparticles using XFEL. *Structural Dynamics* **4**, 044033 (2017).
- 26. Chapman HN, *et al.* Femtosecond X-ray protein nanocrystallography. *Nature* **470**, 73-77
(2011).
- 27. Hunter MS, Fromme P. Toward structure determination using membrane-protein
nanocrystals and microcrystals. *Methods* **55**, 387-404 (2011).
- 28. Juncheng E, *et al.* Expected resolution limits of x-ray free-electron laser single-particle
imaging for realistic source and detector properties. *Structural Dynamics* **9**, 064101
(2022).

- 29. Ayyer K, *et al.* 3D diffractive imaging of nanoparticle ensembles using an x-ray laser. *Optica* **8**, 15-23 (2021).
- 30. Chapman HN. Fourth-generation light sources. *IUCrJ* **10**, 246-247 (2023).
- 31. Altarelli M. From 3rd- to 4th-generation light sources: Free-electron lasers in the X-ray
range. *Crystallography Reports* **55**, 1145-1151 (2010).
- 32. Johansson LC, Stauch B, Ishchenko A, Cherezov V. A Bright Future for Serial
Femtosecond Crystallography with XFELs. *Trends in Biochemical Sciences* **42**, 749-762
(2017).
- 33. Sackmann EK, Fulton AL, Beebe DJ. The present and future role of microfluidics in
biomedical research. *Nature* **507**, 181-189 (2014).
- 34. Labeed FH, Fatoyinbo HO. *Microfluidics in detection science: lab-on-a-chip technologies*.
Royal Society of Chemistry (2014).
- 35. Kleinstreuer C. *Microfluidics and nanofluidics: theory and selected applications*. John
Wiley & Sons (2013).
- 36. Seemann R, Brinkmann M, Pfohl T, Herminghaus S. Droplet based microfluidics. *Reports*
*on progress in physics* **75**, 016601 (2011).
- 37. Whitesides GM. The origins and the future of microfluidics. *Nature* **442**, 368-373 (2006).
- 38. McPherson A, Gavira JA. Introduction to protein crystallization. *Acta Crystallographica*
*Section F* **70**, 2-20 (2014).
- 39. Roedig P, *et al.* High-speed fixed-target serial virus crystallography. *Nature Methods* **14**,
805-810 (2017).
- 40. Schlichting I, Miao J. Emerging opportunities in structural biology with X-ray free-
electron lasers. *Current Opinion in Structural Biology* **22**, 613-626 (2012).
- 41. Hunter MS, *et al.* Fixed-target protein serial microcrystallography with an x-ray free
electron laser. *Scientific Reports* **4**, 6026 (2014).
- 42. Schlichting I. Serial femtosecond crystallography: the first five years. *IUCrJ* **2**, 246-255
(2015).
- 43. Emma P, *et al.* First lasing and operation of an ångstrom-wavelength free-electron laser.
*Nature Photonics* **4**, 641-647 (2010).

- 44. Boutet S, *et al.* High-resolution protein structure determination by serial femtosecond
crystallography. *Science* **337**, 362-364 (2012).
- 45. Boutet S, J Williams G. The Coherent X-ray Imaging (CXI) instrument at the Linac
Coherent Light Source (LCLS). *New Journal of Physics* **12**, 035024 (2010).
- 46. Stauch B, Cherezov V. Serial Femtosecond Crystallography of G Protein-Coupled
Receptors. *Annual Review of Biophysics* **47**, 377-397 (2018).
- 47. Ishchenko A, Gati C, Cherezov V. Structural biology of G protein-coupled receptors: new
opportunities from XFELs and cryoEM. *Current Opinion in Structural Biology* **51**, 44-52
(2018).
- 48. Pandey S, *et al.* Time-resolved serial femtosecond crystallography at the European XFEL.
*Nature Methods* **17**, 73-78 (2020).
- 49. Lee J-H, Zatsepin NA, Kim KH. Time-resolved serial femtosecond X-ray crystallography.
*BioDesign* **6**, 15-22 (2018).
- 50. Spence J. XFELs for structure and dynamics in biology. *IUCr* **4**, 322-339 (2017).
- 51. Aquila A, *et al.* Time-resolved protein nanocrystallography using an X-ray free-electron
laser. *Optics Express* **20**, 2706-2716 (2012).
- 52. Orville AM. Recent results in time resolved serial femtosecond crystallography at XFELs.
*Current Opinion in Structural Biology* **65**, 193-208 (2020).
- 53. Kupitz C, *et al.* Serial time-resolved crystallography of photosystem II using a
femtosecond X-ray laser. *Nature* **513**, 261-265 (2014).
- 54. Kern J, *et al.* Taking snapshots of photosynthetic water oxidation using femtosecond X-
ray diffraction and spectroscopy. *Nature Communications* **5**, 4371 (2014).
- 55. Nango E, *et al.* A three-dimensional movie of structural changes in bacteriorhodopsin.
*Science* **354**, 1552-1557 (2016).
- 56. Panneels V, *et al.* Time-resolved structural studies with serial crystallography: a new
light on retinal proteins. *Structural Dynamics* **2**, 041718 (2015).
- 57. Kang Y, *et al.* Crystal structure of rhodopsin bound to arrestin by femtosecond X-ray
laser. *Nature* **523**, 561-567 (2015).
- 58. Pande K, *et al.* Femtosecond structural dynamics drives the trans/cis isomerization in
photoactive yellow protein. *Science* **352**, 725-729 (2016).

59. Pandey S, Poudyal I, Malla TN. Pump-Probe Time-Resolved Serial Femtosecond
Crystallography at X-Ray Free Electron Lasers. *Crystals* **10**, 628 (2020).
60. Olmos JL, Jr., *et al.* Enzyme intermediates captured "on the fly" by mix-and-inject serial
crystallography. *BMC Biol* **16**, 59 (2018).
61. Stagno JR, *et al.* Structures of riboswitch RNA reaction states by mix-and-inject XFEL
serial crystallography. *Nature* **541**, 242-246 (2017).
62. Grieco A, Quereda-Moraleda I, Martin-Garcia JM. Innovative Strategies in X-ray
Crystallography for Exploring Structural Dynamics and Reaction Mechanisms in
Metabolic Disorders. *Journal of Personalized Medicine* **14**, 909 (2024).
63. Dasgupta M, *et al.* Mix-and-inject XFEL crystallography reveals gated conformational
dynamics during enzyme catalysis. *Proceedings of the National Academy of Sciences*
**116**, 25634-25640 (2019).
64. Zielinski KA, Pollack L. Advances in microfluidic mixers for time-resolved structural
biology with X-rays. *Biophysical Reviews*, (2025).
65. Weinert T, *et al.* Serial millisecond crystallography for routine room-temperature
structure determination at synchrotrons. *Nature Communications* **8**, 542 (2017).
66. Martin-Garcia JM, *et al.* Serial millisecond crystallography of membrane and soluble
protein microcrystals using synchrotron radiation. *IUCrJ* **4**, 439-454 (2017).
67. Nogly P, *et al.* Lipidic cubic phase serial millisecond crystallography using synchrotron
radiation. *IUCrJ* **2**, 168-176 (2015).
68. Perrett S, *et al.* Kilohertz droplet-on-demand serial femtosecond crystallography at the
European XFEL station FXE. *Structural Dynamics* **11**, 024310 (2024).
69. Holmes S, *et al.* Megahertz pulse trains enable multi-hit serial femtosecond
crystallography experiments at X-ray free electron lasers. *Nature Communications* **13**,
4708 (2022).
70. Sonker M, *et al.* Electrically stimulated droplet injector for reduced sample consumption
in serial crystallography. *Biophysical Reports* **2**, 100081 (2022).
71. Beyerlein KR, *et al.* Mix-and-diffuse serial synchrotron crystallography. *IUCrJ* **4**, 769-777
(2017).

- 72. Manna A, *et al.* Cyclic Olefin Copolymer-Based Fixed-Target Sample Delivery Device for
Protein X-ray Crystallography. *Analytical Chemistry* **96**, 20371-20381 (2024).
- 73. Birch J, *et al.* The fine art of integral membrane protein crystallisation. *Methods* **147**,
150-162 (2018).
- 74. Shoeman RL, Hartmann E, Schlichting I. Growing and making nano- and microcrystals.
*Nature Protocols* **18**, 854-882 (2023).
- 75. Martiel I, Muller-Werkmeister HM, Cohen AE. Strategies for sample delivery for
femtosecond crystallography. *Acta Crystallographica Section D* **75**, 160-177 (2019).
- 76. Vakili M, *et al.* 3D printed devices and infrastructure for liquid sample delivery at the
European XFEL. *Journal of synchrotron radiation* **29**, 331-346 (2022).
- 77. Nielsen AV, Beauchamp MJ, Nordin GP, Woolley AT. 3D Printed Microfluidics. *Annual*
*Review of Analytical Chemistry* **13**, 45-65 (2020).
- 78. de Wijn R, Melo DVM, Koua FHM, Mancuso AP. Potential of Time-Resolved Serial
Femtosecond Crystallography Using High Repetition Rate XFEL Sources. *Applied Sciences*
**12**, 2551 (2022).
- 79. Oghbaey S, *et al.* Fixed target combined with spectral mapping: approaching 100% hit
rates for serial crystallography. *Acta Crystallographica Section D* **72**, 944-955 (2016).
- 80. Doppler D, *et al.* Minimized Sample Consumption for Time-Resolved Serial
Crystallography Applied to the Redox Cycle of Human NQO1. *bioRxiv*,
2024.2004.2029.591466 (2024).
- 81. Zhao F-Z, *et al.* A guide to sample delivery systems for serial crystallography. *The FEBS*
*Journal* **286**, 4402-4417 (2019).
- 82. Nam KH. Sample Delivery Media for Serial Crystallography. *International Journal of*
*Molecular Sciences* **20**, 1094 (2019).

[revised manuscript text omitted]
 (4) dioxigenase	(1) 24 (3) 10 (4) 5-7	1-3: 10 ¹¹ 4: 10 ⁸ -10 ⁹	(1) 0.8 μm (2) 0.4x0.2x0.2 μm (3) 5-10 μm (4)	(1) 450μL*/ 10.8mg (3) 765μL*/ 7.65mg (4) 344μL*/ 2.41mg	Glass DFFN

Doppler et al. ¹⁴⁶	PP	EuXFEL (10Hz, 125 pulses/train)	SPB/SFX (vacuum)	Q _r : 10-1 Q _{exp} :5	PSII	NI	NI	1-2×1-2×10-30 μm 10-30 μm	NI	3D-printed Co-flow
Jernigan et al. ¹⁴⁹	N	LCLS (120)	MFX (GDVN-He, coMESH-air)	Q _{exp} : 3-5 (coMESH) 25 (GDVN)	NendoU	40-80	NI	2x2x2 μm, 10-15 μm	NI	CoMESH & GDVN-misc

Electrospinning Injectors

Kern et al. ²¹⁶	PP	LCLS (120)	CXI (vacuum)	Q _{exp} :2.5-3.1 Q _r : 0.14-3.1	PSII	7.4	NI	5-10 μm	NI	MESH
Sierra et al. ¹⁴⁸	N	LCLS (120)	CXI (vacuum)	Q _{exp} : 0.17	Thermolysin	NI	2x10 ¹⁰	2.5x4-3.5x7 μm	NI/0.14mg	MESH
Sierra et al. ¹³⁰	misc	LCLS (120)	CXI (vacuum)	Q _{exp} : 0.75-3	(1) PSII (2) paromomycin complex (3) Thermolysin	(1)70, (3) 25	2: 10 ¹⁰ -10 ¹¹	(1) 5-15 μm (2) 3-10x3-10x20-30 μm (3) 3x5 μm	(2) 360μL/NI	CoMESH
Dao et al. ¹⁵³	misc	LCLS (120)	CXI (vacuum)	NI	Thermus thermophilus	NI	10 ¹⁰ -10 ¹¹	2x2x4 μm	500μL/NI	CoMESH
Ciftci et al. ¹⁵⁴	misc	LCLS (120)	CXI (vacuum)	Q _{exp} :1-3	HIV-1	NI	10 ¹⁰ -10 ¹¹	1x1x5 μm to 5x5x15 μm	500μL/NI	CoMESH
Ishigami et al. ¹⁵²	N	LCLS (30)	MFX (He)	Q _{exp} :3	Cytochrome c oxidase	80-90	10 ¹⁰ ^b	20x20x4 μm	NI	MESH

Viscous Injection

Weierstall et al. ¹⁵⁶	N	LCLS (60-120)	CXI (vacuum)	Q _r : 0.001-0.3 Q _{exp} :0.17	(1) β2 adrenergic receptor (2) A _{2A} AR (3) glucagon receptor	(1-3) 20-50 (6) 12	NI	(5) >5 μm	<100μL/ <0.5mg	LCP Injector with LCP
---	---	---------------	-----------------	--	---	-----------------------	----	-----------	-------------------	-----------------------

					(4) 5-HT _{2B} (5) SMO (6) DgkA						
Liu et al. ¹⁵⁷	N	LCLS (120)	CXI (vacuum)	Q _r : 0.003- 0.3	5-HT _{2B}	NI	NI	5x5x5 μm	NI/0.3mg	LCP Injector with LCP	
Sugahara et al. ¹⁶¹	N	SACLA (30)	BL3 - EH4 (He)	Q _r : 0.12- 0.48 Q _{exp} : (1,2,4) 0.48 (3) 0.46	(1) HEWL (2) glucose isomerase (3) thaumatin (4) fatty acid- binding protein type 3	NI	(1) 6x10 ⁷ (2) 2x10 ⁷ (3) 1x10 ⁷ (4) 0.9x10 ⁷	(1) 7-10 μm (2) 10-30 μm (3) 10-30 μm (4) 10-20 μm 10x10x30 μm	(1) NI/2.4mg, (2) NI/0.7mg (3) NI/0.28mg (4) NI/0.15mg	Needle Extrusion with mineral oil-grease media	
Botha et al. ¹⁶⁷	N	SLS (10)	PXII (He‡)	Q _{exp} : 0.021	HEWL	30-60	NI	15x15x60 μm	NI	HVE with LCP and Vasoline media	
Conrad et al. ¹⁶⁴	N	LCLS (120‡)	CXI (vacuum, He)	Q _{exp} : 1.6	(1) Phycocyanin (2) PS I (3) PS II	(1) 15	(1-3) 2x10 ¹⁰	(1) 1-5 μm	(1) 0.02μL* / 0.3mg	LCP Injector with Agarose media Needle extruder with Super Lube grease and hyaluronic acid media	
Sugahara et al. ⁵⁵	N	SACLA (30)	BL3 - EH4 (He)	Q _{exp} : 0.48	(1) Proteinase K (2) HEWL	40	(1,2) 6.7x10 ⁷	(1,2) 5-10 μm	30μL/1.2mg*		
Kovacsova et al. ¹⁶⁵	N	SLS (10)	PXII (He‡)	Q _r : (1,2) 0.06- 0.15 (3) 0.3-0.15 (4) 0.09-0.15	(1) Lyophilized thermolysin (2) Glucose isomerase (3) HEWL (4) Bacteriorhodopsin	(1) 25 (2) 80 (3) 30-60 (4) 35-50	NI	(1) 50- 130x5- 10x5-10 μm (2) 10- 15x10- 15x10-15 μm (3) 30x20x20 μm (4) 20x50x2 μm	(1) 20μL* / 0.5mg (2) 6μL* / 0.5mg (3) 8μL* / 0.5mg (4) 10μL* / 0.5mg	HVE with sodium carboxy- methyl cellulose and Pluronic media	

Sugahara et al. ¹⁶⁹	N	SACLA (30)	BL3 - EH4 (He)	Q _{exp} : (1) 0.42-0.75 (2) 0.47 (3) 0.38-0.47	(1) HEWL (2) thaumatin (3) proteinase K	(1) 20 (2,3) 40	(1) 1.7x10 ⁷ - 5.8x10 ⁸ (2) 4.3x10 ⁸ (3) 4.9-9.3x10 ⁷	(1) 1x1x1 μm, 5x5x5 μm, 20x20x30 μm (2) 2x2x4 μm (3) 4x4x4 - 5x5x7 μm, 8x8x8 - 12x12x12 μm	(1) 105μL*/2.1mg (2) 12.5μL*/0.5mg (3) 35.8μL*/1.4mg	HVC injector with nuclear grease
Shimazu et al. ¹⁷⁰	N	SACLA (30)	BL3 - EH4 (He)	Q _r : 0.1-5.6 Q _{exp} : (1) 0.24 (2) 0.42	(1) A _{2A} AR (2) HEWL	(1) 50 (2) 20	(2) 2.3x10 ⁸	(1) 20x3x3 μm (2) ~5 μm	(1) 30μL/1.5mg* (2) 40μL/0.8mg*	HVC injector with LCP and nuclear grade grease HVC injector with (i) Paraffin grease (ii) DATPE grease (iii) Nuclear grease (iv) Cellulose (v) Frozen Paraffin grease (vi) Paraffin grease (native) (vii) DATPE grease (pr-derivative) 3D-printed injector with LCP media HVC with hydroxyethyl
Sugahara et al. ¹⁶²	N	SACLA (30)	BL2 (He)	Q _{exp} : 0.24, 0.11	proteinase K		(i,v) 9.4x10 ⁷ (ii, iii, iv) 1.4x10 ⁸ (vi, vii) 7.2x10 ⁸	(i-v) 2x2 μm (vi-vii) 0.8x0.8 μm	(i-v) 20μL/0.8mg* (vi-vii) 8μL/0.32mg*	
Vakili et al. ²¹⁷	MISC	EuXFEL (10Hz, 1 pulse/ train)	SPB/SFX (vacuum)	Q _r : 0.11-0.36	iq-mEmerald protein mixed with CuCl ₂	50	NI	5x15 μm	NI	
Wolff et al. ¹⁶⁶	PP	SACLA (30)	BL2 (He†)	Q _{exp} : 2.5	HEWL	20	NI	NI	NI	

cellulose
media

Table 4: Overview of droplet injection systems in the literature including details on X-ray source, flow rates, analyte, protein and particle concentration, crystal size, and reported sample consumption. Sample consumption is recorded in volume and protein amount in milligrams. Time-resolved experiments in the second column are classified into pump-probe (PP) for light-induced, mix and inject (misc), and N for not time-resolved. ‡The conditions in which the experiment was conducted are not explicitly stated and the recorded condition is assumed based on the X-ray instrument and end-station used. Q_{exp} : The flow rates listed were used for data collection. Q_r : The flow rates recorded were given in a range. *The sample consumption value was estimated from the reported protein concentration in the mother liquor considering that all protein molecules were converted to crystals considering reported volumes consumed or amount of protein consumed. NI refers to data not included in the publication. N/A refers to data not applicable to the experimental type reported in the manuscript. The values listed in the table (symbol a) are derived from a citation within the manuscript providing the crystallization conditions.²⁷

Authors	TR* (pp, misc , N)	X-ray Source (freq)	End-station (environme nt)	Flow Rate ($\mu\text{L}/\text{min}$)	Analyte(s)	Protein Conc. (mg/mL)	Particle Conc. (part/mL)	Crystal Size	Sample Consumption (volume [μL]/ Amount [mg])	Injector Comment/Dro plet Size
Echelmeier et al. ¹⁷²	N	LCLS (120‡)	CXI (vacuum)	Q_{exp} : 5.5	(1)Fluorescein droplets (2) granulovirus	NI	NI	NI	NI	glass GDVN + PDMS segmented drop/NI
Mafune et al. ¹⁴⁷	N	SACLA (30)	BL3 - EH4 (He)	Q_{exp} : 0.5	HEWL	NI	3.2×10^7 to 3.2×10^8	5	NI/0.3mg	Piezo-driven/ 0.268 nL
Roessler et al. ¹⁷¹	N	LCLS (60)	XPP (air)	N/A	(1) HEWL, (2) thermolysin, (3) stachydrine demethylase, (4) MauG mixed with MADH (5) hemoglobin (6) PSII	(1) 70 (2) 30 (5) 18 (6) 20-40	NI	(1) 5-10 μm (2) 10-100 μm (3) 25-50 μm , 50-100 μm , 200-300 μm (4) 10-50 μm (5) 50-250 μm (6) 50-100 μm , 150-400 μm	NI	ADE and levitation/ 0.1-2.5 nL

Awel et al. ¹⁸⁰	N	LCLS (120†)	CXI (vacuum)	Q _{exp} : 2.7-3.5	granulovirus	N/A	3x10 ¹¹	0.2 x 0.2 x 0.37 μm	NI	Ceramic aerosol/0.03 pL
Echelmeier et al. ¹⁷³	N	LCLS (120)	MFX (He)	Q _r : 0.5-20	PSI	1 ^a	10 ^{9 a}	0.2-1 μm	NI	Segmented Drop/ <1 nL
Echelmeier et al. ¹⁷⁴	N	EuXFEL (10Hz, 15 pulses/train)	SPB/SFX (vacuum)	Q _r : 3-20	KDO8PS	21	5x10 ⁹	8-10 μm	962μL/20mg*	Segmented Drop/70-800 pL
Sonker et al. ⁷⁰	N	LCLS (120)	MFX (He)	Q _{exp} :4.1 - 5.1	(1) KDO8PS (2) HEWL	(1) 20 (2) 126	(1) 2x10 ⁴	(2) 5-10 μm	NI	Segmented Drop/NI
Doppler et al. ¹⁷⁶	N	LCLS (120)	MFX (He)	Q _{exp} : 3-4	(1) NQO1 (2) Phycocyanin	(1) 25 (2) 50	NI	(1) 10x2x2 μm (2) 5-15 μm	NI	Segmented Drop/2.3 nL
Perrett et al. ⁶⁸	N	EuXFEL (10Hz, 16 pulses/train)	FXE (He)	NI	HEWL	NI	3-6x10 ⁷	NI	NI	DoD/NI
Doppler et al. ⁸⁰	MIS C	EuXFEL (10Hz, 300 pulses/train)	SPB/SFX (vacuum)	Q _{exp} : 0.5 - 4.9	NQO1+NADH	18 to 26.5	2x10 ⁷	5 x 40 μm	228μL/4.9mg	Segmented Drop/2.5 nL

Table 5: Overview of Hybrid Sample Delivery Methods in the literature including details on X-ray source, flow rates, tape speeds, analyte, protein and particle concentration, crystal size, and reported sample consumption. Sample consumption is recorded in volume and protein amount in milligrams. Time-resolved experiments in the second column are classified into pump-probe (PP) for light-induced, mix and inject (misc), oxygen evolution reactions (O₂), and N for not time-resolved. ‡The conditions in which the experiment was conducted are not explicitly stated and the recorded condition is assumed based on the X-ray instrument and end-station used. Q_{exp}: The flow rates listed were used for data collection. Q_r: The flow rates recorded were given in a range. *The sample consumption value was estimated from the reported protein concentration in the mother liquor considering that all protein molecules were converted to crystals considering reported volumes consumed or amount of protein consumed. NI refers to data not included in the manuscript. N/A refers to data not applicable to the experimental type reported in the manuscript.

Authors	TR* (pp, mix, O ₂ , N)	X-ray Source (freq)	End-station (environment)	Flow Rate (μ L/min)	Analyte(s)	Protein Conc. (mg/mL)	Particle Conc. (part/mL)	Crystal Size	Sample Consumption (volume [μ L]/ Amount [mg])	Tape Speed (mm/s)
Tape-Drive										
Beyerlein et al. ⁷¹	MIX	PETRA III (25)	P11 (He‡)	Q _{exp} = 0.6	HEWL	126	NI	6-8 μ m	300 μ L/18.9mg	0.6
Zielinski et al. ¹⁸⁸	N	PETRA III (25)	P11 (He‡)	Q _{exp} = (1) 2 (2,3) 1	(1) β -lactamase (2) GH11 xylanase (3) Urate oxidase	(1) 22 (2) 15 (3) 20	NI	(1) 11-15 μ m (2) 10-20 μ m (3) 3-20 μ m, 400x400x300 μ m	(1) 91.4 μ L/ 1.76mg*	1
Lee et al. ²⁰¹	N	PAL (30)	NCI (He‡)	Q _{exp} = 0.05–0.1 Q _r = 0.05 - 10	HEWL	50	NI	10x10x10 μ m	NI	1.5
Henkel et al. ²⁰³	MIX	PETRA III (130)	P11 (He‡)	1	HEWL	126	NI	38x38 μ m	NI	1

Drop-on-Tape

Young et al. ¹⁹⁰	PP	LCLS (10, 120)	CXI (vacuum‡) XPP (He‡) MFX (He‡)	NI	Photosystem II	NI	NI	20-50 μm	NI	NI
Fuller et al. ¹⁸⁴	PP/O ₂	LCLS (10, 30, 60)	XPP (He) MFX (He)	N/A	(1) Photosystem II (2) Phytochrome (PSM) (3) Phytochrome (PAS-GAF) (4) RNR	(1) NI (2) 40 (3) 40 (4) NI	NI	(1) 20-50 μm (2) 100μm (3) 50μm (4) 20-30μm	(1) NI/2.4mg (2) NI/6.1mg (3) NI/2.9mg (4) NI/6.9mg	NI
Kern et al. ¹⁵¹	PP	LCLS (10)	MFX (He‡)	NI	Photosystem II	NI	NI	20-60μm	NI	NI
Burgie et al. ²¹⁸	N	LCLS (10)	MFX (He)	Q _{exp} = 2.5-3	TePixJ(GAF)	25	1.2x10 ⁷	NI	NI	300
Ibrahim et al. ²¹⁹	PP	LCLS (20) SACLA (30)	MFX (He‡) BL2-EH3 (He‡)	NI	Photosystem II	40-50	NI	20-60μm	NI	NI
Srinivas et al. ¹⁹⁷	O ₂	SACLA (30) LCLS (20) PAL (15)	NI (NI) MFX (He) NCI (He)	Q _{exp} (grease) = 20-80 Q _{exp} (DOT) = 8	sMMOH:MMOB	NI	NI	20-30μm	NI	300
Rabe et al. ¹⁹⁶	O ₂	LCLS (30) SACLA (30)	MFX (He) BL2 (He‡)	Q _{exp} (ADE- DOT) = 7 (v-ext) = 1-1.5	isopenicillin N synthase	50-52	5 × 10 ⁷	3x3x40-60μm	NI	50, 93.75, 187.5, 300, 375
Ohmer et al. ¹⁹⁴	N	LCLS (30‡)	MFX (He‡)	NI	MCRred1-silent	40	NI	40-80μm	NI	NI
Hussein et al. ²²⁰	PP	LCLS (20)	MFX (He‡)	NI	Photosystem II	NI	NI	20-60μm	NI	NI

Bhowmick et al. ²²¹	PP	LCLS (10,20) SALCA (20)	MFX (He†) BL2 (He†)	NI	Photosystem II	NI	NI	20-60µm	NI	NI
Matika et al. ²²²	PP, O ₂	LCLS (30)	MFX (He†)	Q _r = 7-9	(1) Photosystem II, (2) isopenicillin N synthase	2) 50-52	(1) 2x10 ⁶ (2) 2x10 ⁷	(1) 20-50µm (2) 40-60µm	NI	200-300
Lebrette et al. ¹⁹⁵	N	LCLS (30)	MFX (He†)	Q _{exp} = 9	Class Ie R2 Protein Radical	13	NI	10x10x5 µm	316.8µL*/ 4.1mg*	300
Butryn et al. ¹⁸⁷	MIX	SACLA (30)	BL2 (He†)	N/A	(1) HEWL mixed with GlcNAc (2) CTX-M-15 mixed with ertapenem	(1) 50 (2) 20	(1) ~10 ⁷ (2) 8x10 ⁷	(1) 3-5µm (2) 5x10-20µm	(1) 932µL/ 42.6mg (2) 995µL/8mg	(1) 300, 100, 30 (2) 100, 30
Nguyen et al. ¹⁹⁹	MIX	LCLS (30)	MFX† (He†)	Q _{exp} = 3.6	CYP121	10-14	NI	30µm	NI	200
Other Hybrid Methods										
Mathew et al. ²⁰⁶	N	LCLS (10)	XPP	N/A	Polyketide synthases	NI	NI	50x10x2 µm	300µL/3.3mg	2.5
Mehrabi et al. ¹⁹³	MIX	PETRA III (25†)	P14 (air†) P14-2 (He†)	NI	(1) xylose isomerase (2) HEWL mixed with GlcNAc	(1) 80 (2) 67	NI	NI	NI	NI